# Meteoric water and glacial melt in the Southeast Amundsen Sea: A timeseries from 1994-2020

Andrew N. Hennig[1], David A. Mucciarone[2], Stanley S. Jacobs[3], Richard A. Mortlock[4], Robert B. Dunbar[2]

[1]Earth System Science, Stanford University, Stanford, California, 94305, USA
[2]Oceans, Stanford University, Stanford, California, 94305, USA
[3]Lamont-Doherty Earth Observatory, Columbia University, Palisades, New York, 10964, USA
[4]Earth and Planetary Sciences, Rutgers University, Piscataway, New Jersey, 08854, USA

*Correspondence to*: Andrew N. Hennig (ahennig@stanford.edu)

**Abstract.** Ice sheet mass loss from Antarctica is greatest in the Amundsen Sea sector, where 'warm' modified Circumpolar Deep Water moves onto the continental shelf and melts and thins the bases of ice shelves hundreds of meters below the sea surface. We use nearly 1000 paired salinity and oxygen isotope analyses of seawater samples collected on seven expeditions from 1994 to 2020 to produce a time series of glacial meltwater inventory for the Southeast Amundsen Sea continental shelf. Deep water column salinity-$\delta^{18}O$ relationships yield freshwater endmember $\delta^{18}O$ values from -31.3±1.0‰ to -28.4±1.0‰, consistent with the isotopic composition of local glacial ice. We use a 2-component meteoric water endmember approach that accounts for precipitation in the upper water column, and a pure glacial meteoric water endmember for the deep water column. Meteoric water inventories are comprised of nearly pure glacial meltwater in deep shelf waters, and >74% glacial meltwater in the upper water column. Total meteoric water inventories range from 8.1±0.7 m to 9.6±0.8 m and exhibit greater interannual variability than trend over the study period, based on the available data. The relatively long residence time in the SE Amundsen Sea allows changes in mean meteoric water inventories to diagnose large changes in local melt rates, and improved understanding of regional circulation could produce well-constrained glacial meltwater fluxes. The 2-component meteoric endmember technique improves the accuracy of the sea ice melt and meteoric fractions estimated from seawater $\delta^{18}O$ measurements throughout the entire water column and increases the utility for the broader application of these estimates.

## 1   Introduction

Four decades of observations show significant and increasing glacial mass loss from Antarctica (Rignot et al., 2011; Velicogna et al., 2014; Rignot et al., 2019). A Special Report on the Ocean and Cryosphere in a Changing Climate (SROCC) projected 0.61 m to 1.10 m of sea level rise (SLR) by 2100 under RCP8.5 forcing, with uncertainty largely hinging on the future of the Antarctic ice sheet (IPCC, 2022). Over the past two decades, losses from the West Antarctic Ice Sheet (WAIS) comprised 84±12% of the total Antarctic contribution to SLR (5.5±2.2 mm from 1993-2018; WCRP Global Sea Level Budget

Group, 2018), with glaciers flowing into the Amundsen Sea Sector (particularly the Pine Island and Thwaites glaciers) dominating the overall negative mass balance of the ice sheet (Rignot et al., 2019; Shepherd et al., 2019).

High ice shelf basal melt rates in the Southeast (SE) Amundsen Sea have been linked to the flow of
40 'warm' and salty modified Circumpolar Deep Water (mCDW) onto the continental shelf, separated by cooler but fresher waters above by a thermocline between 300 m and 700 m (Dutrieux et al., 2014; Jacobs et al., 2011). mCDW flows from the continental shelf break towards SE Amundsen Sea ice shelves via "central" and "eastern" glacially-carved bathymetric troughs (Nakayama et al., 2013). This 'warm' mCDW penetrates into sub-ice shelf cavities (Jacobs et al., 1996; Paolo et al., 2015; Pritchard et
al., 2012) where it can access ice shelf grounding lines (Rignot and Jacobs, 2002). To access the Pine Island Ice Shelf (PIIS) grounding line, mCDW passes between the bottom of the ice shelf at ~350 m and a seafloor ridge at ~700 m (Jenkins et al., 2010). Basal melt is driven by total heat transport, dependent more on the thickness of the mCDW layer transported on-shore than its temperature (Dutrieux et al., 2014; Jenkins et al., 2018), with the thickness controlled by local wind forcing of a shelf break
undercurrent, in turn influenced by the Amundsen Sea Low. Despite the strong sensitivity of these ice shelves to ocean forcing, and evidence of increasing mass loss in this region, estimates of Antarctic SLR contributions from basal melt remain poorly constrained (van der Linden et al., 2021, 2023).

Southern Ocean water masses have typically have been differentiated and defined by measurements of
55 temperature and salinity, and less often by including oxygen isotopes ($\delta^{18}O$; Jacobs et al., 1985, 2002; Meredith et al., 2008, 2010, 2013; Brown et al., 2014; Randall-Goodwin et al., 2015; Silvano et al., 2018; Biddle et al., 2019). Salinity- $\delta^{18}O$ relationships can be used to infer the source and concentration of highly $\delta^{18}O$-depleted glacial meltwater from seawater properties (Jacobs et al., 1985; Hellmer et al., 1998; Jacobs et al., 2002; Meredith et al., 2008; Randall-Goodwin et al., 2015). A spatial and temporal
array of T, S and $\delta^{18}O$ can be utilized to track glacial meltwater (GMW) content and distribution, especially in nearshore waters adjacent to melting ice shelves. Prior studies have used $\delta^{18}O$ measurements to estimate meteoric water (precipitation and GMW) abundance in the Amundsen Sea (Biddle et al., 2019; Jeon et al., 2021; Randall-Goodwin et al., 2015) and elsewhere around Antarctica (Meredith et al., 2008, 2010, 2013; Brown and Edmunds, 2016; Silvano et al., 2018) but so far have
revealed little about temporal variability or possible trends in meteoric water content. Here, we use nearly 1000 seawater isotope samples collected during seven austral summers from 1994 to 2020 (**Figure 1**) to investigate meteoric water sources, water column inventories, and their interannual variability in the SE Amundsen Sea.

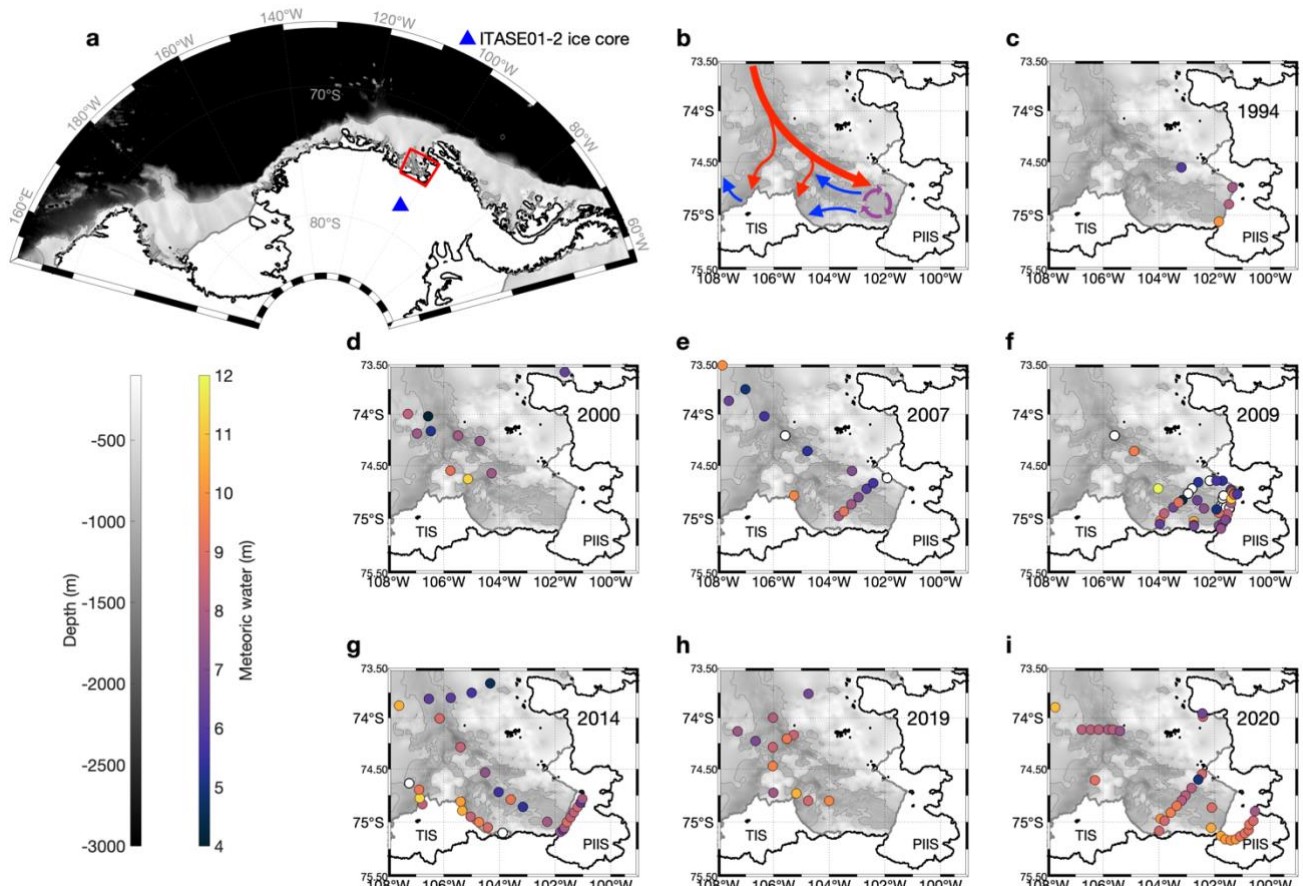

**Figure 1: Study area bathymetry, circulation, and δ¹⁸O sampling locations each of 7 years between 1994 and 2020.**
800 m isobaths are shown as thin gray lines. (a) SE Amundsen study area, and location of the ITASE01-2 ice core. (b)
Location of the Pine Island Bay gyre (purple), pathways of warm deep mCDW (red) toward the ice shelves and pathways of
shallower meltwater rich waters (blue) from beneath Pine Island Ice Shelf (PIIS) (Nakayama et al., 2019; Wåhlin et al.,
2021; Dotto et al., 2022). (c-i) Colored dots show sample locations, with colors representing depth-integrated glacial
meltwater inventories between 0 m and 800 m from 1994 to 2020. Thick gray lines indicate seaward boundaries of Thwaites
Ice Shelf (TIS) and the PIIS. Calving fronts are referenced to 2000 (Schaffer et al., 2014; Fretwell et al., 2013), a relatively
stable location before a ~20 km retreat following calving events between 2017 and 2020 (Joughin et al., 2021a). Stations
where sampling did not extend to the seafloor show only partial water column inventories, and stations shown as white dots
(2007, 2009, 2014) had two or fewer depths sampled. In 2000, 2007, and 2019, access to sampling along the front of PIIS
calving front was precluded by fast ice.

## 2    Data and Methods

### 2.1    Sample collection and analysis

We compile data from samples collected during 7 field seasons in the SE Amundsen Sea from 1994 to
2020 (**Figure 1**, **Table 1**). Salinity profiles were obtained using calibrated conductivity cells on SBE911

conductivity-temperature-depth (CTD) instruments, monitored with shipboard bottle sample analyses using Guildline AutoSal and PortaSal salinometers calibrated with IAPSO seawater salinity standards. $\delta^{18}O$ samples from 2019 and 2020 were collected in glass serum vials capped with rubber stoppers and aluminum seals (**Appendix A8**), which internal lab data demonstrate the maintenance of seawater $\delta^{18}O$ sample integrity for 5+ years. In all other years, samples were collected in bottles with taped (2009) or parafilm-wrapped (2014), threaded caps. 2009 and 2014 samples were stored for several years before $\delta^{18}O$ analysis.

In 1994, 2007, 2009 and 2014, $\delta^{18}O$ was measured using an Isotope Ratio Mass Spectrometer (IRMS; Micromass Optima Multiprep or a Finnigan MAT252 HDO). All samples collected in 2019 and 2020, and some in 2007 and 2009, were measured with a Picarro L2140-i Cavity Ring Down System (CRDS). Equivalence has been demonstrated between CRDS and IRMS measurements (Walker et al., 2016; **Appendix A7**). In all cases, values are reported as per mil(‰) deviations($\delta$), relative to Vienna Standard Mean Ocean Water (VSMOW2; Coplen, 1994).

**Table 1: Summary of $\delta^{18}O$ data sources, sampling intervals, methods & applications**

| Year | Cruise | Sample collection dates | # Samples | $\delta^{18}O$ Technique(s) |
|------|--------|------------------------|-----------|------------------|
| **1994** | NBP94-02 (Hellmer et al., 1998) | 14 Mar. 1994 – 15 Mar. 1994 | 26 | IRMS $CO_2$ equilibration |
| **2000** | NBP00-01 (Jacobs et al., 2002) | 16 Mar. 2000 – 20 Mar. 2000 | 62 | IRMS $CO_2$ equilibration |
| **2007** | NBP07-02 | 24 Feb. 2007 – 27 Feb. 2007 | 74 | IRMS $CO_2$ equilibration, CRDS |
| **2009** | NBP09-01 | 16 Jan. 2009 – 29 Jan. 2009 | 175 | IRMS $CO_2$ equilibration, CRDS |
| **2014** | iSTAR2014 (Biddle et al., 2019) | 5 Feb. 2014 – 20 Feb. 2014 | 213 | IRMS $CO_2$ equilibration |
| **2019** | NBP19-01 | 12 Jan 2019 – 14 Jan 2019 | 107 | CRDS |
| **2020** | NBP20-02 | 5 Feb. 2020 – 8 Mar. 2020 | 280 | CRDS |

Some of the 2009 samples were processed at Rutgers University in 2010 using a Micromass IRMS; the remainder in 2020 using a Picarro CRDS system at Stanford University. The latter samples were subject to extensive quality control before being included in this study. (**Appendix A6**). A subset of 100 samples from 2019 and 2020 were processed concurrently using $CO_2$ equilibration on a Finnigan MAT252 IRMS and CRDS via vaporizer to ensure data comparability between instrumentation (**Appendix A7**). Measurements for all years achieved a precision of 0.04‰ for IRMS and 0.02‰ for CRDS, based on replicate analyses.

After a review of the literature, we considered a possible salt effect in measured seawater $\delta^{18}O$, as suggested by a small number of studies (Lécuyer et al., 2009; Skrzypek and Ford, 2014; Benetti et al., 2017). As no salt effect offset was applied to the previously published data in this study (1994, 2000,

2014) we have not applied any offset to data from other years. The mCDW $\delta^{18}$O value (**Table 2**) for 2014 is significantly higher than other years (**Appendix A2**) – likely due to a calibration offset but may
also point to sample storage issues. The mCDW and meteoric water endmembers are defined from observations each year, minimizing the impact of interlaboratory offsets on the results (**Appendix A2, Data and Methods 2.2**).

## 2.2 Three-endmember mixing model

We adapt an approach from Östlund & Hut (1984) as applied in the Peninsula-Bellingshausen-
120 Amundsen region of West Antarctica (Biddle et al., 2019; Jeon et al., 2021; Randall-Goodwin et al., 2015; Meredith et al., 2010) and near the Totten Ice Shelf (Silvano et al., 2018). We use a 3-endmember mixing model (**Equations 1-3**) to determine water source fractions in the field area. The model assumes the observed $\delta^{18}$O and salinity values result from mixtures of mCDW, sea ice melting/freezing, and meteoric waters contributing a range of $\delta^{18}$O and salinity signatures. Model outputs (mCDW, meteoric
water, and sea ice melt fractions) critically depend on appropriate endmember inputs, which will affect resulting water source fractions and interpretation of any changes through time. To minimize issues that could arise from inter-laboratory calibration offsets (**Data and Methods 2.1, Appendix A2**), we define mCDW and meteoric water endmembers separately for each year (**Table 2**).

**Equations 1-3: Three-endmember mixing model.** The 3-endmember mixing model uses the absolute salinity and $\delta^{18}$O of mCDW, sea ice melt, and meteoric water endpoints to solve for the relative fractions of the three water sources in each sample analyzed.

$$f_{sim} + f_{met} + f_{mcdw} = 1 \qquad\qquad (1)$$
$$f_{sim} * S_{sim} + f_{met} * S_{met} + f_{mcdw} * S_{mcdw} = S_{obs} \qquad\qquad (2)$$
$$f_{sim} * \delta_{sim} + f_{met} * \delta_{met} + f_{mcdw} * \delta_{mcdw} = \delta_{obs} \qquad\qquad (3)$$

$f$ = fraction of water source
$S$ = salinity
$\delta$ = $\delta^{18}$O
$sim$ = sea ice melt
$met$ = meteoric water
$mcdw$ = modified circumpolar deep water
$obs$ = observed sample

The mCDW is the warmest, saltiest, and least $\delta^{18}$O-depleted water mass in the region and comprises the
145 vast majority of the overall water column. Interannual changes in mCDW inflow will result from variable wind forcing (Dotto et al., 2019; Holland et al., 2019; Kim et al., 2021), combined with on-shelf lateral and vertical mixing. In the 3-endmember mixing model, mCDW is defined by the mixing line of data >200 m; with mCDW the $\delta^{18}$O value at the salinity maximum (Biddle et al., 2017) (**Results 3.1**; **Figure 2**, **Appendix A1**). Sea ice endmember isotopic values adopted from previous studies in the
150 Amundsen and Bellingshausen region (Meredith et al., 2008, 2010, 2013; Randall-Goodwin et al., 2015; Biddle et al., 2019) are based on the $\delta^{18}$O of surface water with an offset to account for isotopic fractionation due to freezing (Rohling, 2013).

The greatest endmember uncertainty is associated with the meteoric water endmember, which can
include basal melt, local and imported iceberg melt, and local and non-local precipitation. Meteoric
waters >200 m will consist almost entirely of GMW (Jenkins, 1999; Randall-Goodwin et al., 2015;
Biddle et al., 2019) and can be fingerprinted using the 0-salinity intercept of linear regressions through
salinity-$\delta^{18}$O data, as shown in **Figure 2** (Fairbanks, 1982; Paren and Potter, 1984; Potter et al., 1984;
Jacobs et al., 1985; Hellmer et al., 1998; Jenkins, 1999; Meredith et al., 2008) (**Results 3.1**). Even
meteoric water in the upper 200 m will consist primarily of GMW (Meredith et al., 2008; Bett et al.,
2020), however it is also likely to contain some fraction of precipitation with a less negative $\delta^{18}$O value.

The Pine Island/Thwaites area receives ~0.5 m/y (water equivalent) of precipitation (Boisvert et al.,
2020; Donat-Magnin et al., 2021). Local precipitation collected in 2019 at 72.5°S had a $\delta^{18}$O value of -
165 15‰, consistent with studies of precipitation at this latitude and elevation (Masson-Delmotte et al.,
2008). To calculate the most accurate meteoric water and sea ice melt fractions, we account for the
presence of precipitation in the upper water column. With 2 full years' worth of precipitation (residence
time of local deep shelf waters ~2 years; Tamsitt et al., 2021) in the upper water column, those meteoric
waters must be comprised of >74% GMW (**Appendix A4**). We use a 2-component meteoric water
endmember approach, with a 'pure' GMW endmember as calculated from the linear regression through
those $\delta^{18}$O and salinity data (**Table 2**, **Figure 2**, **Results 3.1**) used in the deep water column, and an
endmember comprised of precipitation and GMW (**Table 2**) used in the upper water column.

**Table 2: Base salinity and $\delta^{18}$O values used in the 3-endmember mixing model.** mCDW and meteoric GMW components are defined
independently using the mCDW-GMW mixing line produced from (>200 m depth) salinity and $\delta^{18}$O observations for each year, as the
salinity maximum and zero-salinity intercept, respectively (**Results 3.1; Figure 2; Appendix A1**). In the upper 200 m, a meteoric
endmember comprised of a weighted average between precipitation and GMW fraction is used (**Appendix A4**). Sea ice melt uses the same
values for each year. Salinities are reported as absolute salinity (g/kg).

| Year | mCDW salinity (g/kg) | mCDW $\delta^{18}$O (‰) | Meteoric water GMW $\delta^{18}$O (‰) | GMW fraction of meteoric water <200 m | Meteoric water precipitation $\delta^{18}$O (‰) | Effective <200 m meteoric water $\delta^{18}$O (‰) | Sea ice melt salinity (g/kg) | Sea ice melt $\delta^{18}$O (‰) |
|------|------|------|------|------|------|------|------|------|
| **1994** | 34.86 | -0.01 | -29.37 | 0.735 | | -25.56 | | |
| **2000** | 34.88 | -0.05 | -28.71 | 0.773 | | -25.60 | | |
| **2007** | 34.90 | -0.02 | -28.42 | 0.785 | | -25.53 | | |
| **2009** | 34.87 | 0.01 | -29.09 | 0.775 | -15 | -25.92 | 7 | 2.1 |
| **2014** | 34.87 | 0.08 | -31.26 | 0.756 | | -27.29 | | |
| **2019** | 34.89 | -0.09 | -30.02 | 0.759 | | -26.40 | | |
| **2020** | 34.89 | -0.10 | -29.14 | 0.781 | | -26.05 | | |

# 3    Results

## 3.1    Meteoric waters defined by the $\delta^{18}O$ – salinity relationship

Freshwater endmembers (zero-salinity $\delta^{18}O$ intercepts) over the seven sampled summers differ by
<3‰, ranging from -28.4±1.0‰ to -31.3±1.0‰ with a standard deviation of 1.0‰ across years (**Figure
2**, **Table 2**). These measurements are consistent with the nearest ice-core (ITASE01-2, from 77.84°S,
102.91°W, **Figure 1a**; Schneider et al., 2006; Steig et al., 2005) with a mass-averaged $\delta^{18}O$ value of -
29.8±1.9‰. Ice cores further east have less negative $\delta^{18}O$ values (~-20‰; Thomas et al., 2009), while
those further west are more negative (~-40‰; Blunier & Brook, 2001). Intercept standard error ranges
from ±0.3‰ in 2020 to ±1.9‰ in 2009. Endmember extrapolations from the regional salinity and $\delta^{18}O$
measurements indicate that freshwater introduced to the water column is dominated by locally derived
GMW, as intimated earlier from 1994 data (Hellmer et al., 1998). While winter water, produced during
sea-ice formation, also has a depleted $\delta^{18}O$ signature, these waters are less depleted than those
influenced by GMW (Jenkins, 1999). Aggressively removing winter water samples ($\Theta < -1.5°C$) from
the regression had no impact on the 0-salinity intercepts. The calculated intercepts were also unaffected
by using only those data falling directly on the mCDW-GMW mixing line in T-S space (distinct from
the mCDW-winter water mixing line).

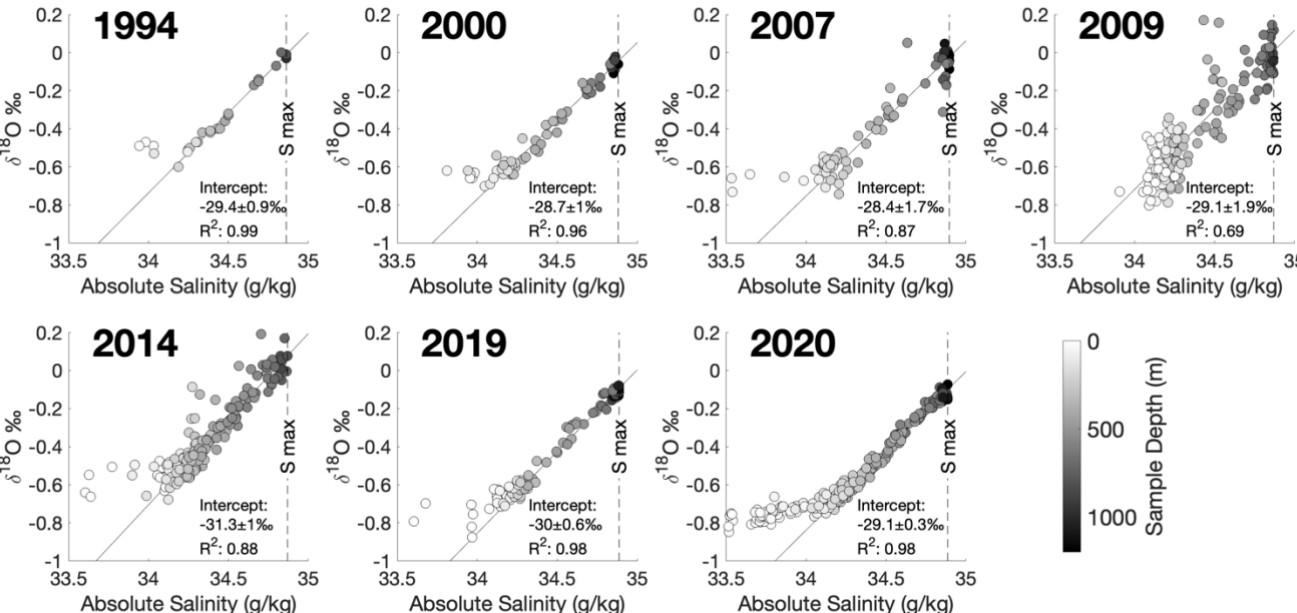

**Figure 2: Salinity vs $\delta^{18}O$ plots for each year, shaded by depth.** Linear regressions (solid) gray lines (with $R^2$ from 0.69 in 2009 to 0.99
in 1994) project to zero-salinity glacial meltwater endmember intercepts using data >200 m. Dashed vertical lines indicate the mCDW
salinity maxima (**Table 2**). Data diverge from the mCDW-meteoric water mixing line in the upper water column, where sea ice melt
freshens the resultant mixture but has an enriching effect on $\delta^{18}O$ (**Table 2**). Years with greater divergence at the surface have more sea ice
melt (**Appendix A5**). The most negative upper water column seawater $\delta^{18}O$ measurements tend to reach minima between -0.9‰ and
-0.6‰. Intercept uncertainty is the standard error of the linear regression intercept through data >200 m.

Samples below 200 m show a strong $\delta^{18}$O-salinity relationship, forming a mixing line between mCDW and a (glacial) meteoric freshwater endmember introduced at depth. Closer to the surface (from 10 m in 2009 to 160 m in 2000) data diverge from the mixing line due to the net influence of sea ice melt and local precipitation, moving the $\delta^{18}$O of the mixture in a more positive direction. Below 200 m the $\delta^{18}$O-salinity relationship is strongly linear in all years, with 1994, 2000, 2019, 2020 showing the strongest fit.

## 3.2    Vertical distribution of meteoric water illustrates basal melt

A 3-endmember mixing model of mCDW, sea ice melt, and meteoric water is used to determine the constituent freshwater components of seawater (**Equations 1-3**) at all depths sampled in the water column (**Figure 3**). By using mCDW and meteoric water endmember values based on data from each year individually, minimizing the potential impact of analytical calibration offsets between laboratories on the calculated meteoric water fractions. (**Data and Methods 2.1**; **Appendix A2**).

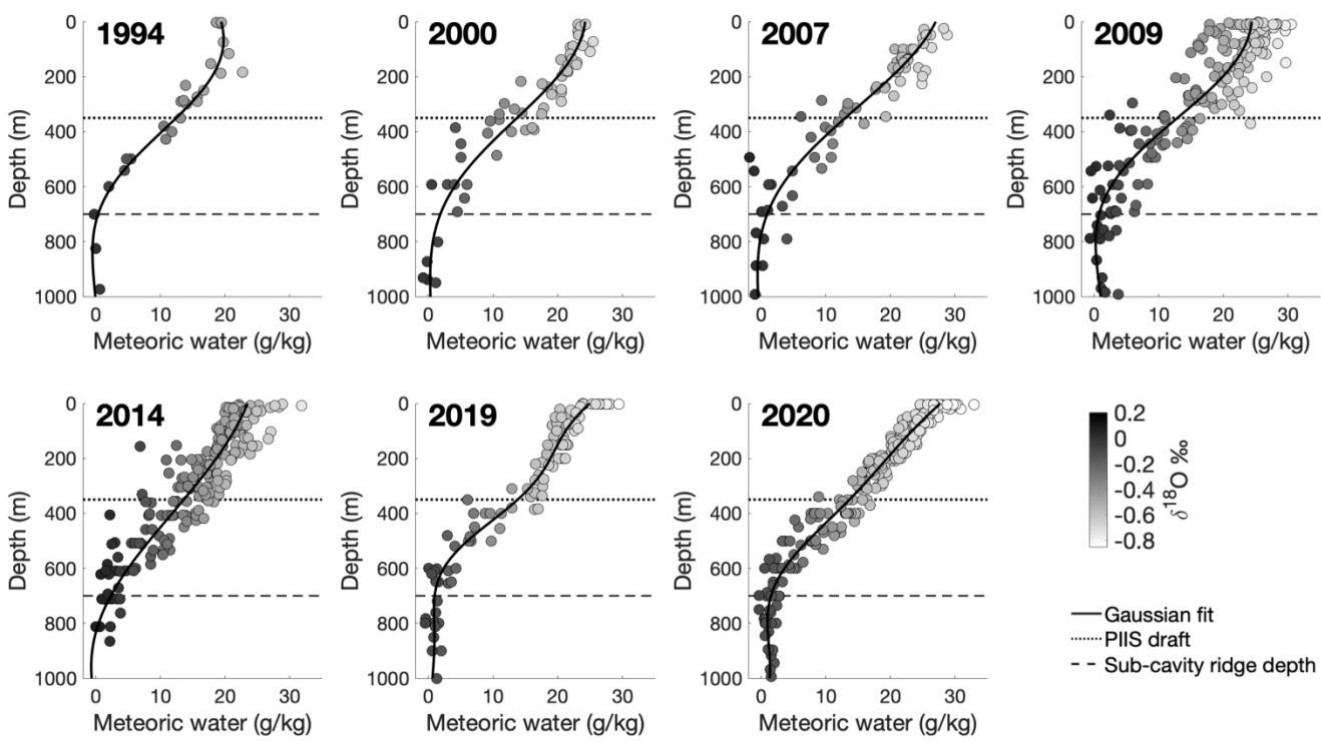

**Figure 3: Meteoric water fractions (g of meteoric water per kg of seawater) vs depth**. Shading shows $\delta^{18}$O value, and solid lines represent Gaussian Process regression fits. Dotted and dashed lines show the depth of the PIIS draft and sub-cavity ridge (Jenkins et al., 2010). Small false negative meteoric water fractions in deep waters are the result of sample salinity and/or $\delta^{18}$O values that are higher than the mCDW endmember value(s) (**Table 2; Data and Methods 2.2**), the result of spatial variability in mCDW properties (discussed in further detail in **Results 3.5**).

Evidence of highly $\delta^{18}$O-depleted freshwater is found at depths above ~700 m (the depth of PIIS sub-ice shelf ridge; Jenkins et al., 2010), with highest concentrations found at depths shallower than 350 m –

above which glacial meltwater has been observed to flow out from beneath the ice shelf (Biddle et al., 2017; Naveira Garabato et al., 2017). Made less dense by the addition of GMW, such outflows rise through denser waters above, along ice shelf calving fronts and strongly influencing surface waters in this region (Dierssen et al., 2002; Mankoff et al., 2012; Thurnherr et al., 2014; Fogwill et al., 2015). Mean integrated total meteoric water column inventories (**Table 3**) range from a low of 8.1±0.7 m in 1994 to a high of 9.6±0.8 m in 2000 and 2020, with meteoric water fraction uncertainty (described in **Results 3.5.1**) ranging from 1.4 g/kg in 2019 to 2.8 g/kg in 2009.

**Table 3: Meteoric water column inventory and uncertainty.** Depth-integrated meteoric water content using the Gaussian Process fit (**Figure 3**) between the sea surface and 800 m depth. Uncertainties are associated with instrumental precision, spatial variability of data, and endmember uncertainty (**Data and Methods 2.2**).

| Year | Meteoric water column inventory (m) | % meteoric water column inventory uncertainty | Meteoric water fraction uncertainty (g/kg) | Estimated GMW fraction of total meteoric water |
|---|---|---|---|---|
| **1994*** | 8.1±0.7 | 8.1% | 1.6 | 0.876 |
| **2000** | 9.6±0.8 | 8.6% | 1.9 | 0.896 |
| **2007** | 9.4±1.1 | 11.7% | 2.3 | 0.893 |
| **2009** | 9.0±1.1 | 12.4% | 2.8 | 0.889 |
| **2014** | 9.4±1.0 | 11.0% | 2.0 | 0.894 |
| **2019** | 8.9±0.7 | 8.0% | 1.4 | 0.887 |
| **2020** | 9.6±0.8 | 8.4% | 1.5 | 0.896 |

\* As 1994 has only 4 sampling locations, and the strongest fit of any year (**Figure 2**) its uncertainty may be artificially decreased

### 3.3 Sea ice melt

Sea ice melt fractions reach as high as 40 g/kg in the upper surface waters (2019, **Figure 4**), while sea ice melt fractions are mostly negative below 100 m, with the minima (i.e., largest negative fraction) occurring just above 200 m in most years. Negative sea ice melt fractions in subsurface waters are produced during sea ice formation (with the opposite signal produced in the near-surface when the sea ice melts), and reach as low as -13 g/kg. Larger positive sea ice melt fractions below 200 m correspond with samples with higher $\delta^{18}O$ than others at a similar depth/salinity. Total integrated mean sea ice melt (**Table 4**) is near 0 across all years, with uncertainty (described in **Results 3.5.1**) in sea ice melt fractions ranging from 1.8 g/kg in 2019 to 3.6 g/kg in 2009. Mean sea ice melt inventories are influenced largely by very high fractions near the surface, potentially the result of sea ice melt "flooding" and import of sea ice melt from areas upstream (Ackley et al., 2020).

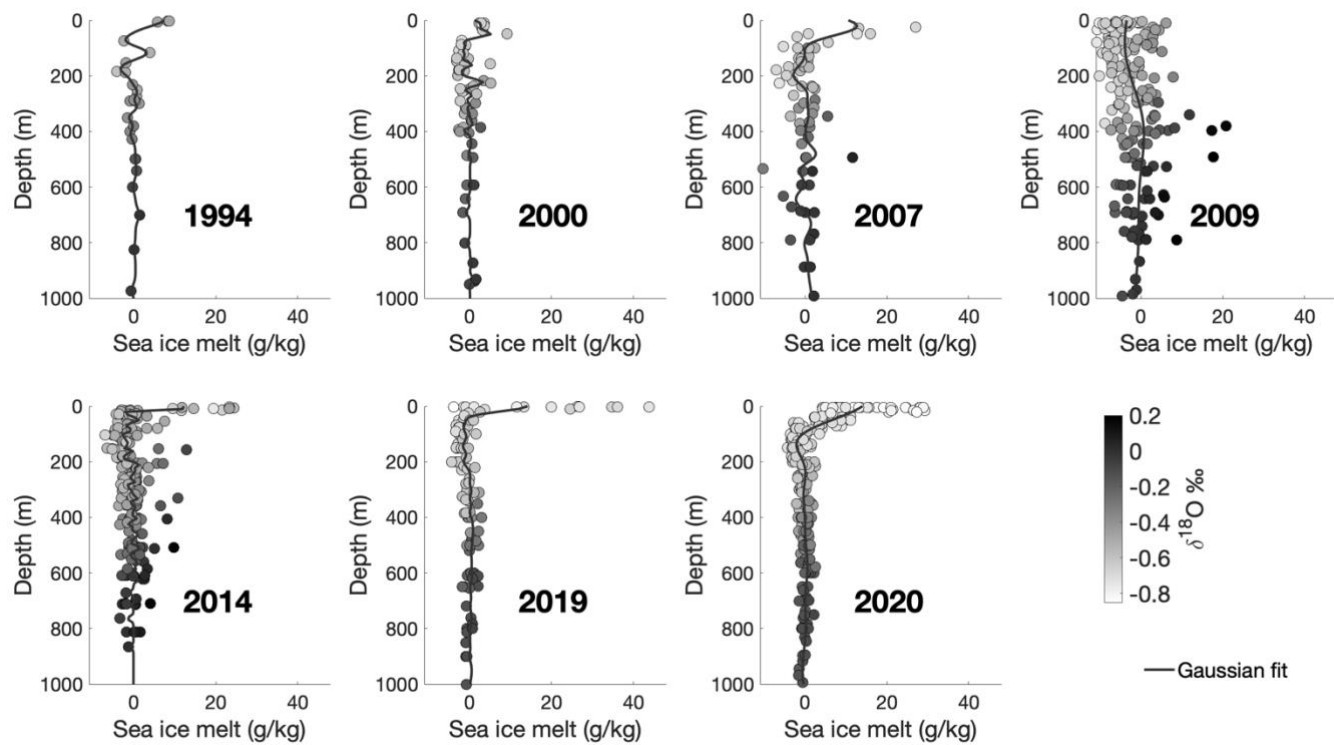

**Figure 4: Sea ice melt fractions (g of sea ice melt per kg of seawater) vs depth.** Shading shows $\delta^{18}$O value, and solid lines represent Gaussian Process regression fits. Sea ice melt fractions are highest at the surface. Negative sea ice melt fractions, occurring mostly below 100 m, are the product of earlier sea ice formation.

**Table 4: Sea ice meltwater column inventory and uncertainty.** Depth-integrated sea ice meltwater content using the Gaussian Process fit (**Figure 4**, **Figure 3**) between the sea surface and 800 m depth. Uncertainties are associated with instrumental precision, spatial variability of data, and endmember uncertainty (**Data and Methods 2.2**).

| Year | Mean sea ice melt column inventory (m) | Sea ice melt fraction uncertainty (g/kg) |
|---|---|---|
| **1994** | 0.1±0.9 | 2.1 |
| **2000** | 0.1±1.0 | 2.4 |
| **2007** | 0.6±1.5 | 2.9 |
| **2009** | -0.9±1.4 | 3.6 |
| **2014** | -0.2±1.1 | 2.5 |
| **2019** | 0.3±1.0 | 1.8 |
| **2020** | 0.5±0.9 | 1.9 |

Sea ice melt is discussed further in **Appendix A6**.

## 3.4  Average meteoric water inventory over the last two decades

Average meteoric water column inventories (**Table 3**) in the study area were estimated by depth integrating the Gaussian Process fit of the calculated meteoric water fractions (solid lines in **Figure 3**). **Figure 5** plots the mean meteoric water inventory in each year, with uncertainty described in **Uncertainty and Sensitivity Analyses 3.5.** The average meteoric water column inventory was relatively low in 1994 and higher from 2000-2020. Strongly influenced by the low meteoric water

inventory in 1994, a linear regression of the mean meteoric water inventories suggests an increase of 0.03±0.02 m/y (p-value 0.25). The increase is small— and given the magnitude of the uncertainty associated with these inventories, may be an artefact stemming from spatial sampling bias, or some fraction of a meteoric water signal imported from upstream.

These results show greater interannual variability than increasing trend in meteoric water content, however are consistent with recent modeling showing an increase in basal melt through the 1990s, followed by relative stability and interannual variability from 2000 through 2020 (Flexas et al., 2022). Meteoric water column inventories have uncertainties of <1.1 m (<12.4%), accounting for analytical precision, endmember uncertainty, and spatial variability of the model inputs (**Results 3.5**). Based on

very liberal evaluation of precipitation influence, these total meteoric water column inventories are likely to consist of >87% GMW (**Methods 2.2, Appendix A4**). While the glacial fraction of these results will include local and non-local iceberg melt, it will not include ice shelf losses via iceberg calving where icebergs melt outside of the study area.

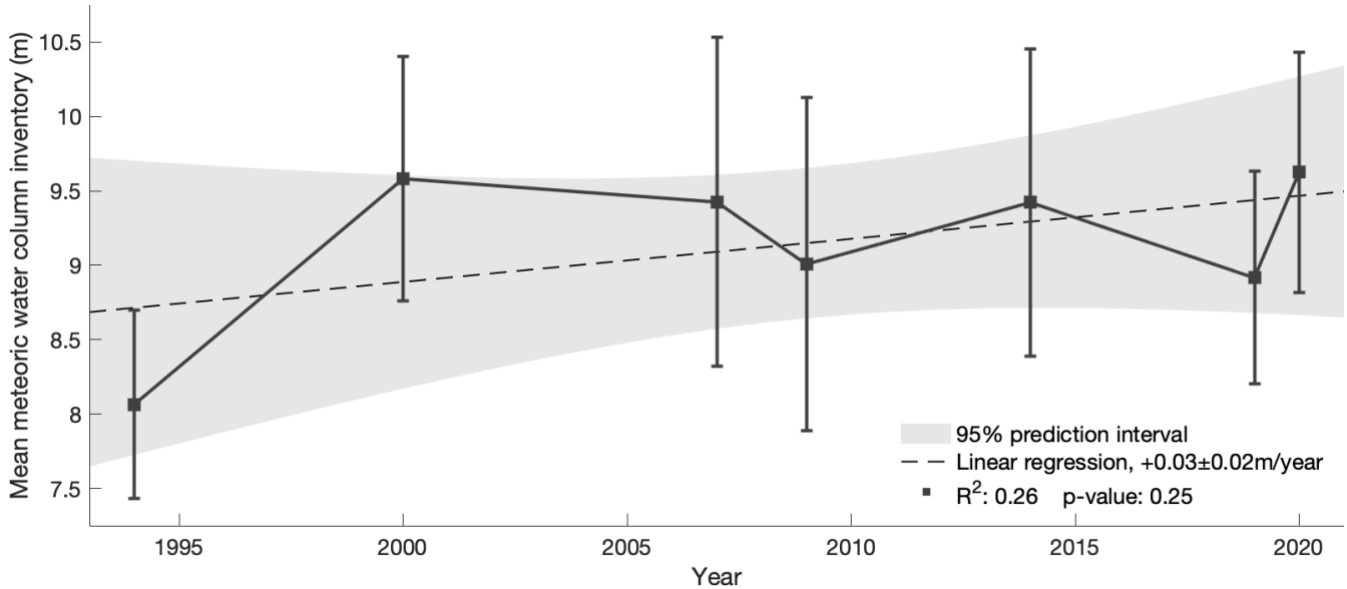

**Figure 5: Average meteoric water column inventory.** Depth-integrated meteoric water content from the Gaussian Process fit (**Figure 3**) between the sea surface and 800 m. Error bars show the uncertainty (**Results 3.5**) associated with data volume, analytical precision, and uncertainty in endmember values (**Data and Methods 2.2**). A linear regression of the mean values shows an increase of 0.03±0.02 m/year (p-value 0.25)**.** Grey shading shows the 95% prediction interval for the linear regression.

### 3.5 Uncertainty and sensitivity analyses

#### 3.5.1 Analytical precision, geographic sampling variability, and endmember uncertainty

The results of the 3-endmember mixing model depend on the inputs and endmembers used – all of which are subject to some level of uncertainty. We ran 15,000 Monte Carlo simulations for each year's data to assess the uncertainty of the results. In each simulation we select 3 stations at random. Each observation is perturbed randomly by the precision associated with analytical precision for that instrument. We determine mCDW and meteoric water endmembers from those perturbed data from the random selection of 3 stations; these two endmembers will differ with each simulation, both due to the observational data perturbations and due to the randomized station selection. We apply an additional perturbation to all three endmembers based on the uncertainty and variability associated with that endmember. In the upper 200 m of the water column, the precipitation fraction, and $\delta^{18}O$ value are perturbed independently, in addition to the perturbations made to the meteoric GMW endmember calculated from the mixing lines (**Table 2**, **Figure 2**). The greatest single source of water fraction uncertainty is endmember uncertainty, followed by geographic sampling variability. More in-depth details of the uncertainty analysis, perturbations used, and breakdowns of different uncertainty impacts can be found in **Appendix A3**.

Uncertainty in mean meteoric water fractions ranges from 1.4 g/kg in 2019 to 2.8 g/kg in 2009, and uncertainty in mean meteoric column inventories ranges from 7.9-12.4%. Calculated water fractions are most strongly influenced by changes made to the mCDW endmember (comprising ~99% of an 800 m water column on average; ~95% in surface waters rich in meteoric water and sea ice melt). Sea ice melt and meteoric water fractions vary inversely. Meteoric water fractions also vary inversely with the magnitude of the meteoric water endmember $\delta^{18}O$ (i.e. a more negative meteoric water endmember will produce smaller meteoric water fractions). 1994 has the fewest samples, but the strongest fit (**Figure 2**), so uncertainty for this year may be artificially low (**Figure 5**).

#### 3.5.2 Geographic variability by grouping

This study relied on the compilation of data collected for 6 independent studies over 7 different cruises. To determine the impact of inconsistency in sampling locations each year, we conducted a spatial sensitivity analysis by separately analyzing different spatial groups of stations across each year (**Figure 6, Table 5, Table 6**), running 5,000 Monte Carlo analyses for each group as described in **Results 3.5.1**. Uncertainty is represented as the standard deviation of those results (**Table 5, Table 6**).

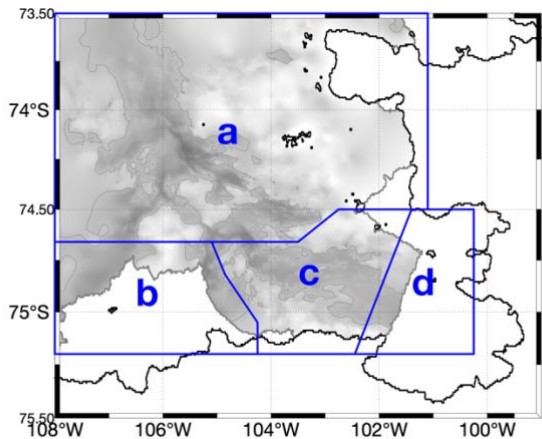

**Figure 6:Boundaries of geographic groupings used for spatial sensitivity analysis.** Blue lines show boundaries of geographic areas analyzed separately. Gray shading shows bathymetry, with isobaths drawn at 800 m. More detailed maps for each year are in Appendix A4.1.

**Table 5: Summarized results of meteoric water column inventory (m) from spatial sensitivity analysis (Figure 6).** Results by year in rows, and by area group in columns. See **Appendix A3.3** for further detail.

|      | a | b | c | d |
|------|-----------|-------------|-----------|-----------|
| **1994** | 6.8±0.5* | - | - | 8.5±0.6 |
| **2000** | 9.6±0.7 | - | - | - |
| **2007** | 8.8±0.8 | 10.6±0.8* | 9.5±0.7 | - |
| **2009** | 7.5±0.6* | | 9.1±0.8 | 8.9±0.7 |
| **2014** | 8.7±0.6 | 10.7±0.6 | 8.3±0.5 | 7.9±0.5 |
| **2019** | 8.7±0.6 | 10.4±0.7 | 8.6±0.5 | - |
| **2020** | 9.3±0.6 | - | 9.1±0.5 | 8.9±0.5 |

\* Data from only one station. For 1994, 2007, and 2009, result is based on only 8, 4, and 4 samples, respectively.

**Table 6: Summarized results of sea ice melt water column inventory (m) from spatial sensitivity analysis (Figure 6).** Results by year in rows, and by area group in columns. See **Appendix A3.3** for further detail.

|      | a | b | c | d |
|------|-----------|-------------|-----------|-----------|
| **1994** | -0.3±0.7* | - | - | 0.7±0.9 |
| **2000** | 0.1±0.9 | - | - | - |
| **2007** | 0.6±1.0 | 1.3±1.1* | 0.7±1.0 | - |
| **2009** | 0.0±0.7* | - | -1.4±1.1 | -0.7±1.0 |
| **2014** | -0.1±1.0 | 0.0±0.9 | -0.5±0.9 | -0.2±0.8 |
| **2019** | 0.1±0.7 | -0.3±0.8 | - | - |
| **2020** | 0.6±0.8 | - | 0.6±0.8 | 0.5±0.8 |

\* Data from only one station. For 1994, 2007, and 2009, result is based on only 8, 4, and 4 samples, respectively.

Grouped geographic sensitivity analyses show little spatial variability in mean water column inventories, accounting for uncertainty (**Table 5, Table 6**). Mean sea ice melt inventories are consistently ~0 m, with near-complete overlap of means and uncertainty envelopes across groupings for each year. Mean meteoric water column inventories are remarkably consistent spatially, except those

calculated from stations in Group **a** in 1994 and 2009, where very few data were available, and Group **b**
alongside the TIS, which showed significantly higher meteoric water inventories than the rest of the
study area. Only 1 and 2 stations alongside the TIS were sampled in 2007 and 2019, the average column
inventories are consistent with the 2014 data, where there were 8 stations in Group **b**. The higher
inventories in Group **b** are suggestive of an accumulation of basal melt, and consistent with findings
from another study showing that basal melt from beneath PIIS ends up along the eastern edge of TIS
(Wåhlin et al., 2021).

In some cases (e.g. 2020), the calculated mean meteoric water inventories from the individual groupings
(**Table 5**) are higher than the overall mean. The apparently lower meteoric water column inventories
result from small differences in calculated endmembers (i.e. less-salty mCDW, more negative mCDW
$\delta^{18}O$, and/or more negative meteoric water $\delta^{18}O$), as mCDW and meteoric endmembers are defined
separately for each geographic grouping in each year. The relative insensitivity of sampling location
(with the exception of those alongside TIS) in calculated mean meteoric water column inventories
suggests that precise reoccupation may be unnecessary, and potentially that a relatively small number of
strategic stations/sampling locations could be used to reliably assess mean meteoric water column
inventory in this region. (See **Appendix A3.3** for detailed results, including individual endmembers for
each grouping).

## 4   Discussion

### 4.1   The utility of zero-salinity $\delta^{18}O$ intercepts

The meteoric water endmember has been described as the least well-constrained component (Meredith
et al., 2008, 2010, 2013; Randall-Goodwin et al., 2015; Biddle et al., 2019) of the three-endmember
mixing model employed here, leading previous studies to use plausible mean meteoric $\delta^{18}O$ values,
falling between the $\delta^{18}O$ value of glacial melt and local precipitation. We use linear regressions through
salinity-$\delta^{18}O$ mixing lines to (e.g. **Figure 2**) to determine glacial freshwater endmembers (Fairbanks,
1982; Paren and Potter, 1984; Potter et al., 1984; Jacobs et al., 1985; Hellmer et al., 1998; Jenkins,
1999). The salinity-$\delta^{18}O$ data presented here (**Figure 2** ) show clear mixing lines between mCDW and
a meteoric freshwater at depths below 200 m, with a $\delta^{18}O$ value indicative of local glacial freshwater
(Steig et al., 2005; Schneider et al., 2006). The uncertainty associated with these intercepts is lower than
the mass-weighted standard deviation of the average $\delta^{18}O$ of the ITASE01-2 ice core (**Figure 2**). Winter
water, a salty, relatively $\delta^{18}O$-depleted product of sea ice formation, has a less $\delta^{18}O$-depleted signature
than those influenced by GMW (Jenkins, 1999), and a liberal removal of samples that may be
considered winter water ($\Theta < 1.5°C$) has insignificant (<0.5‰) effect on the calculated intercepts.

The 2014 data presented here was previously used in another study (Biddle et al., 2019), where they
selected -25‰ as a meteoric water $\delta^{18}O$ endmember, adopting the midpoint between the GMW and
precipitation $\delta^{18}O$ values from another study further to the west (Randall-Goodwin et al., 2015). For the
2014 data, the current study uses base meteoric $\delta^{18}O$ endmembers of -31.3‰ for deep waters, and -

27.3‰ for the upper 200 m (**Table 2**). Using a less-negative endmember (as in Biddle et al., 2019) results in higher meteoric water fractions, and lower sea ice melt (or increased sea ice formation) fractions, very likely overestimating the deep water column meteoric water content, which will consist of nearly pure GMW. We estimate that >73% of the meteoric water in the upper 200 m is comprised of
375 GMW (**Discussion 4.2**, **Appendix A4**). Determining the deep meteoric water (GMW) endmember using the better-constrained zero-salinity $\delta^{18}O$ intercept of the sample data mixing line (**Data and Methods 2.2**), and a separate shallow meteoric endmember (incorporating a realistic precipitation fraction) provides more accurate estimates of both meteoric water and sea ice melt fractions throughout the water column. The length of the intercept extrapolation emphasizes the importance of careful
sample collection, storage, and high-precision analyses.

## 4.2   Glacial meltwater and precipitation

The $\delta^{18}O$ of local precipitation and glacial melt are significantly different from one another, and can be clearly distinguished on the basis of $\delta^{18}O$. Potter and Paren (1985) observed on George VI glacier that the ice flux into the ice shelf had a $\delta^{18}O$ value of ~-20‰, while accumulation of precipitation onto the
385 northern form of the ice shelf was much less depleted, with a $\delta^{18}O$ of ~-13‰. Precipitation grows increasingly depleted in $^{18}O$ with latitude (Dansgaard, 1964; Gat and Gonfiantini, 1981; Ingraham, 1998; Masson-Delmotte et al., 2008) and altitude (Dansgaard, 1964; Friedman and Smith, 1970; Siegenthaler and Oeschger, 1980; Ingraham, 1998; Araguás-Araguás et al., 2000; Sato and Nakamura, 2005; Masson-Delmotte et al., 2008). 88% of spatial variation in the $\delta^{18}O$ of Antarctic precipitation can
be explained by linear relationships between latitude, elevation, and distance from the coast, with elevation being the primary driver (Masson-Delmotte et al., 2008).  Precipitation collected during the NBP19-01 cruise in the study had a $\delta^{18}O$ value of -15‰, consistent with other data from that latitude and elevation. Precipitation is discussed further in **Appendix A4**.

The nearest Ice cores to our site (ITASE01-2, Steig et al., 2005; Schneider et al., 2006; Siple, Mosley-Thompson et al., 1990) have average $\delta^{18}O$ compositions of -29.8±1.9‰ (ITASE01-2) and -29.6±1.1‰ (Siple). Using locally collected salinity and $\delta^{18}O$ data from deeper than 200 m to calculate a zero-salinity intercept, we identify average freshwater endmembers ranging from -31.3±1.0‰ to -28.4±1.0‰ (mean 29.4±1.0‰). The similar zero-salinity intercept, and strong linear salinity- $\delta^{18}O$ relationship
below 200 m demonstrates that glacial freshwater is responsible for the observed freshening signal. We find roughly half of the total meteoric water inventory in the upper 200 m, below which inventories yield the same relative trend in interannual variability (**Appendix A5**)**,** indicating that the observed variability results from changes in glacial meltwater content, and not from interannual variability in local precipitation. A substantial fraction of precipitation (both local and non-local) will be deposited on
sea-ice, much of which is subsequently advected out of the study area (Assmann et al., 2005), and as a result have no impact on locally measured meteoric water content; suggesting that GMW could comprise an even greater fraction than the base level used here (**Table 2**).

Nearly half of the water column meteoric water content resides in the upper 200 m, and >73% that
meteoric water is comprised of glacial meltwater (**Appendix A5**). The use of a midpoint GMW-

precipitation intercept will overestimate GMW fractions through the water column, directly impacting the accuracy of other techniques using $\delta^{18}$O GMW measurements for calibration (Pan et al., 2023). While previous studies (Randall-Goodwin et al., 2015; Biddle et al., 2019) have not quantified upper water column GMW due to uncertainty surrounding the impact of local precipitation. Our analysis

indicates that precipitation comprises a relatively small fraction of even upper water column meteoric water, even in the most extreme case. The 2-component meteoric endmember approach presented here accounts for a fraction of precipitation in the upper water column, providing more accurate meteoric water and sea ice melt fractions for both deep and shallow waters and more accurate estimates of full column meteoric water (and GMW) inventories.

### 4.3    Temporal changes in mean meteoric water column inventories

We estimated average meteoric water column inventories in the SE Amundsen Sea seawater oxygen isotopes and salinity using a 2-component meteoric water endmember approach in the three-endmember mixing model (**Methods 2.2**). In 1994, 2007, 2014, and 2020, there is a tendency for the maximum

integrated meteoric water volume to extend westward from the SW corner of the PIIS, and along the eastern TIS (**Figure 1**), consistent with the gyre-like circulation there (Thurnherr et al., 2014). This pattern of meteoric water distribution is consistent with local GMW patterns previously observed using traditional hydrographic tracers (Thurnherr et al., 2014; Naveira Garabato et al., 2017; Wåhlin et al., 2021).

Local meteoric water content varies from a low of 8.1±0.7m in 1994 to highs of 9.6±0.8 m in 2000 and 2020. Inventories fluctuated over the latter period, without apparent trend, dependent on the spatial and temporal coverage of available datasets. The inventories presented here are likely to include some fraction of meteoric water (in the form of precipitation and GMW) imported from upstream, however

the consistency of our 0-salinity $\delta^{18}$O intercepts with local ice core values suggests that the meteoric water estimated in this study is predominantly local in source. While salinity and $\delta^{18}$O alone cannot be used to determine basal melt rates, the average meteoric water inventories are sufficient to identify relatively small changes in melt rates, assuming a constant residence time.

The mCDW entering the SE Amundsen Sea and accessing the underside of the ice shelves has been shown to exhibit little seasonal variability, with a maximum variance in T of <0.1°C, salinity of <0.05 g/kg, and thickness of <50 m (Mallett et al., 2018). All samples used in this study were collected from 12 January to 15 March, while melt rates for the PIIS and TIS exhibit very little seasonal variability (Kimura et al., 2017). With a residence time of ~2 years (Tamsitt et al., 2021), it is unlikely that the

variability in yearly meteoric water column inventories is a product of a seasonal signal.

While the meteoric water inventories presented here are not equivalent to melt rates, they should be indicative of relative year-to-year changes in melt rates. Our mean meteoric water inventories suggest an increase in melt rates after 1994 followed by relative stability, with greater year-to-year variability

than the trend in melt rates across the years covered by the study data. We find the lowest mean

meteoric water inventory in 1994, while other studies using hydrographic data estimate 1994 as an average melt year for PIG – lower than 2007 and 2009, but higher than 2014 (Joughin et al., 2021b). Another study identified a slowdown in melt rates between 1992-2017 (Paolo et al., 2023). One modeling study found a low basal melt flux in 1994 followed by a higher, but variable flux through 2013 (where their simulation ends) under ERA5 forcing, while PACE forcing resulted in interannual variability over our study period, but a high melt year occurring in 1994 (Naughten et al., 2022). Adusumilli et al. (2020) found an increase in basal melting between 1994 and 2000 followed by a peak in basal melt in 2009, however their results show relative stability with significant interannual variability between 1994 and 2018. The pattern in mean meteoric water inventories presented in this study is most consistent with recent modeling efforts finding an increase in melting from 1993-2000, followed by relative stability in melt rates through 2019 (Flexas et al., 2022).

Glacial meltwater measured in the SE Amundsen Sea includes mCDW-driven basal melt, iceberg melt (some of which may consist of icebergs imported from upstream), and meltwater entering the ocean at the grounding zone that is driven by the geothermal heat flux to the base of the ice sheet (~5.3 Gt/y; Joughin et al., 2009). Other studies have noted an increase in ice sheet losses to iceberg calving in recent years (Joughin et al., 2021a), comprising a non-trivial component of ice sheet mass loss (Rignot et al., 2013; Greene et al., 2022)– any icebergs that are exported and melt outside of the study area (Mazur et al., 2019, 2021) will not contribute to the mean meteoric water inventories presented here. The greatest limitation for using average meteoric water inventories as a means for GMW accounting arises from the poorly constrained residence time of regional shelf waters, as there has been little study of this component. This uncertainty is further confounded due the intensification of coastal circulation in the Amundsen Sea resulting from increased melt rates (Jourdain et al., 2017). With local circulation generally moving waters westward (Nakayama et al., 2013; Thurnherr et al., 2014; Naveira Garabato et al., 2017; Nakayama et al., 2019; Wåhlin et al., 2021; Dotto et al., 2022) , it is likely that the calculated meteoric water fractions in the study area (with the exception of those on the western side of TIS) are primarily comprised of basal melt from PIIS.

While circulation and residence time are unknown, and increasing melt rates may make them a moving target; the broad assumption that a mean residence time of 2 years (Tamsitt et al., 2021) and GMW comprising >87% of total meteoric water column content (**Appendix A4**) is representative of the whole study area (~30,000 km$^2$ ocean), produces GMW input estimates of between 106±17 Gt/y in 1994 to 129±17 Gt/y in 2000/2020. Though empirical, these figures are consistent with satellite-based estimates of mass loss from PIIS via basal melt (Rignot et al., 2013, 2019), demonstrating the potential utility of geochemical ocean measurements for estimating ice shelf melt rates, and helping to calibrate and constrain larger scale remote sensing estimates.

## 5    Conclusion

We use seawater $\delta^{18}O$ and salinity data collected in the SE Amundsen Sea from 1994 to 2020, to calculate inventories of meteoric (fresh) water through the water column. Freshwater intercepts from

490 salinity-$\delta^{18}$O plots produce a well-constrained meteoric water endmember consistent with measurements from regional ice cores, and indicative of glacial meltwater. Using a meteoric water endmember determined by a regression through salinity-$\delta^{18}$O eliminates much of the uncertainty around meteoric endmembers at depth. In the upper water column, using a meteoric water endmember comprised of glacial meltwater and a volume of local precipitation determined from local climatology,

at an appropriate $\delta^{18}$O, produces more accurate meteoric (and sea ice melt) water fractions both at depth, and in the near surface than approaches taken by earlier studies. With cutting edge advanced optical techniques using $\delta^{18}$O-based meteoric water measurements to calibrate their surface-GMW estimates from satellite data (Pan et al., 2023), it is more important than ever that meteoric water fractions (entirely dependent upon selected endmembers) estimated with $\delta^{18}$O are as accurate as

possible. The application of local precipitation quantities and $\delta^{18}$O values to the meteoric water column inventories presented here has also allowed us to demonstrate that meteoric waters from the ocean surface to the floor are comprised primarily of glacial meltwater; >73% in the upper water column, and nearly 100% in deep waters. While we have been very liberal with the inclusion of a high precipitation fraction in our analysis, a large portion of local precipitation is likely exported by sea ice without ever

entering the surface ocean.

The WAIS is an important region for understanding sea level rise, as changes in winds and ocean circulation can increase basal melting of ice shelves, and the flow of their ice streams into the sea. We present an advancement in endmember determination and application for use in the three-endmember

mixing model over previous studies, producing more robust, well-constrained results, and suggest broader applications for these types of data. Changes in meteoric water inventories in the SE Amundsen Sea study region are consistent with satellite-based estimates of annual mass loss from the PIIS. These results demonstrate the potential utility of seawater $\delta^{18}$O and salinity data as an independent method for estimating ice shelf basal melt rates. While subject to an increased level of uncertainty due to the

enhanced circulation resulting from increased ice shelf melt and potential influx of meltwater from upstream, monitoring meteoric water fractions, and average inventories can provide insight into the fast-changing conditions in the SE Amundsen Sea. Regular sampling for $\delta^{18}$O and salinity in this region could reveal if the existing record and its variability will extend into an era when ice shelves are likely to be thinner, with their grounding lines deeper and farther south. Integration of $\delta^{18}$O data into

numerical models – with $\delta^{18}$O and associated meteoric water inventories used to constrain and calibrate ocean-ice models, and model outputs informing ongoing $\delta^{18}$O sampling strategy could further our understanding of ocean circulation and ice loss along this climatically sensitive sector of the WAIS.

 **APPENDIX**

## A1 Defining mCDW

Modified Circumpolar Deep Water (mCDW) is one of three endmember waters we use in a mixing model to determine glacial meltwater fractions. As the salinity and $\delta^{18}O$ of mCDW are well observationally well constrained, with interannual variability and properties that are defined separately
for each year, as in **Figure A1** with 2020 data. Being the warmest, saltiest water on the continental shelf, mCDW appears at the top-right on a T-S diagram (Panel **a**), where it also identifies waters that are the least-depleted in $\delta^{18}O$.

In Panels **b** and **c**, the same 2020 data show the keystone positions of mCDW in
temperature/salinity/$\delta^{18}O$/depth space. The red and blue dashed lines show property mixing lines between mCDW, glacial meltwater (GMW) and sea ice melt, with the colder waters being fresher and more depleted in $\delta^{18}O$. Most data are above ~800 m, with the least $\delta^{18}O$ depletion in a few deep depressions. Waters that fall off the mCDW-GMW mixing line in the upper 200 m have been influenced by sea ice melting/formation and atmospheric processes. Sea ice melt has a slightly positive
(+2.1‰) $\delta^{18}O$, while GMW has a very negative (~-30‰) $\delta^{18}O$. Both freshen seawater, with the sea ice melt slightly counterbalancing the strong negative $\delta^{18}O$ of GMW.

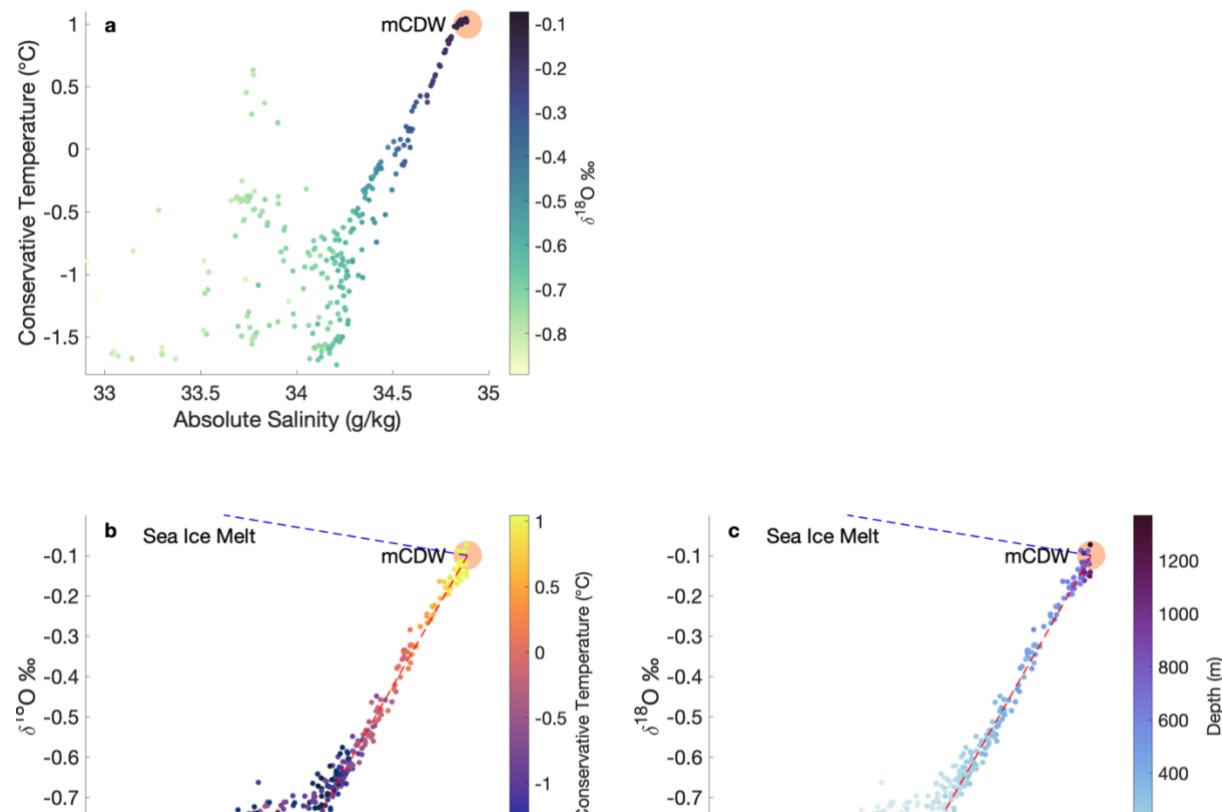

**Figure A1: Temperature, salinity, and δ¹⁸O and depth from 2020 data.** A) T-S diagram with colorbar showing δ¹⁸O. b) δ¹⁸O vs salinity with colorbar showing temperature. C) δ¹⁸O vs salinity with colorbar showing sample depth. Data diverge from the mCDW-glacial melt mixing line at depths shallower than 200 m due to the presence of sea ice melt in the admixture. In Panels b and c, dashed lines show the associated property mixing lines for mCDW mixing with sea ice melt, or GMW.

In **Figure 2** of the main text, δ¹⁸O-salinity plots for each year reveal several data points near the salinity maximum, with some variability in the corresponding δ¹⁸O. Below 200 m, trendlines extrapolated to zero-salinity intercepts define the mCDW and meteoric water (GMW) endmemebers used in the mixing model. mCDW and meteoric water δ¹⁸O are defined at the salinity maximum and zero-salinity intercepts of the trendlines (**Table 2**). The mCDW location corresponds to conventional measures of the deepest and warmest waters on the continental shelf. The calculated zero-salinity intercept values are consistent with the properties of locally available GMW.

## A2 Inter-laboratory offsets

$\delta^{18}O$ data from different laboratories are subject to possible systematic offsets. For example, a ~0.1‰ $\delta^{18}O$ offset between the 2014 data and other years (**Figure A2**) is likely the result of an inter-laboratory calibration offset. On the other hand, greater scatter in the 2009 data suggests that evaporation during sample storage left some samples less depleted in $\delta^{18}O$. Here, we primarily compare calculated meteoric water fractions rather than $\delta^{18}O$ values, with mCDW and meteoric water signatures defined separately for each year so that any offset will not affect the values of samples from that year relative to their mCDW/meteoric water signatures.

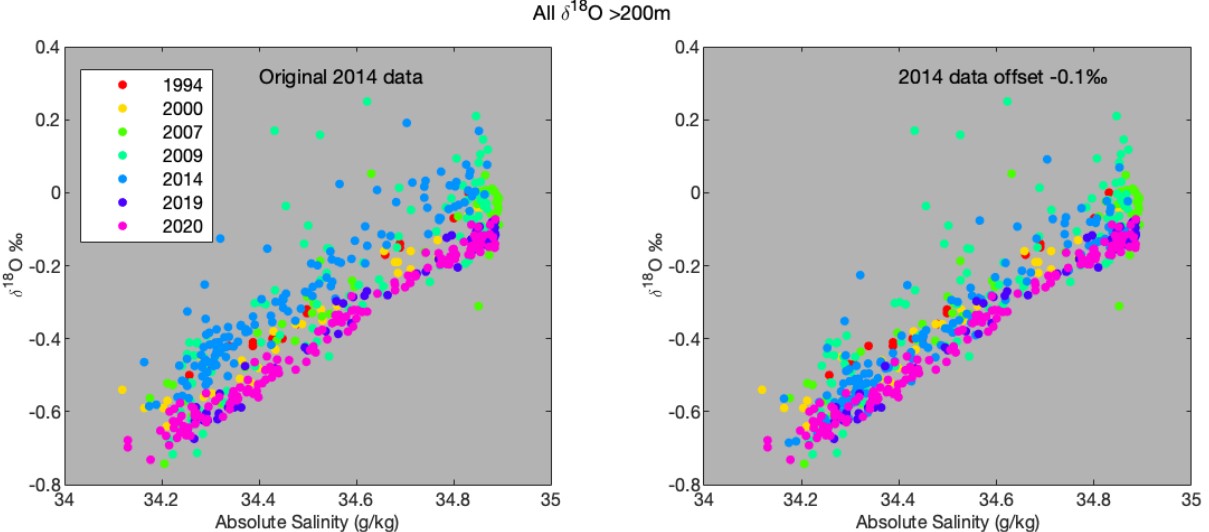

**Figure A2: $\delta^{18}O$ vs Absolute salinity for all years, from data >200 m.** Left panel: All $\delta^{18}O$ vs salinity data, with the 2014 data as published in (Biddle et al., 2019). Right panel: same, with a -0.1‰ offset correction applied to the 2014 data.

A sensitivity analysis, all sample data from a given year were offset, and mCDW/meteoric water signatures re-calculated using the offset data. The result and endmembers were used to calculate meteoric water fractions in the 3-endmember mixing model, with sea ice melt values remaining static. We found that an offset of 5.7‰ $\delta^{18}O$ (**Figure A3**) would be necessary to change the calculated meteoric water fraction by an amount greater than the analytical precision (±0.04‰ $\delta^{18}O$, ±0.003 g/kg for salinity) and environmental uncertainty based on ice core measurements (±1.9‰ for $\delta^{18}O$) and year-to-year variability in mCDW values (±0.06‰ $\delta^{18}O$). Inter-lab offsets should be less than 0.1‰ (Walker et al., 2016), so any offsets will not be significant when comparing calculated meteoric water fractions.

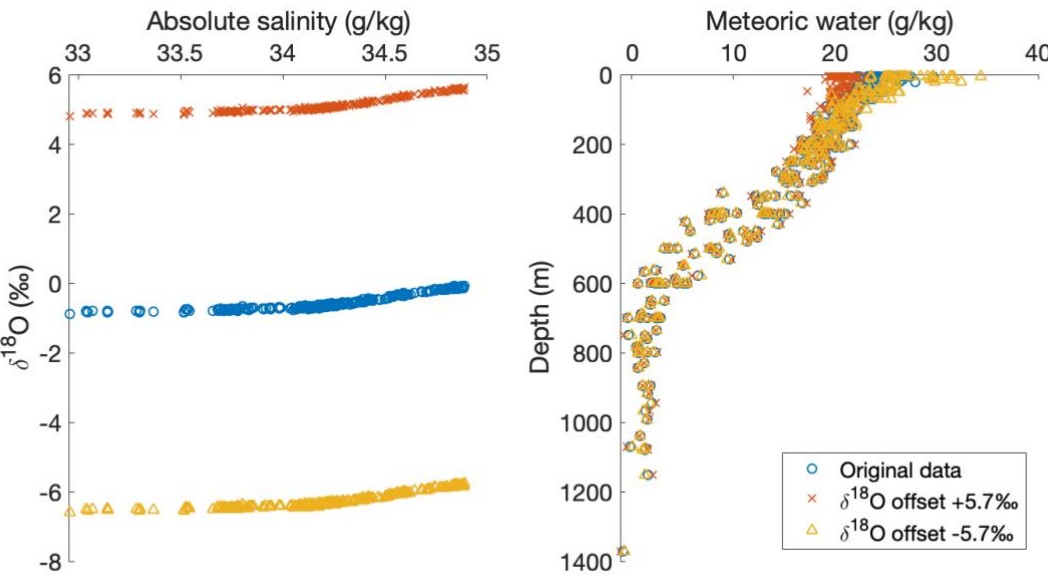

**Figure A3: Impact on inter-lab offsets on calculated meteoric (glacial melt) water fraction for NBP20-02 data.** The left panel shows
the δ¹⁸O offset (±5.7‰) necessary to significantly affect calculated meteoric water fractions, when using mCDW and meteoric water
endmembers calculated from that data. The right panel shows the calculated meteoric water fractions produced using the original, and
offset data. The calculated meteoric water fractions are impacted very little because two of the three endmembers (mCDW and meteoric
water) are defined *by* the offset data.

## A3 Uncertainty and sensitivity analyses

Since meltwater fractions are calculated using analytical measures of salinity and $\delta^{18}$O, the accuracy and precision of these measurements are important. CTD salinity sensors have a reported precision of ±0.002. The Isotope Ratio Mass Spectrometer (IRMS; 1994 to 2014) measurements have a measured precision of ±0.04‰ based on replicates, while the Cavity Ring-Down Mass Spectrometer (CRDS) achieved a precision of ±0.02‰. The meteoric (GMW) endmember is arguably the least-well constrained, with glacial ice in West Antarctica ranging from -20‰ to -40‰, but much of that uncertainty has been eliminated by using the zero-salinity intercept determination on a $\delta^{18}$O-salinity mixing line, corroborated by nearby ice core values as discussed in the main manuscript. mCDW is well-constrained, based on many accurate in-situ measurements. Our sea ice melt endmember is adopted from previously published studies in the region (Meredith et al., 2008, 2010, 2013; Randall-Goodwin et al., 2015; Biddle et al., 2019).

### A3.1 Precision and geographic sampling variability impact on endmembers

mCDW and meteoric water endmembers are determined from sample data. We tested the sensitivity of these endmember values to instrumental precision and geographic sampling variability. We ran a Monte Carlo analysis calculating results from random sets of 3 stations 15,000 times. For each group of 3 stations, mCDW and meteoric water endmembers, and average meteoric water column inventories were calculated using only those data. Stations with fewer than 2 samples >200 m were excluded. As the 3-endmember mixing model is most sensitive to the mCDW endmember, a set of stations lacking samples >800 m (deeper than the sub-cavity ridge; 'pure' mCDW) had a random mCDW sample from >800 m that year added to the data set for analysis.

**Table A1: Sensitivity of mCDW and meteoric water endmembers to instrumental precision and geographic sampling variability.** mCDW is defined as the $\delta^{18}$O value at the salinity maximum on the linear regression of all salinity-$\delta^{18}$O measurements deeper than 200 m in each group of stations; meteoric water $\delta^{18}$O is defined as the zero-salinity intercept on that same line. Average meteoric water inventory is the depth integration of the Gaussian fit of all calculated meteoric water fractions within each group. In all cases, uncertainty is represented by the standard deviation in the results obtained across 15,000 Monte Carlo simulations for each year. 1994 had only 4 stations, variability represented by this analysis for that year may be artificially low

| | Precision | | | Geographic sampling variability | | | Combined | | |
|---|---|---|---|---|---|---|---|---|---|
| **Year** | **mCDW salinity (g/kg)** | **mCDW $\delta^{18}$O (‰)** | **Meteoric water $\delta^{18}$O (‰)** | **mCDW salinity (g/kg)** | **mCDW $\delta^{18}$O (‰)** | **Meteoric water $\delta^{18}$O (‰)** | **mCDW salinity (g/kg)** | **mCDW $\delta^{18}$O (‰)** | **Meteoric water $\delta^{18}$O (‰)** |
| **1994** | 0.002 | 0.018 | 1.62 | 0 | 0.004 | 0.59 | 0.002 | 0.020 | 1.92 |
| **2000** | 0.002 | 0.011 | 0.91 | 0.010 | 0.017 | 1.61 | 0.010 | 0.027 | 2.35 |
| **2007** | 0.001 | 0.009 | 0.80 | 0.007 | 0.034 | 3.17 | 0.007 | 0.039 | 3.80 |
| **2009** | 0.002 | 0.006 | 0.59 | 0.001 | 0.045 | 3.77 | 0.002 | 0.050 | 4.30 |
| **2014** | 0.002 | 0.008 | 0.59 | 0.015 | 0.030 | 2.96 | 0.015 | 0.036 | 3.43 |
| **2019** | 0.001 | 0.004 | 0.39 | 0.005 | 0.013 | 1.61 | 0.006 | 0.015 | 1.85 |
| **2020** | 0.001 | 0.003 | 0.25 | 0.016 | 0.014 | 1.37 | 0.016 | 0.017 | 1.60 |

* As 1994 has only 4 sampling locations, and the strongest fit of any year **(Figure 2)** its uncertainty may be artificially decreased

mCDW properties display low geographic sensitivity, though 2009 and 2014 exhibit higher variability than other years, potentially due to sample collection and/or storage issues (**Appendix A2, A6**). Meteoric water endmember properties showed greater spatial uncertainty in 2007, 2009, 2014. In these 3 years, the data show the greatest scatter, and mCDW samples were not always collected from the deep water column. The lengthy meteoric water endmember extrapolation benefits from many samples
collected below 200 m, all the way to the seafloor. 2014, spatial uncertainty is somewhat inflated due to the very high (>10 m) meteoric water inventories at stations immediately alongside TIS, while the 2009 data are impacted by sample storage issues, and poor depth resolution at some locations. In all cases, instrumental precision had insignificant impact on endmember determination.

### A3.2 Uncertainty in calculated meteoric water fractions

We use Monte Carlo simulations to estimate uncertainty in water mass fraction calculations. We ran 15,000 simulations, selecting 3 stations at random with input values varied randomly within these bounds, and represent uncertainty by the standard deviation of the difference between the simulated, and initial runs. Observations were varied randomly by analytical precision for each run. mCDW and meteoric water endmembers vary through each run, depending on the subset of stations selected (**Table**
**A1**), and the perturbations made to the observational data. Additional perturbations are made to the endmember values, including sea ice melt (**Table A2**).

mCDW salinity varied by the results of a spatial sensitivity analysis (**Appendix A3**) and $\delta^{18}O$ was varied by the 68% prediction interval (1 σ) of the > 200 m $\delta^{18}O$-salinity relationship at the salinity
maximum.

The GMW meteoric water endmember, used for samples >200 m is additionally perturbed by the standard deviation of the ITASE01-2 ice core – also consistent with the largest 0-salinity $\delta^{18}O$ intercept standard error any year (2009). For those samples shallower than 200 m, the precipitation $\delta^{18}O$ was
perturbed by values with a standard deviation of 4.5‰ (the mean standard deviation of precipitation collected at sites in West Antarctica (Global Network of Isotopes in Precipitation (GNIP), 2023), and the GMW fraction was perturbed by 0.05.

The sea ice melt endmember is perturbed based on theoretical and experimental values (Rohling, 2013).

Perturbations used in the uncertainty analysis are summarized in **Table A2**, and the impacts on meteoric water fractions and column inventories are summarized in **Table A3.**

**Table A2: Endmember perturbations for uncertainty analysis.** Perturbations are based on analytical precision for observations, the ITASE01-2 ice core for meteoric water, and theoretical values for sea ice melt (Rohling, 2013). mCDW perturbations are based on the 68% prediction interval (~1 σ) of the >200 m $\delta^{18}$O-salinity relationship at the salinity maximum, and salinity perturbations are based on the results of the spatial randomization analysis **(Appendix A3)**.

| Parameter | Absolute Salinity perturbation (g/kg) | $\delta^{18}$O perturbation (‰) |
|---|---|---|
| Observations | 0.002 | 0.04 (0.02 for CRDS) |
| Meteoric water (GMW) | N/A | 1.9 |
| Meteoric water (precipitation) | N/A | 4.5 |
| Effective <200 m meteoric water | N/A | 2.9** |
| Sea ice melt (Rohling, 2013) | 2 | 0.1 |
| mCDW, 1994 | 0.009* | 0.011 |
| mCDW, 2000 | 0.010 | 0.020 |
| mCDW, 2007 | 0.007 | 0.042 |
| mCDW, 2009 | 0.001 | 0.050 |
| mCDW, 2014 | 0.015 | 0.024 |
| mCDW, 2019 | 0.005 | 0.011 |
| mCDW, 2020 | 0.016 | 0.011 |

* For 1994, we perturbed salinity by the average salinity standard deviation for the other years, to compensate for the smaller number of samples
** Mean effective perturbation across all years

**Table A3: Uncertainty in meteoric water fractions and column inventories resulting from different types of perturbations.** Mean uncertainty in meteoric water fractions and column inventory associated with perturbations to each of the evaluated sources of uncertainty.

| Year | Precision Fraction (g/kg) | Precision Inventory (m) | Endmember Fraction (g/kg) | Endmember Inventory (m) | Geographic Fraction (g/kg) | Geographic Inventory (m) | All Fraction (g/kg) | All Inventory (m) |
|---|---|---|---|---|---|---|---|---|
| 1994 | ~0 | 0.15 | 0.79 | 0.50 | 0.12 | 0.23 | 1.61 | 0.63 |
| 2000 | ~0 | 0.12 | 0.97 | 0.63 | 0.74 | 0.46 | 1.91 | 0.82 |
| 2007 | ~0 | 0.13 | 1.08 | 0.72 | 1.00 | 0.65 | 2.25 | 1.10 |
| 2009 | ~0 | 0.06 | 1.14 | 0.72 | 1.86 | 0.69 | 2.84 | 1.12 |
| 2014 | ~0 | 0.05 | 0.91 | 0.58 | 1.02 | 0.82 | 1.98 | 1.03 |
| 2019 | ~0 | 0.05 | 0.81 | 0.52 | 0.60 | 0.42 | 1.41 | 0.71 |
| 2020 | ~0 | 0.03 | 0.96 | 0.55 | 0.69 | 0.56 | 1.53 | 0.81 |

The mean uncertainty in meteoric water fractions ranges from ±1.4 g/kg in 2019 to 2.8 g/kg in 2009, corresponding to average meteoric water column inventories uncertainty between ±0.5 m in 2019 and ±0.7 m in 2009 (**Table A3**). Meteoric water and sea ice melt fractions vary inversely, while mCDW fractions remain relatively stable. Calculations are most impacted by changes to the mCDW endpoint, as mCDW makes up ~99% of the (800 m) water column on average; >95% in the meteoric water and sea ice melt rich surface waters, and >98% at all depths below 200 m.

14 samples from 2007 (3), 2009 (9), and 2014 (2) suggest negative meteoric water fractions, nine beyond the uncertainty described above. The negative meteoric water fractions result from high-salinity deep waters with $\delta^{18}$O values less negative than the mCDW endpoint in the 3-endmember mixing

model, reflective of uncertainty in the data and/or model limitations. Those years also display a wider spread in mCDW $\delta^{18}O$ than other years, likely due to evaporation during storage.

Sea ice melt and mCDW fractions are discussed in **Appendix A5**.

### A3.3 Geographic grouping analysis

We analyzed the spatial sensitivity of results by splitting the study area into four groups and analyzing data from those groups for each year (**Results 3.5.2, Figure 6, Figure A4**). For each area, mCDW and meteoric water endmembers were defined based on only those data. In 15,000 Monte Carlo simulations,

the observations and endpoints were perturbed by uncertainty associated with analytical precision and environmental variability (**Table A2**).

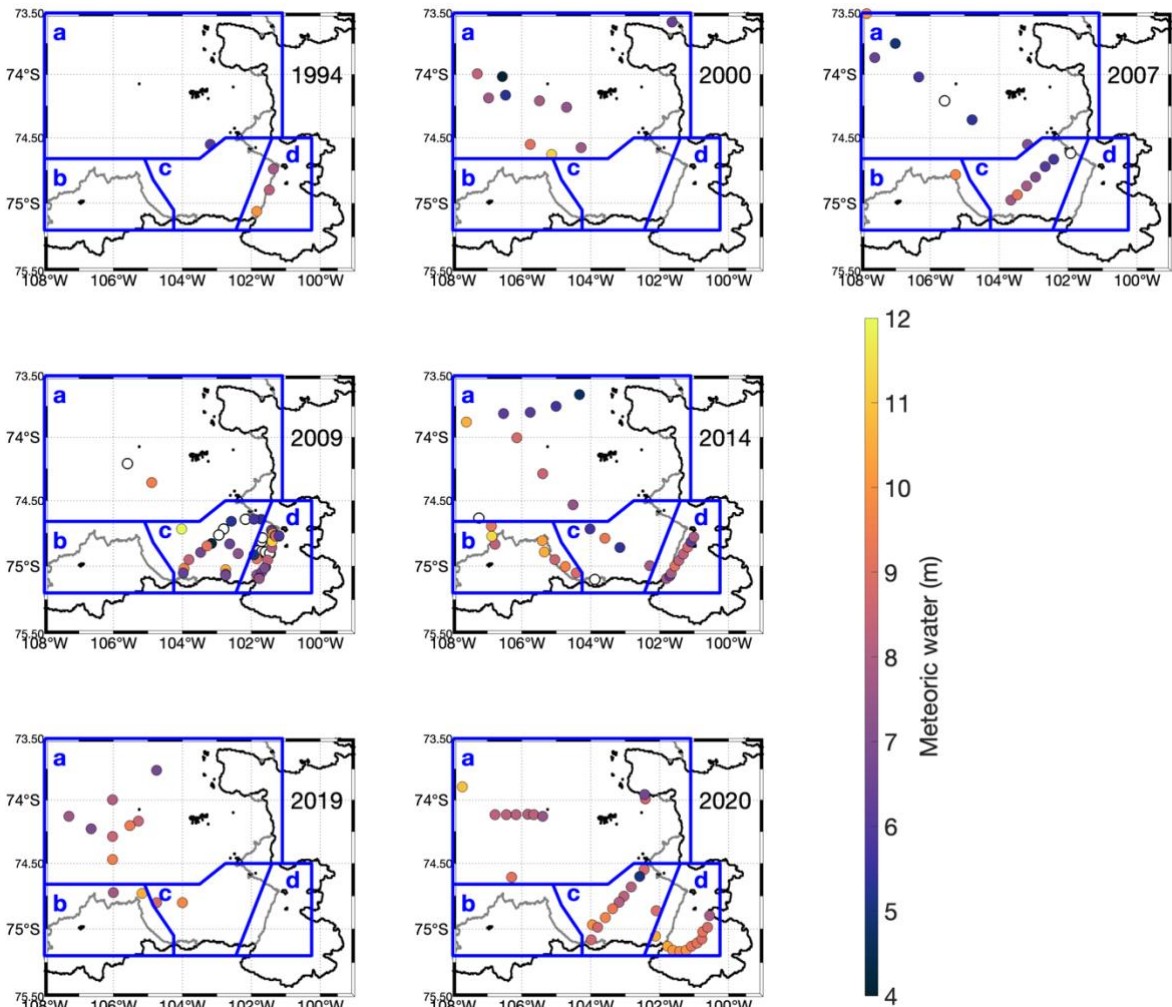

**Figure A4: Sampling locations and geographic group boundaries for all years.** Dot colors show meteoric water column inventory at individual stations, and outlines showing geographic groupings of stations for geographic sensitivity analysis. Some locations provided only partial column inventories. White dots show stations where only one or two depths were collected.

**Table A4: Results of geographic grouping sensitivity analysis.** mCDW is defined as the $\delta^{18}O$ value at the salinity maximum falling on the linear regression of all salinity-$\delta^{18}O$ measurements deeper than 200 m in each group of stations; meteoric water $\delta^{18}O$ is defined as the zero-salinity intercept on that same line. Uncertainty in mCDW $\delta^{18}O$ is represented by the 68% prediction interval (1 σ) at the salinity maximum (~1 σ), and uncertainty in salinity is the result of the randomization spatial sensitivity analysis, plus variation from perturbation of observations by analytical precision. Average meteoric water inventory is the depth integration of the Gaussian fit of all calculated meteoric water fractions within each group, with uncertainty represented as the standard deviation in meteoric water fractions achieved using 10,000 Monte Carlo simulations perturbing the observations and endpoints by associated analytical and environmental uncertainty (**Table A2**). For each group of stations, mCDW and meteoric water endmembers used in meteoric water calculations are defined using only those data.

| Year | Group | # of Stations | # of samples | mCDW absolute salinity (g/kg) | mCDW $\delta^{18}O$ (‰ vs VSMOW) | Meteoric water $\delta^{18}O$ (‰ vs VSMOW) | Mean meteoric water column inventory (m) | Meteoric water fraction uncertainty (g/kg) | Mean sea ice melt water column inventory (m) | Sea ice melt fraction uncertainty (g/kg) |
|---|---|---|---|---|---|---|---|---|---|---|
| 1994 | a | 1 | 8 | 34.83±0.002 | -0.02±0.03 | -31.9±3 | 6.8±0.5 | 1.3 | -0.3±0.7 | 1.7 |
| | d | 3 | 18 | 34.86±0.002 | -0.02±0.02 | -28.3±2 | 8.5±0.6 | 1.6 | 0.7±0.9 | 2.1 |
| 2000 | a | 10 | 62 | 34.88±0.002 | -0.05±0.01 | -28.7±0.9 | 9.6±0.7 | 1.6 | 0.1±0.9 | 2.1 |
| 2007 | a | 8 | 34 | 34.90±0.001 | -0.04±0.01 | -26.1±1.4 | 8.8±0.8 | 1.8 | 0.6±1.0 | 2.3 |
| | b | 1 | 4 | 34.85±0.002 | -0.05±0.04 | -29.7±2.8 | 10.6±0.8 | 1.4 | 1.3±1.1 | 1.9 |
| | c | 6 | 36 | 34.87±0.002 | 0.01±0.02 | -32.1±1.4 | 9.5±0.7 | 1.6 | 0.7±1.0 | 2 |
| 2009 | a | 2 | 4 | 34.87±0.002 | 0.01±0.03 | -43.7±5.2 | 7.5±0.6 | 1.1 | 0±0.7 | 1.4 |
| | c | 18 | 61 | 34.87±0.002 | -0.01±0.01 | -28.8±1.1 | 9.1±0.8 | 1.8 | -1.4±1.1 | 2.3 |
| | d | 26 | 110 | 34.87±0.001 | 0.02±0.01 | -29±0.8 | 8.9±0.7 | 1.7 | -0.7±1.0 | 2.2 |
| 2014 | a | 9 | 57 | 34.86±0.002 | 0.04±0.01 | -27.2±1.2 | 8.7±0.6 | 1.8 | -0.1±1.0 | 2.4 |
| | b | 8 | 61 | 34.87±0.002 | 0.07±0.01 | -32.1±1.0 | 10.7±0.6 | 1.5 | 0±0.9 | 2 |
| | c | 5 | 19 | 34.83±0.002 | 0.06±0.03 | -33.6±2.0 | 8.3±0.5 | 1.4 | -0.5±0.9 | 1.8 |
| | d | 9 | 76 | 34.83±0.002 | 0.06±0.01 | -31.7±1.1 | 7.9±0.5 | 1.4 | -0.2±0.8 | 1.9 |
| 2019 | a | 8 | 68 | 34.89±0.001 | -0.1±0.01 | -27.6±0.6 | 8.7±0.6 | 1.2 | 0.1±0.7 | 1.6 |
| | b | 2 | 21 | 34.87±0.002 | -0.12±0.01 | -30.6±0.9 | 10.4±0.7 | 1.1 | -0.3±0.8 | 1.4 |
| | c | 2 | 18 | 34.85±0.002 | -0.11±0.01 | -31.5±1.0 | 8.6±0.5 | 0.9 | 0.3±0.6 | 1.2 |
| 2020 | a | 10 | 90 | 34.89±0.002 | -0.11±0.01 | -28.2±0.4 | 9.3±0.6 | 1.3 | 0.6±0.8 | 1.9 |
| | c | 11 | 70 | 34.86±0.002 | -0.12±0.01 | -29.6±0.5 | 9.1±0.5 | 1.1 | 0.6±0.8 | 1.6 |
| | d | 11 | 120 | 34.85±0.001 | -0.13±0.01 | -29.7±0.4 | 8.9±0.5 | 1.1 | 0.5±0.8 | 1.5 |

mCDW (as defined in **Appendix A1**) exhibits little geographic sensitivity. In all years, mCDW salinity varied by less than observed seasonal variation (0.01 g/kg; Mallett et al., 2018) and $\delta^{18}O$ exhibited less variation than that associated with instrumental precision.

The meteoric water $\delta^{18}O$ fingerprint calculated for different geographic groupings each year is not geographically sensitive – as would be expected with deep meteoric water (basal meltwater) having a single source. 2009 Group a rendered a significantly different meteoric water endmember, however this number is based on data from just 4 samples (3 from >200 m); given the data limitations and the sample quality issues for 2009, it is unlikely that the -43.7±4.8‰ endmember is representative.

In general, the meteoric water column inventories appear insensitive to geographic groupings. The exceptions are Group a in 1994 and 2009, and Group b in 2007, 2014, 2019. In 1994, Group a contains

only a single station in 1994 (8 samples), and only 4 samples (2 stations) in 2009. Group b consists of those samples collected alongside TIS; locations likely to be dominated by meltwater originating from beneath PIIS (Wåhlin et al., 2021). Surprisingly, given the small number of samples collected near TIS in 2007 and 2019, the meteoric water inventories from Group b stations are consistent.

## A4 The impact of precipitation on meteoric water inventories

Precipitation collected at Halley Bay (75.58°S, 20.56°W, 30 m elevation) has an average composition of -22.0±5.6‰, while that collected at Rothera Point (67.57°S, 68.13°W, 5 m elevation) has an average composition of -13.5±3.4‰, and precipitation collected at Vernadsky (65.08°S, 63.98W, 20 m elevation) has an average composition of -10.2±3.0‰ (Global Network of Isotopes in Precipitation (GNIP), 2023).

Sea-level precipitation collected from the field site during NBP19-01 had $^{18}$O values of ~-15‰, consistent with expected local values from other studies (Gat and Gonfiantini, 1981; Ingraham, 1998; Noone and Simmonds, 2002; Masson-Delmotte et al., 2008). This region of the Amundsen Sea receives ~0.5 m water equivalent of precipitation per year (Donat-Magnin et al., 2021), and mCDW on the shelf has a residence time of ~2 years (Tamsitt et al., 2021). We recalculated water column meteoric water inventories assuming 1 m (2 full years) of local precipitation (-15‰ $\delta^{18}$O) in the upper 200 m of the water column at the time of sampling.

We find that adding 1 m of precipitation to the water column decreases the amount of meteoric water (as defined using the zero-salinity intercepts, **Figure 2**) by an average of 0.55±0.01 m, and decreases sea ice melt by an average of 0.57 ±0.03 m (**Table A5**). These results suggest that even with two years' worth of precipitation present in the water column at the time of sampling, the calculated meteoric water inventory could consist of >87% glacial meltwater.

**Table A5: Impact of precipitation on total meteoric water column inventory.** Mean meteoric water inventory is the integrated mean meteoric water content between the surface and 800 m. The upper water column will include meteoric water from both precipitation, and GMW introduced at depth and mixed upward. The three rightmost columns in the table show the impact on meteoric and sea ice melt water column inventories of recalculating column inventories (meteoric water ~-30‰ $\delta^{18}$O, **Table 2**) assuming 2 years of precipitation (~-15‰ $\delta^{18}$O) in the water column at the time of sampling.

| Year | Mean meteoric water inventory (m) using GMW endmember only | Impact of adding 1 m (~2 years) precipitation (-15‰ $\delta^{18}$O) | | |
| --- | --- | --- | --- | --- |
| | | Change in (glacial) meteoric water (m) | Change in sea ice melt water (m) | Estimated water column GMW fraction | Estimated GMW fraction in upper 200 m water column |
| 1994 | 7.62 | -0.56 | -0.61 | 0.876 | 0.781 |
| 2000 | 9.16 | -0.57 | -0.62 | 0.896 | 0.759 |
| 2007 | 8.95 | -0.56 | -0.55 | 0.893 | 0.756 |
| 2009 | 8.57 | -0.55 | -0.55 | 0.889 | 0.775 |
| 2014 | 8.93 | -0.54 | -0.55 | 0.894 | 0.785 |
| 2019 | 8.42 | -0.54 | -0.57 | 0.887 | 0.773 |
| 2020 | 9.18 | -0.56 | -0.54 | 0.896 | 0.735 |

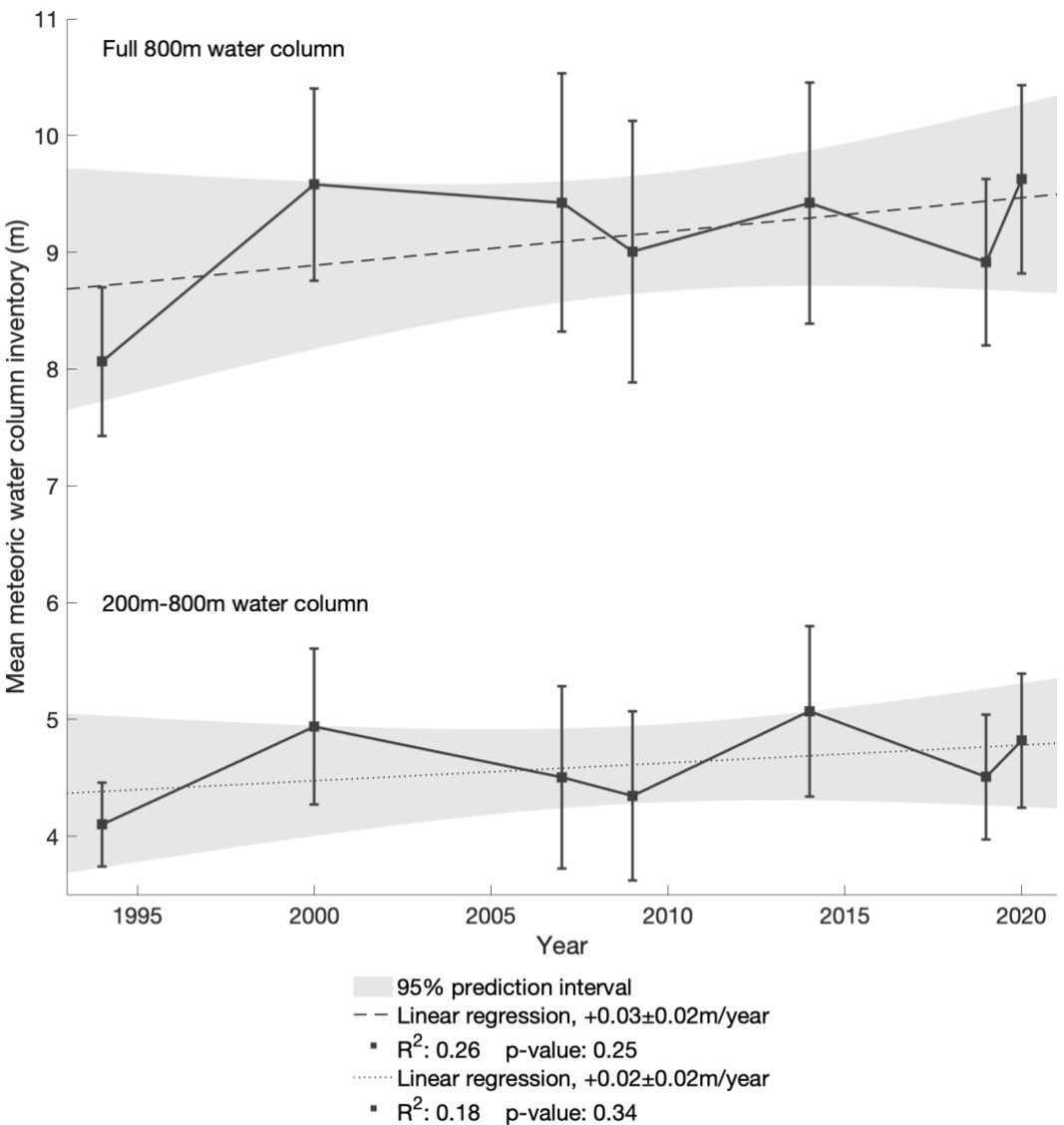

**Figure A5: Mean meteoric column inventory for each sampled year.** Points represent the depth-integrated meltwater volume from the
745 Gaussian Process fit (grey lines in **Figure 3**) between 200 and 800 m depth. Error bars show the uncertainty in mean meteoric water
column inventory associated with analytical precision and environmental variability (**Data and Methods 2.2**). The relative year-to-year
inventories here show the same general empirical trend (within uncertainty) as **Figure 5**. 2014 shows the highest sub-200 m meteoric
water content, owed to the sampling immediately alongside TIS - directly in the pathway of glacial meltwater from PIIS (Wåhlin et al.,
2021).

**Figure A5** and **Table A6** show a comparison of the yearly inventories in the total water column vs the
water column deeper than 200 m. Both the full and partial water columns show the same relative trend
in meteoric water content, indicating that the observed variability is not an effect of interannual
variability in precipitation.

**Table A6: Relative Fractions of yearly meteoric water inventory in the 800 m and 200-800 m water columns.** Reported column inventories are the depth integration of the Gaussian fit of all measurements in the field area between the specified depths. The Relative fraction is the normalized relative volume of the average inventory from year to year.

| Year | 0 m – 800 m | | 200 m – 800 m | | Fraction of total meteoric water in upper 200 m |
|------|-------------|---|---------------|---|---|
| | Column Inventory, meteoric water (m) | Normalized relative fraction | Column Inventory, meteoric water (m) | Normalized relative fraction | |
| **1994** | 8.1±0.6 | 0.84±0.07 | 4.1±0.4 | 0.81±0.07 | 0.492 |
| **2000** | 9.6±0.8 | 1.00±0.09 | 4.9±0.7 | 0.97±0.13 | 0.485 |
| **2007** | 9.4±1.1 | 0.98±0.11 | 4.5±0.8 | 0.89±0.15 | 0.522 |
| **2009** | 9.0±1.1 | 0.94±0.12 | 4.3±0.7 | 0.86±0.14 | 0.518 |
| **2014** | 9.4±1.0 | 0.98±0.11 | 5.1±0.7 | 1.00±0.14 | 0.462 |
| **2019** | 8.9±0.7 | 0.93±0.07 | 4.5±0.5 | 0.89±0.11 | 0.494 |
| **2020** | 9.6±0.8 | 1.00±0.08 | 4.8±0.6 | 0.95±0.11 | 0.499 |

 **A5 Sea ice melt and mCDW fractions**

**A5.1 Sea ice melt**

**Figure A6** shows the net sea ice melt water column inventories in all sample locations. In locations where integrated sea ice melt fractions are negative, net sea ice formation at the time of sampling is indicated.

In 2007, 2009, and 2014 positive ice melt fractions >200 m likely resulted from samples subject to evaporation before analysis. Evaporation leads to positive fractionation of seawater $\delta^{18}O$, leading to a less-depleted $\delta^{18}O$ observation at time of analysis; less depleted $\delta^{18}O$ relative to salinities fresher than mCDW will be interpreted by the 3-endmember mixing model as sea ice melt. The stratification in this

region makes it unlikely that there are significant sea ice melt fractions below 200 m. As with the $\delta^{18}O$-salinity (**Figure 2**) and meteoric water-depth (**Figure 3**) plots, 1994, 2000, 2019 and 2020 exhibit the tightest distribution, suggesting higher quality data. Grey lines show the Gaussian process fit, and points are shaded to show sample $\delta^{18}O$.

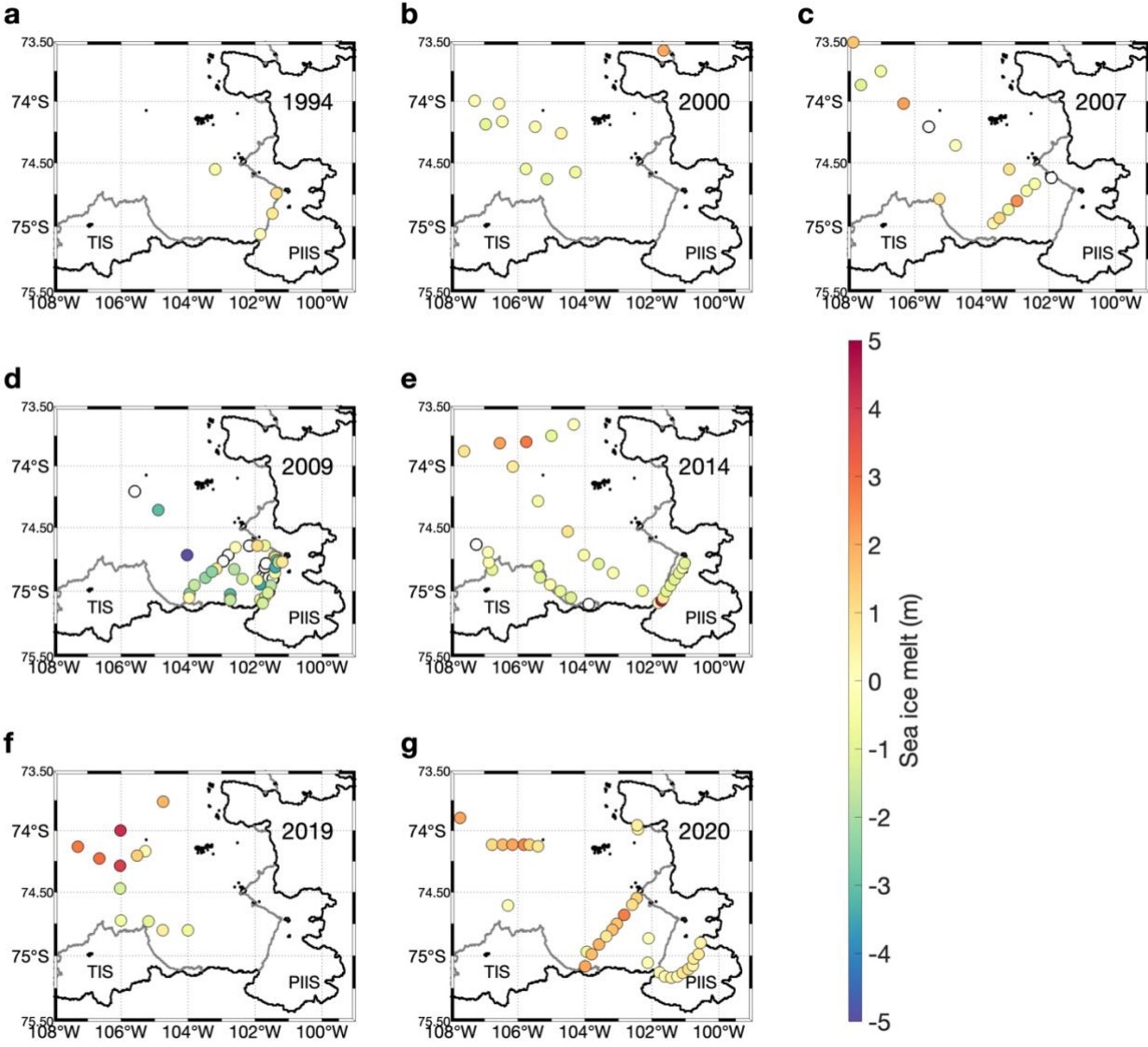

**Figure A6: Integrated sea ice melt fractions at sampling locations each year.** Negative sea ice melt fractions indicate areas of net sea ice formation. Stations with partial water column sampling show only partial inventories. White dots with black outlines are stations where only one or two depths were sampled (2007, 2009, 2014). Years with greater sea ice melt (2007, 2020) than formation show greater divergence from the mCDW-GMW mixing line in surface waters (**Figure 2**).

**A5.2 mCDW fractions**

Waters deeper than ~800 m are comprised of pure mCDW; moving toward the surface, meteoric freshwater from basal melt is introduced starting at ~700 m. The near surface waters are rich in meteoric water and/or sea ice melt are comprised of >92% mCDW (**Figure A7**).

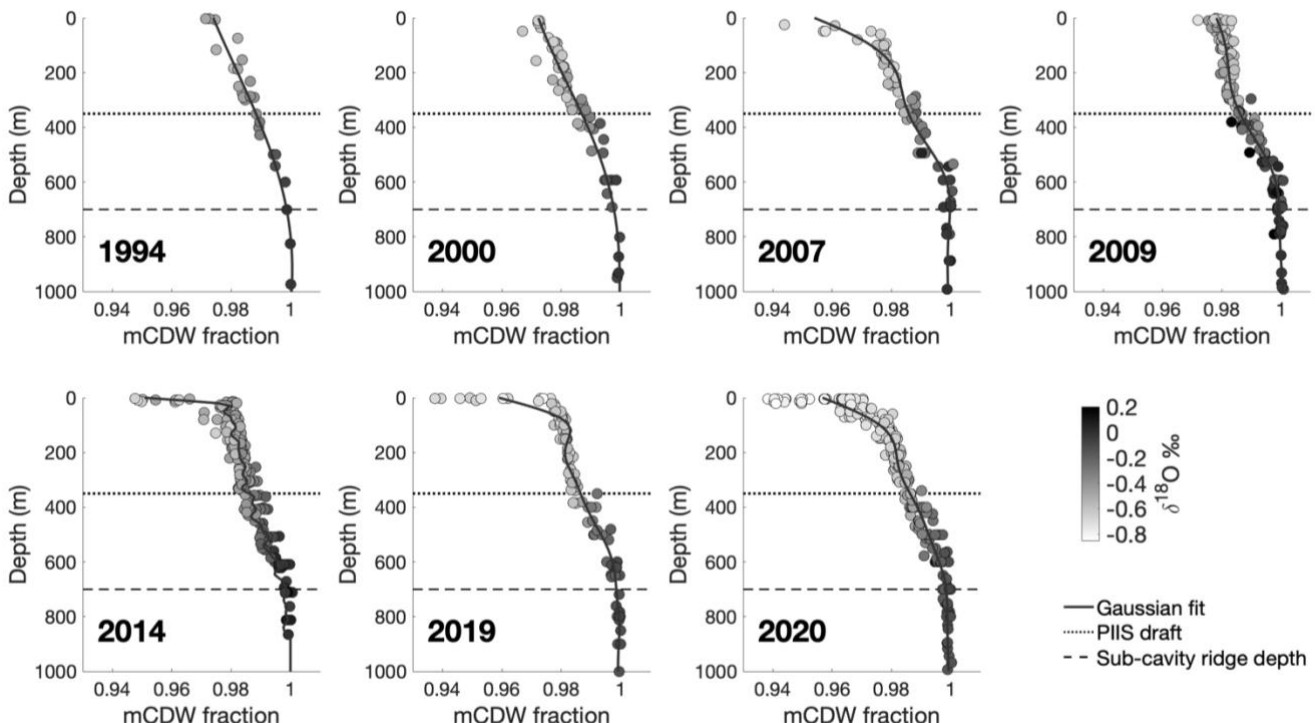

**Figure A7: mCDW fractions vs depth.** Calculated mCDW fractions using salinity and $\delta^{18}O$ measurements in 3-endmember mixing model. Shading of dots indicates the measured $\delta^{18}O$ of that sample. Deep waters (>800 m) characterize relatively unadulterated mCDW, while near surface waters contain the highest concentrations of sea ice melt and meteoric water. Dotted horizontal lines show the depth of the PIIS draft, and dashed lines show the depth of the PIIS sub-cavity ridge.

**Table A7: mCDW column inventories and uncertainty.** Mean sea ice melt column inventories are produced by depth integrating the Gaussian process fit (grey lines **Figure A7**) between the surface and 800 m. Uncertainty as described in **Appendix A3.**

| Year | Mean mCDW column inventory (m) | mCDW fraction uncertainty (g/kg) |
|------|--------------------------------|----------------------------------|
| **1994** | 781.3±0.5 | 0.6 |
| **2000** | 780.2±0.7 | 0.7 |
| **2007** | 779.9±1.2 | 0.8 |
| **2009** | 781.7±0.6 | 0.9 |
| **2014** | 780.5±0.9 | 0.9 |
| **2019** | 780.7±0.6 | 0.6 |
| **2020** | 779.7±0.8 | 0.9 |

## A6 2009 Sample quality control

A subset of the samples for 2009 were analyzed on an IRMS in 2010, while the remainder were stored until a 2020 CRDS analysis. At the latter time, 56% of the samples analyzed contained an unknown, clear, needle-shaped precipitate. Several bottles also had a lower-than-expected sample volume, suggesting evaporation, which would likely have altered the $\delta^{18}O$ content via isotopic fractionation. Several steps were taken to ensure the quality of samples analyzed after a decade in storage.

## A6.1 SEM EDS Analysis of Precipitate

Samples of the precipitate were extracted from multiple sample bottles and analyzed using a Scanning Electron Microscope, equipped with a FEI Magellan 400 XHR SEM with Bruker Quantax XFlash 6 | 60 SDD EDS detector, at the Stanford Nano Shared Facilities (SNSF). Peaks were observed at the spectra associated with Mg, Si, and O, indicating the precipitate is likely some form of Magnesium Silicate Hydroxide ($Mg_3Si_2O_5(OH)_4$), or Magnesium Silicate Hydrate ($Mg_2Si_3O_8 \cdot H_2O$). $Si(OH)_4$ is the simplest soluble form of silica and is found universally in seawater at low concentrations (Belton et al., 2012). The maximum amount of silicate that could be expected in this area of the ocean is ~100 µmol/kg (Rubin et al., 1998). In this case, even if the entire 100 µmol/kg of Si were drawn down to 0, solely into a Magnesium Silicate, with a very high fractionation factor, e.g. the -40‰ reported for diatoms (Leclerc and Labeyrie, 1987) the greatest effect on a sample would be 0.0003‰ – well below the analytical precision of the CRDS (0.025‰) or IRMS (0.04‰). Therefore, it is highly unlikely that the precipitate contributed a detectable fractionation or alteration of seawater $\delta^{18}O$ in our samples.

## A6.2 Quality control for evaporation

In all years, the bulk of the $\delta^{18}O$ data fall in a broadly predictable pattern, less depleted at depths below ~600 m, and more depleted near the surface. Values >2 standard deviations from this pattern denote samples that were likely subject to significant evaporation (**Figure A8**)

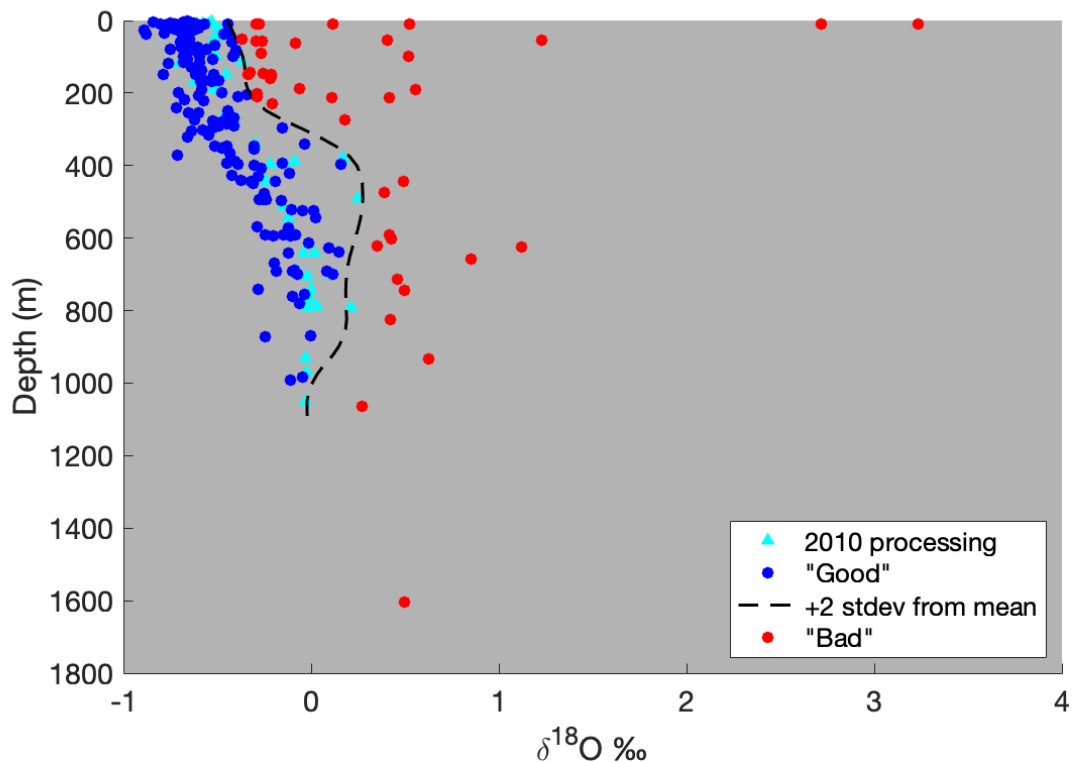

**Figure A8: δ¹⁸O vs depth for all 2009 samples, coded for likelihood of evaporation.** The dashed line represents +2 standard deviations from a moving depth-averaged δ¹⁸O based on the 2010 processing, beyond which the results are unacceptable. Archivable data will be made available upon publication.

As a secondary check, the δ¹⁸O of all samples was plotted vs depth with a qualitative indicator of the amount of precipitate found in the sample vial (**Figure A9**) to see if any patterns emerged compared to that of depth-comparable samples processed in 2010. No clear trend was evident.

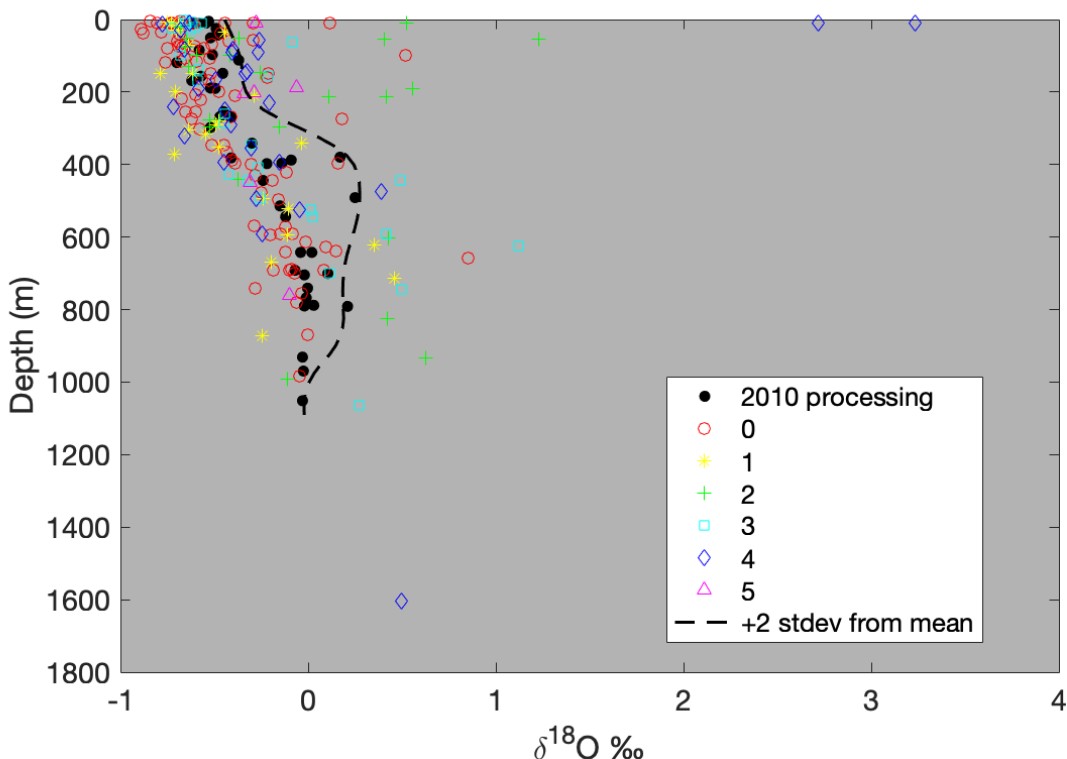

**Figure A9: δ¹⁸O vs depth of all 2009 samples, coded by amount of precipitate present.** Each bottle was graded by eye based on the volume of precipitate present with 0 being no precipitate present, and 5 being the most precipitate present. As with **Figure A8**, the dashed line represents +2 standard deviations from the mean δ¹⁸O at each depth.

Evaporation is accompanied by isotopic fractionation, with $H_2^{16}O$ evaporating preferentially, leaving the remaining liquid relatively enriched in the $H_2^{18}O$. Evaporation also increases the salinity and thus density of the remaining sample. We measured the density of each seawater sample 5 times using a calibrated 1 ml pipet and mg balance. The theoretical density of each sample was calculated from its associated CTD salinity and temperature. Differences between measured and theoretical densities for each sample are plotted in **Figure A10**.

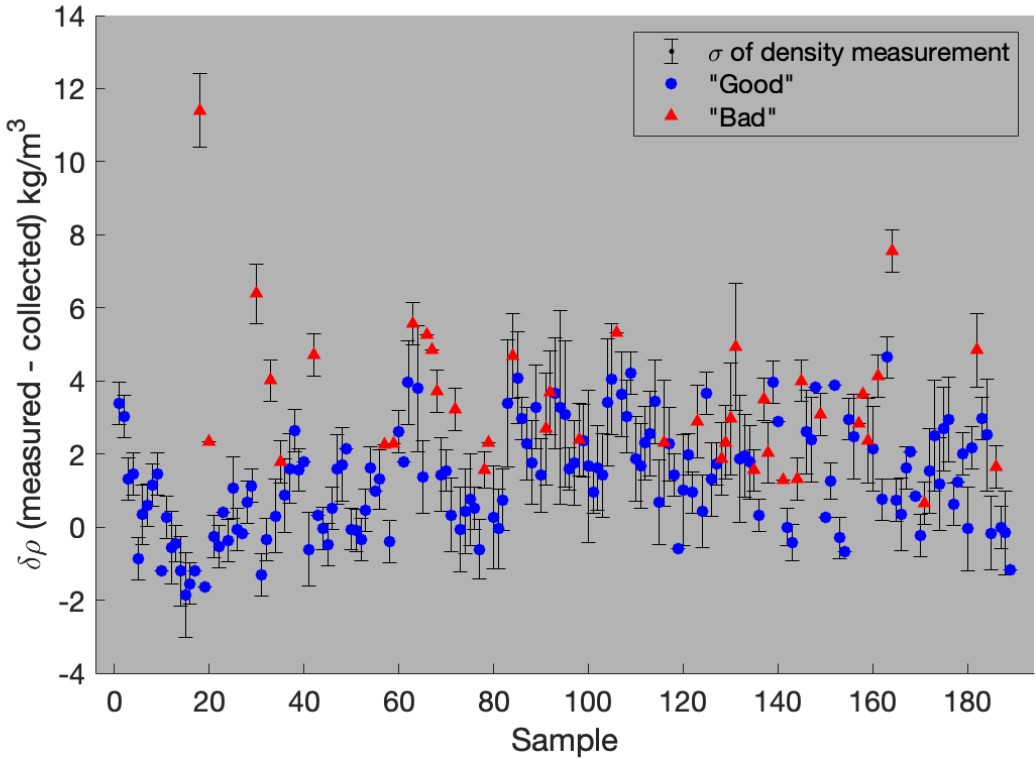

**Figure A10: Difference in measured density vs theoretical density for each 2009 sample analyzed in 2020.** Theoretical density is based on CTD salinity at each sample location, and measured density calculated from 1ml sample aliquots weighed on a mg scale, with sample coding as in **Figure A8**. Error bars represent the standard deviation of replicates for each measurement.

While a few samples show clear evidence of evaporation, and correspondingly high $\delta^{18}O$ values, most
show less obvious density anomalies, exposing the limitations of our scale accuracy at that level. 75% of the samples measured showed a higher than expected density and 25% measured a lower than expected. **Figure A10** displays a significant overlap in measured density space between samples previously identified as "good" or "bad" (**Figure A8**). **Figure A10** shows that there are no samples flagged as compromised ("bad") from our earlier depth-based analysis with a $\delta\rho$ greater than 1.3 kg/m³.
At an aggressive first pass, we removed all sample data with a $\delta\rho$ greater than 1.3 kg/m³ and looked at each hydrocast profile individually, using the remaining data. Excluded samples flagged as "good" were returned to the dataset, and individual profiles re-scrutinized individual profiles to check for any qualitative anomalies.

### A6.3 Conclusion and final 2009 sample inclusion

While it is very unlikely that the precipitate changed sample values, some samples do appear to have been subject to evaporation. The inclusion of all samples flagged as "Good" does not qualitatively

change our analyses when compared with the data processed in 2010. We exclude the 41 samples initially flagged as "Bad," (**Figure A8**) and retain the remaining 148 flagged as "Good".

## A7 CRDS and IRMS cross-calibration

We processed 100 samples from 2019 and 2020 concurrently using the Picarro L2140-i CRDS, and on a Finnigan MAT252 IRMS (**Figure A11**) using $CO_2$ equilibration (Epstein and Mayeda, 1953). Both instruments were independently calibrated using international standards VSMOW, SLAP, and GISP, and all samples were run in duplicate. The data from both machines was comparable, with the Picarro achieving a precision of 0.02‰, and the IRMS achieving a precision of 0.03‰ on replicates. The offset between CRDS and IRMS data averaged -0.02‰, with the CRDS data being more negative. Since the offset between the two machines was less than either instrument's precision, data from the CRDS was used as-is. Values reported for the CRDS are the average of 6 individual injections/measurements from each vial; reported precision is based on the standard deviation between multiple 6-injection averages from replicate analyses, separated by days, weeks, or months.

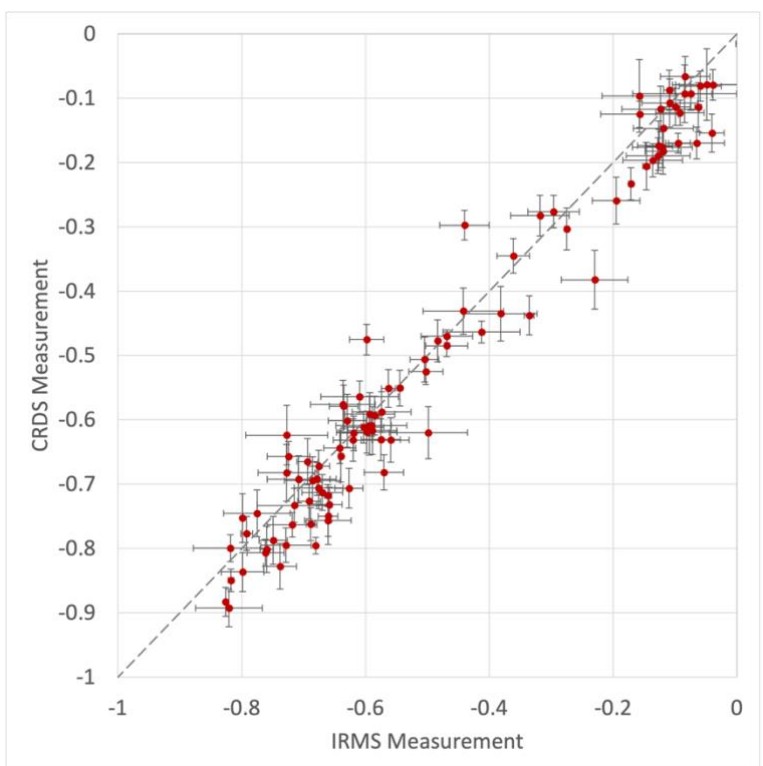

**Figure A11: 101 seawater $\delta^{18}O$ samples collected in 2019 and 2020 analyzed with both CRDS and IRMS.** Each point represents the value obtained by measuring the same sample on the IRMS (x-axis) and the CRDS (y-axis). Error bars represent the corresponding standard deviation of the IRMS and CRDS measurements. The dashed grey line is a 1:1 slope.

## A8 CRDS Methods

570 of the isotope samples for this study (all samples from 2019 and 2020, portions from 2007 and 2009) were run on a Picarro L2140-i CRDS system, rather than a traditional IRMS. Using this system, we were able to achieve an average precision of <0.02‰ on multiple replicate analyses.

Samples were collected in 10 ml or 30 ml glass serum vials (Fisher Scientific part number: 06-406D/06-406F), sealed with rubber stoppers (Fisher Scientific part number: 06-406-11B) and aluminum seals (Fisher Scientific part number: 06-406-15).

Sample vials should be filled to just below the "neck" (narrowest part). Minimizing the headspace of the vial is important for minimizing evaporation, however it is important to leave *some* headspace to allow for expansion/contraction of the sample (if collecting samples larger than 30 ml, slightly more headspace should be left, with 250 ml vials being filled to ~1/2 way up the "shoulder" before the neck). When sealing serum vials with rubber stoppers and aluminum seals, it is important that the tops of the aluminum seals are crimped tightly but remain flat after crimping/capping. An upward "buckling" of the aluminum seal indicates over-crimping and will produce an inferior seal.

Internal lab (Stanford SIL) data shows that samples can be preserved in this manner for multiple years without significant degradation, while bottles with threaded caps and parafilm reliably prevent sample evaporation up to the level of instrument analytical precision for no more than 1 year.

The instrument setup used was as follows:

- 10µl syringe (Trajan Part number: 002982)
- A single standard or unknown run consists of 7 injections (measurements) per sample
- Sample injection volume 2.2 µl
- 3x 5µl rinses with fresh water from inkwell (IW) between each sample
- 3x 2.2µl rinses with sample before first measurement of new sample or standard vial
- Rinse only between sample vials, or 1 rinse for every 7 standard or unknown injections.

Several protocols were also followed with regards to sample and instrument handling.

- Fresh vial of internal lab standard (ILS) used each day. The ILS was prepared to have a $\delta^{18}O$ in the middle of the range expected from the unknowns (i.e. ~-0.3‰) to minimize memory issues between samples. The IW was also filled with water of approximately this composition.
- A fresh 2ml vial of ILS was used each day and discarded at the end of the sequence.
- Samples were pipetted from sealed 10 ml or 30 ml serum into 2 ml vials (Fisher Scientific part number: 03-391-15) for analysis on the day they were to be analyzed. The 2ml vials used for analysis were found to only reliably preserve sample $\delta^{18}O$ for <1 week.

- After each sequence, the syringe was cleaned with DI water, and then rinsed thoroughly with water from the Inkwell, to minimize memory/contamination issues of residual water left in the syringe.

    o Treated in this way, syringes can be expected to last for 1500 to 2500 injections

- Fresh vaporizer septa (Trajan Part number: 0418240) was used every day

- ILS were analyzed no less than every 5th unknown, and no fewer than 3 ILS were measured per run

- All data were corrected based on the slope of the ILS measurements over the course of the sequence.

- Each run began with no fewer than 10 injections from the IW, to allow the instrument to reach baseline.

- Syringe cleaned thoroughly with DI water each day, and manually rinsed with IW water prior to sequence.

- No more than 5 unknowns (7 injections each) measured between run of ILS (7 injections)

- ILS measured at least 3 times during each sequence – at the beginning, end, and midpoint. least 3 standards measured during each sequence.

A typical 24h sequence ran 16 unknowns. The sequence was set up, as follows:

- 15 injections from IW

- ILS (7 injections)

- 4 unknowns (4x7 injections)

- ILS (7 injections)

- 4 unknowns (4x7 injections)

- ILS (7 injections)

- 4 unknowns (4x7 injections)

- ILS (7 injections)

- 4 unknowns (4x7 injections)

- ILS (7 injections)

Overall, this sequence consists of 162 injections, 112 of which contained salt, for a vaporizer load of ~8.6 mg of salt/day. The instrument vaporizer was cleaned at least every 200 mg worth of salt injected.

- @ 35PSU & 2.2µl injections, this is 2597 salty injections, or 371 samples @ 7 injections each. (~ every 23 analytical days)

Finally, analytical data quality control was conducted in the following way

- The first injection of each sample was discarded, to minimize instrument memory issues

- If the standard deviation of the remaining 6 injections was >0.04‰, up to one outlier could be removed. Any samples where the standard deviation of measured values was still >0.04‰ were rerun the following day from the same vial, using the same septa.

- If a rerun would not be possible the following day, the vial septa was replaced with a new one.

- Data from each hydrocast were inspected as a group. Any samples that appeared inconsistent with the rest of the hydrocast (e.g. with regards to salinity, or neighboring $\delta^{18}O$ values) were rerun. If the rerun occurred within 1 week of the initial run, the same vial was used. Otherwise, a fresh aliquot of sample was drawn from the resealed serum vial.

## Data availability

All data, excluding that flagged as "bad," (**Appendix A6**) used in this study can be accessed at:
https://doi.org/10.25740/zf704jg7109

## Author contributions:

Conceptualization: RBD
Methodology: RBD, ANH, DAM
Investigation: ANH, DAM, RBD
Visualization: ANH
Funding acquisition: RBD
Project administration: RBD
Supervision: RBD
Writing – original draft: ANH
Writing – review & editing: ANH, SSJ, RBD, DAM, RAM
Contribution of data: ANH, DAM, RBD, SSJ, RAM

## Competing interests

The authors declare that they have no conflict of interest.

## Acknowledgments

We acknowledge the staff and crew of Nathaniel B. Palmer cruises 94-02, 00-01, 07-02, 09-01, 19-01, 20-02, and iSTAR2014 for assistance with data acquisition. We extend special thanks to Isa Rosso, Michael Burnett, and Emilia Fercovic for their help with sample collection. MAC3 Impact Philanthropies assisted with CRDS instrumentation and development. Peter Schlosser & Ronny Friedrich provided the 2007 CRDS data, and Shigeru Aoki was consulted regarding the 2014 data offset. Thanks also to Cindy Ross and Stanford Nano Shared Facilities, supported by NSF ECCS-2026822 for SEM analyses.

**Funding:** This project was funded by NSF:
NSF-OPP-1644118
NSF-OPP-1644159

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
