# Peer review of "Meteoric water and glacial melt in the Southeast Amundsen Sea: A timeseries from 1994-2020"

_EGUsphere, 2023_

## Referee Comment (RC1)

General comments:

The authors have done an important job in assembling oxygen-18 observations in this study to investigate meltwater inventories in the Admundsen Sea sector of WAIS. It is of great interest for the community to address questions related to freshwater flux evolution especially in this region. However, I was left wondering what the key message is here and what the results are.

I am a bit confused about the main results and it seems the authors are as well. The flow and progression of ideas is missing with a lot of unsupported speculation rather than solid results and I wonder what the conclusion is here. While some paragraphs are largely true because they are based on a literature review, I am not sure why they are sometimes presented suddenly and how the results of this study influence them.

The glacial meltwater (GMW) term is used in different places, while in other places it is referred to as meteoric freshwater; the terminology is inconsistent and may reflect the suspect methodology used here; it is not possible to separate glacial meltwater from precipitation directly from the combined salinity and 18O observations.

While the data cover 26 years, there are in fact seven summers of observations that are not evenly spatially distributed across the region of interest with high spatial variability and a very low amount of data in 1994. While the authors claim a modest increase in meltwater, this is at odd with the insignificant change and interannual variability mentioned in different places in the manuscript that are ultimately consistent with downstream freshening in the Ross Sea. We are puzzled by these contradictory remarks on the evolution of meltwater content. If there is no change, it is difficult to see how this could influence downstream freshening.

The lack of strong signal is indeed surprising, as one would have assumed that increased melt from ice shelves in the region would have significantly influenced the meltwater content in the water column. A possible solution would be to adjust the focus of the study and concentrate on why there is such constant meltwater fraction which to me at least is an unexpected and interesting finding to investigate.

I apologize for the negative comments, I really think the paper needs more work to make the chain of reasoning clearer and the main results compelling. I have raised a few points below that I hope will be helpful for the authors.

Specific points:

Line 38 – I think the authors can add references here, many studies have included 18O to the temperature-salinity combination to define the characteristics of SO water masses

Line 38-39 – The sentence is a bit confusing; *zero-salinity* is probably too much and I would mention meteoric water rather than glacial freshwater; 18O is useful for differentiating freshwater signals coming from meteoric (precipitation and continental ice) or oceanic (sea ice) sources. References are also needed here

Line 39-40 – I do not understand this sentence. Are the authors saying that only in regions where basal melting is deep (and deep relative to what?), glacial meltwater is more depleted in oxygen-18 than local precipitation? Or is the content of glacial meltwater more important than

the content of local precipitation in these regions? I am not convinced in either case. Are there any references to support these claims?

Figure 1 – The wide map of Antarctica is not very useful here nor is panel b showing only bathymetry. I am not sure how relevant they are to the results. Jet colormap is not perceptually inconsistent and a poor choice for data visualization as it can mask significant changes.

Line 84 – Actually no, meteoric freshwater can be continental ice or precipitation, as they cannot be separated on the basis of salinity and 18O.

Line 85 –

Line 84-93 – I am not sure about the analyses here. Clearly, the 18O endmember does not vary on interannually, the differences among -28.4‰ and -30.2‰ certainly reflect uncertainty in the data. I do not see the need to separate years here to determine endmembers since the isotopic composition of the continental ice is not expected change suddenly from one year to another. Furthermore, as the authors point out and by Masson-Delmotte et al., 2008 show, there is a large variability in 18O of meteoric water on a local scale, so estimating the endmember separately every year is a key result here.

Figure 2 – There is no discussion of the scatter toward lower salinity and constant 18O from the mixing line above 200 m depth and even in the deep waters in 2009 and 2014. It would be interesting to explore and discuss sea ice imprint if possible

Line 104 – sea ice melt and/or sea ice formation. The mixing model can give a negative estimate for sea ice endmember reflecting net sea ice formation.

Line 110 – sea ice and meteoric water are not water masses. I would rather write water sources.

Line 125 – influenced by sea ice melt and formation. But also, non-local precipitation, mixing, advection…

Line 128 – I am not sure how this approach differs from studies that use an average 18O for meteoric water. Biddle et al., 2019 and Meredith et al., 2010 do not use approximate 18O values for glacier but a plausible average of meteoric water because again 18O does not disentangle continental ice and precipitation. Using the zero-salinity intercept on 18O-salinity plots, the authors use the same method; they deduce an average 18O of meteoric water in region where this component is highly variable (Masson-Delmotte et al., 2008).

Line 149 – The data are collected during summers so it is unlikely that GMW endmember values are based on the average of the annual data

Line 160-162 – I do not see the connection between the results discussed in figure 3 and the mCDW heat extent

Line 145-162 – I am not convinced that the authors are properly discussing the results here, but rather a speculative explanation of the origin of glacial melt

Line 167 – The authors point to a modest increase (what is a modest increase? relative to what?) of the mean GMW inventories and at the same time acknowledge that the low average in 1994

may be responsible for this 'trend'. The low number of samples in 1994 should be reflected in the estimate, and if this year is not taking into account, no claim of an increase can be made. Also, how is the linear trend calculated? Are there any uncertainties associated with this calculation?

Line 168-170 – I do not see the link between these two sentences and the discussion of results here, what is the connection between interannual GMW inventories in summer and seasonal variability of mCDW in the region? It is hard to understand; the authors claim a modest increase of GMW inventories and then refer to invariable overall melt rates during the austral summer when the samples used in this study were collected (do they mention a particular year?)

Figure 4 – It is nice to see the series here, but I doubt the authors are showing a volume as mentioned in the caption. Also, integrating from the surface will include precipitation, even if it is negligible. Therefore, I do not really agree with the following assumption; depth-integrated GMW between surface and 800 m depth. How is the linear regression calculated? What is the uncertainty? If the GMW inventory is time invariant as the authors claim, beside a modest increase (a choice has to be done here), I think the linear regression in the figure does not provide crucial information.

Line 177 – This section belongs to Methods rather than Results

Line 178-180 – I assume this analysis corresponds to Appendix A4?

Line 181 – Then why use a single salinity and 18O value for mCDW each year if the authors claim that this water mass is relatively stable over time?

Line 184 – Do the authors really compare GMW content and 18O-salinity relationships? I do not see any comparisons of 18O-salinity relationships in Appendix A4

Line 186 – I am not convinced to the authors' decision to simply remove the 2014 data near to TGT from the analysis (which are still shown in figure 1 in glacial meltwater inventory panel g and I assume in figure 2 as well? Which is confusing) to improve interannual comparability. I would keep all available data from the region if the aim is to make comparisons on a regional scale. Excluding data because the GMW values are simply higher seems problematic and not a good reason to me.

Line 196 – But the authors stated earlier that the properties of mCDW endmember are invariant

Line 199 – Unable to estimate glacial meltwater using salinity and 18O

Line 201 – *and compare GMW fractions rather than d18O values.* Not helpful, it did not need to be said

Line 201 –

Line 203 – Are there any uncertainties in the meltwater content values? I do not think *low* and *high* are useful here

Line 215 – Hard to tell if 2000 was a local high compared with subsequent years due to uncertainties. And what about the year 2020?

Line 217 – Again, it is not clear that there was an increase in average GMW after 1994

Line 219 – How is a steady GMW inventory consistent with a linear, long-term freshening trend? This assumption, which I think is confusing, does not add much here because the study does not examine the contribution of freshwater input on the reported downstream freshening in the Ross Sea

Line 222 – meteoric water inventories from the surface to 800 m. Integrating from the surface will include precipitation

Line 223 – volume or content?

Line 224 – *gyre-like circulation;* adding regional circulation to the map would be helpful and how it would influence meltwater advection

Line 243 – If it is statistically insignificant, there is no linear increase, I am not sure it is useful here

Line 244 – This is tricky, because of the error bars, the lowest and highest melt periods claimed here seem difficult to believe

Line 250 – 18O observations do not allow estimation of basal melt rates

Line 251 – *interannual fluctuations potentially masking an increase over 2.6 decades;* this is very speculative

Line 255 – the last sentence is very confusing; how can meltwater volume rates be measured with 18O observations? How is the invariant GMW inventory mentioned in the paper consistent with any downstream freshening?

---

## Author Comment (AC1)

**Glacial meltwater in the Southeast Amundsen Sea: A timeseries from 1994-2020**

Andrew N. Hennig, David A. Mucciarone, Stanley S. Jacobs, Richard A. Mortlock, Robert B. Dunbar

**Response to Referee #1 Comments**

*Thank you for your comments – they've been helpful in improving the clarity and presentation of this paper.*

**General comments:**

The authors have done an important job in assembling oxygen-18 observations in this study to investigate meltwater inventories in the Amundsen Sea sector of WAIS. It is of great interest for the community to address questions related to freshwater flux evolution especially in this region. However, I was left wondering what the key message is here and what the results are.

*We have rewritten the results section and added more information about changes in meltwater content and the influence of precipitation. We have removed the term "glacial meltwater" (GMW) when referring directly to the results but argue that >90% of the measured meteoric water consists of GMW. We have also changed the title of the manuscript to reference "meteoric water." We carried out an analysis where we recalculated the meteoric and sea ice melt water fractions after adding 2 years worth of precipitation to the water column – the results of this analysis are described in depth in a new Appendix.*

I am a bit confused about the main results and it seems the authors are as well. The flow and progression of ideas is missing with a lot of unsupported speculation rather than solid results and I wonder what the conclusion is here. While some paragraphs are largely true because they are based on a literature review, I am not sure why they are sometimes presented suddenly and how the results of this study influence them.

> We have tried to better explain or eliminate extraneous information throughout the manuscript. We have expanded the discussion of spatial, seasonal and interannual variability. Much of the spatial sensitivity analysis has been moved into the main body of the manuscript, and we now point more explicitly to the Appendix for further depth. We have also added an additional, independent spatial sensitivity analysis.

The glacial meltwater (GMW) term is used in different places, while in other places it is referred to as meteoric freshwater; the terminology is inconsistent and may reflect the suspect methodology used here; it is not possible to separate glacial meltwater from precipitation directly from the combined salinity and 18O observations.

> While it is true that precipitation cannot be explicitly separated from GMW on the basis of $\delta^{18}O$ and salinity alone (since GMW too, consists of precipitation), the $\delta^{18}O$ of Antarctic precipitation at sea level differs substantially from its values in continental precipitation (and thus glacial ice). This is confirmed both by our observations, and by several other studies. We have significantly elaborated on this in the discussion.

While the data cover 26 years, there are in fact seven summers of observations that are not evenly spatially distributed across the region of interest with high spatial variability and a very low amount of data in 1994. While the authors claim a modest increase in meltwater, this is at odd with the insignificant change and interannual variability mentioned in different places in the manuscript that are ultimately consistent with downstream freshening in the Ross Sea. We are puzzled by these contradictory remarks on the evolution of meltwater content. If there is no change, it is difficult to see how this could influence downstream freshening.

> A linear increase in freshening only requires a relatively constant influx of freshwater that is greater than the rate at which it is cycled out by saltier waters being brought on from off the continental shelf. However, the Ross Sea is out of scope for this paper, and so references to the Ross Sea and freshening therein have been removed.

The lack of strong signal is indeed surprising, as one would have assumed that increased melt from ice shelves in the region would have significantly influenced the meltwater content in the

water column. A possible solution would be to adjust the focus of the study and concentrate on why there is such constant meltwater fraction which to me at least is an unexpected and interesting finding to investigate.

> The result of a relatively constant meltwater inventory (after 1994) is now elaborated in further detail in the discussion and conclusion. The result is consistent with more recently published studies (Flexas et al., 2022). The meteoric water column inventories measured in our study might not directly pace mass loss as measured by satellite remote sensing methods. However, assuming relatively a constant seawater residence time, the relative inventories we calculate are indicative of a significant change in melt rates, and further sampling and analysis in the vein of the paper would be diagnostic of changes in the future, if and when they happen.

I apologize for the negative comments, I really think the paper needs more work to make the chain of reasoning clearer and the main results compelling. I have raised a few points below that I hope will be helpful for the authors.

**Specific points:**

Line 38 – I think the authors can add references here, many studies have included 18O to the temperature-salinity combination to define the characteristics of SO water masses.

> We have added the following references: $\delta^{18}O$ (Jacobs et al., 1985, 2002; Meredith et al., 2008, 2010, 2013; Brown et al., 2014; Randall-Goodwin et al., 2015; Silvano et al., 2018; Biddle et al., 2019).

Line 38-39 – The sentence is a bit confusing; zero-salinity is probably too much and I would mention meteoric water rather than glacial freshwater; 18O is useful for differentiating freshwater signals coming from meteoric (precipitation and continental ice) or oceanic (sea ice) sources. References are also needed here

> We are puzzled about "zero-salinity" term being "too much". Meteoric water (which includes precipitation and glacial meltwater) has zero salinity. We have elaborated in

the discussion about how $\delta^{18}O$ and salinity can in-fact be used to discriminate between meteoric water sources, as sea-level precipitation at this latitude has a very different $\delta18O$ relative to even low-altitude continental ice. We have added the following references: (Jacobs et al., 1985; Hellmer et al., 1998; Jacobs et al., 2002; Meredith et al., 2008; Randall-Goodwin et al., 2015), as well as some analysis of all available Antarctic precipitation $\delta18O$ data from the IAEA Global Network of Isotopes in Precipitation (GNIP) database.

Line 39-40 – I do not understand this sentence. Are the authors saying that only in regions where basal melting is deep (and deep relative to what?), glacial meltwater is more depleted in oxygen-18 than local precipitation? Or is the content of glacial meltwater more important than the content of local precipitation in these regions? I am not convinced in either case. Are there any references to support these claims?

This has been edited for clarity. The intention of that sentence was to explain how the subsurface introduction of meteoric water must be dominated by glacial meltwater. Glacial meltwater is a significantly greater freshwater contributor than precipitation in this region, as now elaborated on in the discussion. A section has been added to the Appendix wherein we examine the impact of 2 years of precipitation on the calculated meteoric water inventories.

Figure 1 – The wide map of Antarctica is not very useful here nor is panel b showing only bathymetry. I am not sure how relevant they are to the results. Jet colormap is not perceptually inconsistent and a poor choice for data visualization as it can mask significant changes.

We have changed the colormap to something more linear and added a simplified diagram of the local circulation. We have retained the wide map of Antarctica to locate the field site, and the relative location of the shelf break is likely to be helpful for many readers. We removed the bathymetry-only map and added bathymetry to the maps showing sample locations and column inventories. Sea floor topography provides information about bathymetrically-influenced ocean flow pathways.

Line 84 – Actually no, meteoric freshwater can be continental ice or precipitation, as they cannot be separated on the basis of salinity and 18O.

> We now expand on our rationale in the discussion, noting the very different δ18O contents of sea-level precipitation at this latitude and melting continental ice formed at higher latitudes and elevations. In addition, the highly [18]O-depleted freshwater at depths hundreds of metres below the surface mixed layer have $\delta^{18}O$ values consistent with glacial ice and not with local precipitation.

Line 85 – (Data and Methods)

Line 84-93 – I am not sure about the analyses here. Clearly, the 18O endmember does not vary on interannually, the differences among -28.4‰ and -30.2‰ certainly reflect uncertainty in the data. I do not see the need to separate years here to determine endmembers since the isotopic composition of the continental ice is not expected change suddenly from one year to another. Furthermore, as the authors point out and by Masson-Delmotte et al., 2008 show, there is a large variability in 18O of meteoric water on a local scale, so estimating the endmember separately every year is a key result here.

> Although there is some analytical uncertainty in the data, variability within and between our intercepts is significantly smaller than the variability seen throughout local ice core depth profiles (Steig et al., 2005). Since this study compiles data from different laboratories using different techniques, inter-laboratory variation could impact the results, if the same endmembers are used for each year. Salinity and δ18O data produce a strong mixing line each year, and we think the use of annual mixing line endpoints is the most appropriate procedure for our purposes.

> Masson-Delmotte et al., (2008) do not show large local variability, but demonstrate that 88% of the variability in Antarctic precipitation $\delta^{18}O$ can be attributed to latitude, altitude, and distance from the coast, with altitude having the strongest impact. Their data, and the output from their model show coastal sea-level precipitation at the latitude of our field site with a $\delta^{18}O$ of ~-15‰ – consistent with $\delta^{18}O$ content of multiple local

precipitation samples we collected in 2019, and with several other studies now cited in our discussion (Gat and Gonfiantini, 1981; Ingraham, 1998; Noone and Simmonds, 2002). We also place these measurements and data in context of available δ18O data from the IAEA Global Network of Isotopes in Precipitation (GNIP) database. Precipitation collected at Halley Bay (75.58°S, 20.56°W, 30m elevation) has an average composition of -22.0‰, while that collected at Rothera Point (67.57°S, 68.13°W, 5m elevation) has an average composition of -13.5‰, and precipitation collected at Vernadsky (65.08°S, 63.98W, 20m elevation) has an average composition of -10.2‰ (Global Network of Isotopes in Precipitation (GNIP), 2023).

Figure 2 – There is no discussion of the scatter toward lower salinity and constant 18O from the mixing line above 200 m depth and even in the deep waters in 2009 and 2014. It would be interesting to explore and discuss sea ice imprint if possible

We discuss the scatter at shallow depths as the influence of sea ice, and the scatter shown in 2009 and 2014 in deeper waters as likely resulting from sample storage issues. The revised text makes this more explicit and expands on data quality and interlaboratory offsets in the appendix.

While sea ice fractions are another output from the three-endmember mixing model we use, our focus on meteoric waters includes a specific link to ice shelf basal melt. We have added a discussion of the sea ice melt/formation and mCDW fractions to the appendix.

Line 104 – sea ice melt and/or sea ice formation. The mixing model can give a negative estimate for sea ice endmember reflecting net sea ice formation.

We have amended the text accordingly.

Line 110 – sea ice and meteoric water are not water masses. I would rather write water sources.

We have amended the text accordingly.

Line 125 – influenced by sea ice melt and formation. But also, non-local precipitation, mixing, advection

> Text has been amended to include sea ice formation, and non-local precipitation. Discussion of mixing and advection are expanded upon in the discussion.

Line 128 – I am not sure how this approach differs from studies that use an average 18O for meteoric water. Biddle et al., 2019 and Meredith et al., 2010 do not use approximate 18O values for glacier but a plausible average of meteoric water because again 18O does not disentangle continental ice and precipitation. Using the zero-salinity intercept on 18O-salinity plots, the authors use the same method; they deduce an average 18O of meteoric water in region where this component is highly variable (Masson-Delmotte et al., 2008).

> Meredith et al., 2008, 2010, 2013; Biddle et al., 2019; Randall-Goodwin et al., 2015 use plausible average meteoric water values. Meredith et al. 2008 shows mCDW-precipitation and mCDW-glacial melt mixing lines on salinity-$\delta^{18}$O plots, with surface water observations falling between the two, as they did for their "mean meteoric water" calculations. Biddle et al., 2019 adopts values used by Randall-Goodwin et al. 2015, who followed Meredith at al. 2008 by selecting a midpoint between a range of freshwater endmembers.
>
> In this study, we demonstrate that by producing a mixing line between mCDW and a freshwater (meteoric) endmember for subsurface (>200m) depths, the extrapolated 0-salinity intercept more tightly constrains the (glacial) meteoric water endmember. This procedure reveals the fingerprint of glacial meltwater introduced at depth. While surface waters may contain precipitation with a less-depleted $\delta^{18}$O signature than deep waters, we demonstrate that >92% of the meteoric water can be assumed to be comprised of glacial meltwater.
>
> Choosing a midpoint between precipitation and glacial ice as an "average" meteoric water endmember assumes a 50/50 mixture of precipitation and glacial melt. Since the meteoric water is dominated (>92%) by GMW, using a precipitation-GMW midpoint endmember will overestimate meteoric water fractions and underestimate sea ice melt

fractions. It could be argued that an endmember comprised of ~92% GMW (~-30‰) and ~8% precipitation (~-15‰) would be the most appropriate "mean" meteoric endmember (~-28.8‰), however our primary interest is in basal melt, so using the zero-salinity freshwater defined by $\delta^{18}$O-salinity >200m is most appropriate.

Line 149 – The data are collected during summers so it is unlikely that GMW endmember values are based on the average of the annual data

This is a miscommunication in phrasing. What we intend to describe is that the meteoric water endmember we use for each year is the one produced using only that year's data. While these data were all collected during summer, and the region experiences seasonal variability, the residence time of seawater here is ~2 years (Tamsitt et al., 2021), reducing the impact of seasonality. We have added discussion about the seasonal variability of mCDW in the region.

Line 160-162 – I do not see the connection between the results discussed in figure 3 and the mCDW heat extent

The discussion of remaining mCDW heat and the formation of a polynya are perhaps beyond the scope for this study, and we have removed this inference.

Line 145-162 – I am not convinced that the authors are properly discussing the results here, but rather a speculative explanation of the origin of glacial melt

We have changed references to GMW to meteoric waters when discussing our results. However, the distributions of meteoric water shown in our results indicate introduction of basal meltwater at depth. Revised figures show this more clearly, as do added references demonstrating the depth of outflow from beneath ice shelves. (Biddle et al., 2017; Naveira Garabato et al., 2017)

Line 167 – The authors point to a modest increase (what is a modest increase? relative to what?) of the mean GMW inventories and at the same time acknowledge that the low average in 1994

may be responsible for this 'trend'. The low number of samples in 1994 should be reflected in the estimate, and if this year is not taking into account, no claim of an increase can be made. Also, how is the linear trend calculated? Are there any uncertainties associated with this calculation?

> The linear trend is calculated from the mean meteoric water column inventories as calculated using the Gaussian fit lines depicted in Figure 3. This has been made more explicit. The low number of samples in 1994 is accounted for in the depicted uncertainty, aligns with estimates from other years, and displays the strongest fit. We have added further expressions of uncertainty to Figure 4, and discussion of the pattern of results – revealing little about changes in meteoric water inventory, other than an increase since 1994. We also discuss the consistency between the pattern of our results and a more recently published modeling study ( Flexas et al., 2022).

Line 168-170 – I do not see the link between these two sentences and the discussion of results here, what is the connection between interannual GMW inventories in summer and seasonal variability of mCDW in the region? It is hard to understand; the authors claim a modest increase of GMW inventories and then refer to invariable overall melt rates during the austral summer when the samples used in this study were collected (do they mention a particular year?)

> The mention of relative stability in mCDW and melt rates, along with the ~2 year residence time of waters in this region was intended to show that the meteoric water inventories described by our results are not simply a seasonal melt signal, but representative of a longer period of change. The discussion of variability in mCDW properties is intended to describe the impact of a source of uncertainty. This has been rewritten for clarity, and the discussion of the impact of mCDW variability moved to the discussion.

Figure 4 – It is nice to see the series here, but I doubt the authors are showing a volume as mentioned in the caption. Also, integrating from the surface will include precipitation, even if it is negligible. Therefore, I do not really agree with the following assumption; depth-integrated GMW between surface and 800 m depth. How is the linear regression calculated? What is the

uncertainty? If the GMW inventory is time invariant as the authors claim, beside a modest increase (a choice has to be done here), I think the linear regression in the figure does not provide crucial information.

> The caption has been amended to show meteoric water content. An added section discusses the impact of precipitation on the meteoric water column content. The linear regression is calculated from the meteoric water column inventories produced from the Gaussian fits in Figure 3, updated with appropriate uncertainty expressions. While the trend is not statistically significant, an increase from 1994 is evident "by eye" and we feel the trend is worth retaining.

Line 177 – This section belongs to Methods rather than Results Line 178-180 – I assume this analysis corresponds to Appendix A4?

> We have retained uncertainty analysis in the Results section, but it has been rewritten and now contains several tables and figure displaying the results of uncertainty analyses. References to the relevant appendices have been made more explicit throughout the text.

Line 181 – Then why use a single salinity and 18O value for mCDW each year if the authors claim that this water mass is relatively stable over time?

> The data compiled for this study comes from different labs and was analyzed using different techniques. Calculating results based on each dataset independently minimizes concerns over inter-lab offsets. We have expanded the discussion of seasonal and interannual variability of mCDW in the discussion and added another section to the Appendix about data quality and offsets.

Line 184 – Do the authors really compare GMW content and 18O-salinity relationships? I do not see any comparisons of 18O-salinity relationships in Appendix A4

> The meteoric water endmembers used for each geographic grouping were defined based on $\delta^{18}$O-salinity data for only those stations, so any comparison of the meteoric water

endmembers effectively compares the δ¹⁸O-salinity relationships. Appendix A4 has been rewritten for clarity.

Line 186 – I am not convinced to the authors' decision to simply remove the 2014 data near to TGT from the analysis (which are still shown in figure 1 in glacial meltwater inventory panel g and I assume in figure 2 as well? Which is confusing) to improve interannual comparability. I would keep all available data from the region if the aim is to make comparisons on a regional scale. Excluding data because the GMW values are simply higher seems problematic and not a good reason to me.

We have added the 2014 data alongside Thwaites back into the broader analysis and discussion, noting the higher inventories at these sampling locations. There is also further discussion of the inventories alongside Thwaites in our updated spatial sensitivity analysis.

Line 196 – But the authors stated earlier that the properties of mCDW endmember are invariant

This is a description of model sensitivity. While the mCDW signatures are relatively stable and the most well-constrained endmember, changes to the selected mCDW endmember have a larger impact on the outputs of the 3-endmember mixing model than changes to other endmembers.

Line 199 – Unable to estimate glacial meltwater using salinity and 18O

See responses to earlier comments re: distinguishing between sea level vs continental precipitation.

Line 201 – and compare GMW fractions rather than d18O values. Not helpful, it did not need to be said

This was intended to emphasize that the calculated meteoric water content will not be influenced by inter-laboratory variability in δ¹⁸O values with our method. The discussion has been rewritten.

Line 203 – Are there any uncertainties in the meltwater content values? I do not think low and high are useful here

> Uncertainties in the meteoric water content values are described in that analysis. Corresponding uncertainties have been added to the discussion where relevant.

Line 215 – Hard to tell if 2000 was a local high compared with subsequent years due to uncertainties. And what about the year 2020?

> We have rewritten the discussion. 2020 measures the highest average meteoric water column inventory (though 2000 is very close). We have also added a table summarizing the results.

Line 217 – Again, it is not clear that there was an increase in average GMW after 1994

> Discussion has been rewritten to more appropriately describe the increase after 1994, followed by relative stability thereafter.

Line 219 – How is a steady GMW inventory consistent with a linear, long-term freshening trend? This assumption, which I think is confusing, does not add much here because the study does not examine the contribution of freshwater input on the reported downstream freshening in the Ross Sea

> Linear freshening in the Ross Sea does not require accelerating freshwater input – only consistent freshwater input at a level above its output rate. However, discussion of the Ross Sea freshening is out of scope for this paper and has been removed.

Line 222 – meteoric water inventories from the surface to 800 m. Integrating from the surface will include precipitation

> We have added a section to the discussion with more in-depth analysis and discussion of the influence of precipitation on meteoric water inventories, including a new Appendix.

223 – volume or content?

       Corrected to content.

224 – gyre-like circulation; adding regional circulation to the map would be helpful and it would influence meltwater advection

       We have amended Figure 1 to include a simplified schematic of local circulation.

243 – If it is statistically insignificant, there is no linear increase, I am not sure it is useful

       We have modified our discussion of the trend in meltwater inventories, however retained the description of the linear increase, with multiple updated measures of uncertainty.

244 – This is tricky, because of the error bars, the lowest and highest melt periods claimed seem difficult to believe

       Text has been revised to better describe the uncertainty and (lack of) significance in the trend. We have also added multiple expressions of uncertainty to the figure.

250 – 18O observations do not allow estimation of basal melt rates

       While $\delta^{18}O$ (and salinity) observations alone cannot allow the estimation of basal melt rates, with the ~2 year residence time of waters here, so our results will integrate meteoric (GMW) content over a similar timescale. A sudden change in the average meteoric water column inventories are strong indications of a change in basal melt rates, and would be diagnostic thereof. The discussion has been amended to more clearly reflect our intention.

251 – interannual fluctuations potentially masking an increase over 2.6 decades; this is very speculative

We have removed this sentence from the conclusion and rewritten it in a way that is less speculative, and more explicitly referential to our results. It now focuses more on the technique's utility for diagnosing changes in melt rates (assuming a relatively constant residence time).

255 – the last sentence is very confusing; how can meltwater volume rates be measured 18O observations? How is the invariant GMW inventory mentioned in the paper consistent with any downstream freshening

As mentioned previously – a linear freshening only requires a constant influx of freshwater that is out of balance with the system's capacity to cycle it out. However, Ross Sea freshening is out-of-scope, and we have removed this mention. We have changed the reference to "meltwater volume" to meltwater content.

As in response to a previous comment, while $\delta^{18}O$ and salinity cannot be used to directly and explicitly assess melt rates, a sudden change in measured meltwater content would be diagnostic of a change in basal melt rates, since the measured meteoric (meltwater) content integrates melt over the residence time (~2 years) of waters here.

**References**

[revised manuscript text omitted]

---

## Author Comment (AC2)

**Glacial meltwater in the Southeast Amundsen Sea: A timeseries from 1994-2020**

Andrew N. Hennig, David A. Mucciarone, Stanley S. Jacobs, Richard A. Mortlock, Robert B. Dunbar

**Response to Referee #2 Comments**

Thank you very much for reviewing our submission with insightful comments. A revised discussion and text, with additions to the Appendices, has improved the manuscript in ways that we trust will satisfy your concerns.

**General comments:**

The pace of melting of Antarctic ice shelves due to warming along the coastal margin and the associated changes in the grounded ice sheet are a major concern in terms of future sea level rise. Models that are used to project future changes still entail large uncertainties and current estimates of changes largely stem from remote sensing data. Ocean tracer measurements that can be used to quantify the glacial meltwater content and its changes accumulated in the ocean provide an opportunity to better understand the melting of ice shelves and its temporal variability.

The study by Hennig et al. provides novel data collected over more than two decades from the Amundsen Sea sector, which is a region where a large increase in melt has been reported previously, mainly driven by warm water intrusion on the shelf. Using the isotopic composition, they find that the regional freshwater budget is dominated by glacial meltwater and that the meltwater inventory exhibits large decadal fluctuations superimposed on a comparatively small long-term trend. These results support other recent studies based on remote sensing data that have found substantial fluctuations of the ice shelf melt on decadal time scales.

This is a very timely and interesting study that is of importance to the wider Antarctic ice shelf and ice sheet community as well as the oceanographic community. It is overall well written and I

think that the methods are mostly robust and support the results. Particularly the authors' approach to circumvent issues of laboratory offsets in the isotopic measurements, that have been a known issue for a while, is quite elegant and I think leads to meaningful results. However, I also think that the paper would benefit from a more in-depth comparison to previous work and from highlighting the novel aspects of this work more clearly. In addition, I have some concerns regarding the uncertainty discussion, in particular to biases induced by the spatial sampling and I think that caveats should be communicated more clearly. Overall, I think that the manuscript is suitable in principal for publication in The Cryosphere, after addressing some points.

> We have expanded the discussion, including a direct comparison to the results of the original study using the 2014 data (Biddle et al., 2019). We have revised our spatial sensitivity analysis along the lines of your suggestions, and moved much of that content into the main body of the paper. We have also conducted another independent spatial sensitivity analysis, which is also presented in the main manuscript body.

**Specific comments:**

1. I think that the motivation for this study and the importance of the results is not communicated sufficiently. Currently there is a strong focus and emphasis on the collection of a timeseries, but very little on why the timeseries is collected and what we can learn from such a timeseries. I think that discussing this in more detail, in particular in relation to the recent literature on the temporal evolution of melt in the Amundsen Sea, is critical to highlight the novelty of the results. A particular example is the following sentence in the introduction (P2L31-33): "[…] some studies have shown a greater interannual variability in the basal melt rates than increase […], and some have even suggested a slowing of basal melt rates […] and grounding line retreat […]." I think that this point has to be extended by rewriting the sentence, adding a time perspective (what happened when / what timescales are we talking about), has to put into perspective of natural climate variability versus anthropogenic forcing, and used as an explicit motivation for the study and how the seawater isotopic composition might help to contribute to this discussion.

> The data used in this study was not collected with the explicit intent of producing a timeseries of meteoric water inventories (or even a timeseries of seawater $\delta^{18}O$ data) but was compiled from multiple datasets collected independently for different projects. This has now been clarified in the Introduction and Methods. Most data were obtained with the intention of enhancing understanding of ocean-ice shelf interactions and melting at the time of measurement. We have rewritten the discussion of interannual variability of basal melt in the introduction.

> The motivation for this study was to aggregate as much $\delta^{18}O$ data as possible from a single region important to the WAIS, and use it to examine changes through time. This allows us to assess the viability of the technique for monitoring basal meltwater input from ice sheets.

2. Following the point above, I think that the paper would benefit from an extension of the discussion on the temporal variability shown in Figure 4. To me, this is the key result of the paper. However, the discussion on details in variability seen in this Figure and how they relate to other recent findings and what new aspects can be learned from this Figure is very limited. In fact, there is not even a reference to Figure 4 in the main text.

> We have revised our results section and now Figure 4 more explicitly. The discussion section has been expanded, and now includes consideration of variability in meteoric water content through time. We focus on the utility of this technique for identifying changes in melt rates, and its potential utility in better constraining mass loss through basal melt.

3. P4-5L80-82: I think that the approach taken here indeed mitigates some of the known issues of salt effects between IRMS and CRDS. However, it is not very clear in these sentences here that the salt effect is indirectly removed by using different CDW reference values for each respective data set. I think that should be written more explicitly at this point. In addition it might be useful to actually point to the differences in CDW d18O in Table 2 where the CRDS measurements (2019/2020) yield a much lower CDW value than the IRMS measurements (2014). Is this difference in line with the values reported for the salt effect in literature?

We ran a subset of 100 samples from 2019 and 2020 on both IRMS and CRDS systems and observed no analytical offset between the two instruments. The literature on possible salt effects on seawater $\delta^{18}O$ measurements shows inconsistent offsets between instruments and labs, and is based on a very small number of samples. We are not convinced that there is a significant salt effect impacting our results as well as all other published paired isotopic datasets from CRDS and IRMS methods.

We now explicitly describe how the potential impact of interlaboratory offsets is indirectly removed by defining mCDW and meteoric water sources using data from each year separately. We have also added a section and pointed the reader to the appendix where we discuss the observed offset between the 2014 data and other years.

4. P7L132-135: I think it is important to discuss the difference in results associated with using a constant and varying mCDW and meteoric endmember at this point. A constant value would yield a GMW estimate that is spatially integrated and the varying endmember yields local fluxes. Likely, this choice will also affect the long-term trends in the GMW estimate (largely through changes in the meteoric endmember), which I think should be discussed as a possible caveat at this point.

We have added an extended discussion about the utility of defining the meteoric water endmember using salinity-$\delta^{18}O$ intercepts. We include a comparison of the 2014 data using our methods, vs those used in Biddle et al. (2019), where the 2014 data were originally published.

5. I am still a bit concerned about potential artifacts from the changes in the spatial sampling from one year to the other. Fig. 1 and also Fig. A3 clearly show substantial spatial differences in GMW content in the region and I think that the paragraph on p. 9 Lines 184-187 is not sufficiently accounting for the issue. I appreciate that this issue is investigated in Section A4. However, I think that the manuscript would benefit in terms of the credibility of the results, if a more detailed spatial analysis was added to the main text. In the end, the main results in Figure 4 are interannual variations with a magnitude of about 1.5m, which seems to be within the range of spatial variations shown in Figs. 1 and A3.

o So, I am wondering if the reported uncertainties in Table A2, last column ("Average GMW inventory (m)), as well as the uncertainties shown in the main text also include the spatial standard deviation of the samples? Is this included in the "environmental" uncertainty within the Montecarlo simulation? I think that it would be transparent and beneficial to simply report the spatial standard deviation of GMW for each box also in Table A2, which would give a measure of the range of spatial variations.

o In addition, I have difficulties understanding how the boxes were chosen and why they seem to be not consistent between the years, i.e. sometimes a location falls in one box and sometimes in another. I think it would be helpful to have boxes that are rather fixed in time and represent certain regimes within the region. For example, I found the Boxes in Fig. A3 for 2014 quite logic, since there is an "offshore" box (c), a TGT box (d), a PIIS box (a) and a central box (b). Looking at these boxes over all years and samples would be, i.e. having a figure similar to Figure 4 for each of these regions would be very helpful to understand how the variability might differ spatially and if the variability is a signal that is consistent across the entire domain or just arises from local signals would be very helpful to have. I would suggest to actually have a figure like this with a brief discussion in the main text if possible.

The boxes used in the spatial sensitivity analysis were based on groupings of stations as sampled. The groupings were selected based broadly on the criteria described in your comment – a group as close as possible to the ice shelf, a second group more distant from the ice shelf front, a third further offshore – and a fourth group around Thwaites Glacier Tongue for 2014.

We have redone the spatial sensitivity analysis across all years using consistent geographic boundaries. We have tried (occasionally unsuccessfully) to draw boxes in a way to avoid any groupings with only 1 or 2 stations.

We have also added an additional spatial sensitivity analysis, wherein results were calculated using random selections of 3 stations; this process was performed 10,000

times, with the standard deviation measured as the standard deviation of meteoric water inventories produced using 10,000 random groups of 3.

The Monte Carlo simulations described in the original manuscript will incorporate some spatial variation of the endmembers – mCDW and meteoric water endmembers vary in each simulation based on the observations. All three endmembers (mCDW, meteoric water, sea ice melt) are subject to appropriate environmental uncertainty. For mCDW, that uncertainty is based on the variation in mCDW S and $\delta^{18}O$ signatures observed across the whole field site, and for meteoric water, the uncertainty is based on the standard deviation of $\delta^{18}O$ in the nearest ice core (ITASE01-2).

The spatial sensitivity analyses, with supplementary tables have been moved up into the main body of the manuscript, with more detailed results tabulated in the Appendix.

6. I am a bit concerned about the conclusion (P11 Line 243) that the long-term trend is insignificant without discussing the fact that this only reflects the data presented here but might not reflect the actual trend in the melting. It would be good to discuss some of the caveats of the use of the data set and its limitations. In particular, I think that the data set will not capture the entire amount of meltwater coming from the Amundsen Sea, as the authors' report that the residence time of the water in the region is only about 1 year. So, it may well be that there is a strong long-term trend in glacial melt in the region, but that the signal largely propagates out of the region and does not accumulate there. Also, the fact that the endmembers vary throughout the years, in particular the glacial melt endmember, could affect the long-term trend. So, I think it is important to discuss such potential limitations here.

While not all of the Amundsen Sea meltwater will accumulate in our study area, all of that from the Pine Island Ice Shelf, and much of that from the Thwaites Ice Shelf will necessarily pass through our study area, so the results we present are likely specific to those two ice shelves. While the glacial melt endmember can be expected to remain relatively stable, it will vary with depth (ITASE01-2 and Siple Dome ice cores have $\delta^{18}O$ standard deviations of 1.9‰ $\delta^{18}O$ – greater than the standard deviation of our yearly meteoric water endmembers) and would not be expected to be static through

time. Given the extrapolation required to determine meteoric water endmemebers as we do in this paper, sampling and analytical uncertainty will also play a role.

We have expanded the discussion and conclusion to include the limitations of measuring meltwater content this way, including the influence of residence time, and the export of meltwater. Icebergs calving will be a significant component of glacier mass loss (and a contributor of glacial freshwater flux to the Southern Ocean), however if melting occurs outside of the study area, this component of mass loss will not be accounted for in the meteoric water inventories.

**Technical corrections:**

- P2L35: I don't think that "SE" has been defined yet.

  Corrected

- Figure 1: Please do not use "rainbow" colormaps that are not scientific colormaps. For detailed reasons and tools to generate an appropriate colorbar e.g. for Matlab, please see for example this paper by Stauffer et al. (2015; https://doi.org/10.1175/BAMS-D-13-00155.1)

  We have revised Figure 1 with a more appropriate colormap.

- Figure 2: I found it difficult to depict the difference in blue. Since only dark blue is used, it may be good to keep those dark blue sample and exchange the other blue(s) by gray.

  We have revised Figure 2 using a gray colormap showing all depths, which is more illustrative of the deep-shallow mixing line described elsewhere in the paper.

- P6L107: Probably important to add that also "sea ice formation and melt" will affect the signal at this point.

  Thank you – reference to sea ice melt and formation added here.

- Equations 1-3: the placement of these equations seems odd as there are somewhere in the text where they are not discussed. Please place them right below a description of and reference to these Equations.

  We have added a reference to the equations in the text and revised the text to more clearly reference them.

- P6L125: I guess it should be not just sea ice melt but also "formation"

  Added reference to sea ice formation.

- P7L149: I think that "extremely unlikely" is a stretch here. Please reformulate to "[…] mean, the potential impact of analytical calibration offsets between laboratories on the calculated GMW fractions are mitigated.

  Corrected

- P8L154: "mathematical artifacts" seems odd and would not understand what is meant, do you mean "sampling and analytical uncertainties"?

  Text amended as suggested.

- P8L166: It is unclear at this point where the uncertainty estimate is coming from. Could you please refer to the part of the manuscript where it is calculated and/or briefly mention it here.

  We have described the uncertainty estimate and directed the reader to the discussion for further depth.

- Figure 4: Is this trend statistically significant or not. Please report the statistical significance here.

  We have added additional measures of uncertainty (standard deviation, 95% confidence interval, p-value) here to expound upon the statistical significance of the trend.

- P10L194: I have difficulties understanding the meaning of "a range of the range" (e.g. +-1.5 – 1.7 g/kg) in uncertainty, if I understand this correctly. Please report a single range, e.g. +-1.7 g/kg.

  The reason for a range of values is because the uncertainty varies by year, largely dependent on the fit of the data in $\delta^{18}O$ -salinity space. In 1994, 2019, 2020, the uncertainty of the meteoric and mCDW endmembers is quite low, while in 2009 and 2014 it is higher, due to the spread of the data. We have adjusted the text to make this clearer.

- P10L201: Change to "This minimizes systematic isotopic offsets"

  Text amended.

- P11L249: I think what the authors are really trying to say here is that the decadal variability of the melt is actually substantially larger than the long term trend (1994 to 2020). The way that this is currently written it is difficult to understand what is actually meant. It seems not surprising that there is interannual variability in the first place, but what is in fact interesting is the magnitude of the variability compared to the trend and the time scale over which this variability occurs. I think that needs clarification.

  We have amended the text to make the meaning clearer.

- P11/12L252-253: It would be helpful to note at this point that the tracer approach has the advantage that the ocean integrates the temporal meltwater changes and thus a single measurement actually reflects a longer period of melting.

  Thank you – we have added a description of the period captured by these meltwater measurements.

- P13L260-264: Please correct typological and formatting errors.

  Errors corrected.

- P17L331: I have difficulties understanding the meaning of "a range of the range" (e.g. +-0.5 – 0.7 m) in uncertainty, if I understand this correctly. Please report a single range, e.g. +-0.7 m.

  The reason for a range of values is because the uncertainty varies by year. In 1994, 2019, 2020, the uncertainty of the meteoric and mCDW endmembers is quite low, while in 2009 and 2014 it is higher, due to the spread of the data. We have adjusted the text to make this clearer.

- P17L333: I think that this should read "95.1%", right? Otherwise I would not understand this number.

  Typo corrected, and the description has been expanded upon for clarity.

- Generally, the numbering of subsections is wrong; always starts with "1.x"

  Thanks - Appendix subsection numbering has been amended.

**References**

Biddle, L. C., Loose, B., and Heywood, K. J.: Upper Ocean Distribution of Glacial Meltwater in the Amundsen Sea, Antarctica, J. Geophys. Res. Oceans, 124, https://doi.org/10.1029/2019JC015133, 2019.

---

## Referee Report (RR1)

I am pleased to see that the authors have addressed some of my detailed comments and suggestions. I still hope that this paper, with its valuable technical insights, will be published for the benefit of the community. However, it is somewhat frustrating to note that most of my suggestions, which would have required substantial revisions to the manuscript, were politely rejected by the authors and/or does not reflect the work needed in the manuscript. The paper may benefit from further work, additional analyses, or retaining the annexes to create a new comprehensive technical paper. In light of my significant disagreements with the authors, both previously in Nature Comm&Env and now in The Cryosphere, I have chosen to terminate my review at this juncture. This decision stems from the realization that I may no longer be the most qualified individual to provide a fair assessment. Additionally, I would like to reiterate my primary concern:

- One particular concern is the authors' argument, unproven, that the use of salinity and $\delta^{18}O$ in tandem allows for the estimation of glacial meltwater content, for instance: this statement is made in Line 49. Additionally, in the Methods section, they assert that meteoric water deeper than 200 meters is dominated by ice shelf basal melt, but they also mention surface waters (Lines 111 and 131). It isn't until Line 220 that the authors discuss that meteoric water inventories are likely composed of 90% glacial meltwater. It would be more logical to introduce this hypothesis, supported by literature and analyses, in the Methods section to establish that meteoric water in this study is synonymous with glacial meltwater. This consistency would eliminate the need to use terms like 'glacial' or 'GMW' in parentheses in the manuscript. I concur that deep freshwater primarily originates from glacial melt, which is not a novel observation. However, considering that $\delta^{18}O$ cannot effectively distinguish meteoric sources in this region, it would have been beneficial for this hypothesis to have been proposed and substantiated earlier in the study.

- Regarding the methodology, the authors suggest using a simple linear regression on all data below 200 meters depth to constrain the meteoric water endmember each sampling year. Upon initial examination of the $\delta^{18}O$ /S plot, the data appear to form a single mixing line for salinities greater than 34. This is surprising considering the clear distinction in the T/S plot between waters influenced by melt and those composed of simple CDW/WW mixtures with WW characterized by salinity higher than 34 and temperatures near the surface freezing point. The implication is that runoff and precipitation are less negative in δ18O than ice shelf basal melt. However, when sea ice formation increases surface water salinity to around 34, the $\delta^{18}O$/S of the WW ends up to the right of the meltwater mixing line influencing the CDW/GMW line estimated by the authors. The extrapolation of the meltwater mixing line to zero salinity gives a more negative intercept due to the influence of deep WW. Furthermore, the authors must acknowledge that the estimated freshwater content reflects contributions from large spatial and temporal scales, including meltwater from ice shelves upstream, icebergs within the Bellingshausen Sea and WAP, and precipitation inputs carried into the region by surface circulation. The use of data below 200 meters excludes a portion of meteoric water in the surface and subsurface layers. Therefore, it is essential to consider the broader context of freshwater sources and their uncertainties. Distinguishing subsurface glacial meltwater, a meteoric input, from surface-derived meteoric inputs using $\delta^{18}O$ is challenging, as isotopic values in local precipitation and ice shelf melt are similar. The age and source of ice melting at the base of the ice shelves remain uncertain, further complicating the $\delta^{18}O$ analysis. Given these uncertainties, considering a larger range of plausible $\delta^{18}O$ meteoric endmembers based on

regional literature would provide more realistic estimates with appropriate uncertainties. Finally, I also disagree with the authors' claim that Antarctic precipitation at sea level substantially differs from its values in continental precipitation and glacial ice in the region of their study.

- In their mass balance calculation only three endmembers are used, and even I am a passionate user of this approach, it does make numerous assumptions, necessitating the consideration and propagation of errors. Among other water masses in the region, the authors also fail to address the presence of WW in their analysis, which can extend deeper than 200m. The use of 200 meters as the limit between the surface layer and the mCDW layer is questionable and requires clarification. Again, the authors rely on a single value of approximately -30‰ to represent the $\delta^{18}O$ of GMW, which is weak given the accumulation of meteoric water in the WW layers from various sources, both local and remote. The authors claim that the mean $\delta^{18}O$ of the glacial melt may be represented by their methodology, but it does not offer a more accurate estimate of freshwater source fractions compared to previous studies by for instance Meredith et al. 2010 or Biddle et al., 2019 (but there are many other works), because without a proper GMW isotopic composition range, they might over or under estimate the freshwater source fractions. An invariant $\delta^{18}O$ endmember is not a surprise as we would not expect GMW isotopic change in annual time scales. The primary source of uncertainty lies in the meteoric endmember, which dominates the uncertainty in the freshwater fractions. Furthermore, subsurface inputs of glacial meltwater can be differentiated from other freshwater sources based on potential temperature and dissolved oxygen concentration, making $\delta^{18}O$ analysis less advantageous. Since the authors concentrated solely on meltwater fractions in the main manuscript without discussing any sea ice influence, they could be calculated using the Gade line analysis. Therefore, I fail to see the advantage of employing $\delta^{18}O$ in this context, again.

- Another issue is the potential outlier or interannual variability in the year 1994. The claim of an increase in meteoric water inventories based on data spanning only seven years between 1994 and 2020, with limited observations in the initial year, lacks statistical significance. The authors assert the stability of meteoric water inventories after 1994, but it's important to note that this conclusion is based solely on the data presented in this study, which spans only seven sampling years and may not fully represent the broader dynamics of meteoric water in the region. These findings also do not account for the significant 2012 event in the region, and overall, they suggest a relatively stable pattern. The absence of sensitivity tests on the linear regression is a concern, and the authors' reliance on visual assessment to justify the trend as an answer to my previous review is not scientifically robust.

- Additionally, it is unclear whether the authors are using problematic data in their analysis, likely stemming from sample storage issues. If these data points are problematic and from storage issues, they should not be discussed in the main results as they do not contribute meaningful insights to the study.

L. 90: What does "scrutinized visually" entail or involve?

L. 96: If the comparison between CRDS and IRMS analyses revealed no discernible salt effect, it could be argued that this paragraph may not provide significant value to the current discussion and might be more appropriately placed in an appendix.

L. 140: I'm having difficulty comprehending the trends that the authors are discussing

L. 193: The characteristics of mCDW are derived from observations, while those of meteoric water are estimated rather than directly observed in this context

Figure 3: If the authors assert that negative meteoric fractions, which are theoretically impossible, result from erroneously flagged data, they may need to consider either removing such data points and explain why from the analysis or adjusting the mass balance calculation to restrict negativity to only the sea ice term.

L. 210: What is the rationale for discussing sea ice melt and mCDW fractions in the appendix?

L. 219: What does the environmental variability of the model inputs entail?

L. 233: I continue to be impressed by the superior analytical precision offered by CRDS compared to IRMS.

L. 236: How do the authors go about estimating the uncertainty related to environmental variability? Additionally, while the term "environmental variability" is recurrent in the text, I'm still curious about the specific processes that the authors consider within this framework.

L. 281: Is there any substantial evidence supporting the notion that the increased inventories in group B are indicative of basal melt accumulation, or is this interpretation more in the realm of speculation?

L. 283: With the exception of those locations adjacent to the TIS, the sensitivity of sampling location remains a significant factor to consider.

L. 393: This is a speculative assertion.

L. 397: The GMW inputs exhibit consistent standard deviations from 1994 to 2020. However, the authors do highlight a lower meteoric water content in 1994 and a higher content between 2000 and 2020.

---

## Author Response (AR2)

**Meteoric water and glacial melt in the Southeast Amundsen Sea: A timeseries from 1994-2020**

Andrew N. Hennig, David A. Mucciarone, Stanley S. Jacobs, Richard A. Mortlock, Robert B. Dunbar

**Response to Reviews**

**Editor comment**

I am sorry for the long review process, but I thank the two referees for their comprehensive second examination of your work. You will see that in spite of several improvements, they still require major revisions. Please provide a point-by-point response to their comments and revise manuscript where necessary. I consider that the discussion on precipitation minus evaporation versus glacial meltwater has clearly improved the manuscript, although it could be clarified earlier. On this point, you may consider adding a reference to Bett et al. (JGR, 2020, their Figs. 5-7) to further support your claim. You will see that this reference also makes it clear that the Amundsen Sea continental shelf is a region of net sea ice production, and it therefore seems essential that you address the 2nd comment from Referee#2 on sea ice formation/melt.

Thank you for your comments. The revised manuscript brings our discussion of GMW vs precipitation fraction of meteoric water closer to the beginning of the manuscript. This conclusion is based on our results, so we have retained the extended discussion/explanation later in the manuscript. We have also re-analyzed our dataset and now include a fraction of precipitation in the upper water column, which we think reviewers will find agreeable. With respect to major revisions, we cannot respond further to reviewer #1's main issues without degrading the credibility of our paper. This is discussed below.

I also would like to add a personal comment: models show that the coastal circulation in the Amundsen Sea is intensified for higher melt rates (Jourdain et al., JGR 2017, Fig. 8). This makes the interpretation of meteoric water inventory in terms of glacial meltwater trends more complicated as more melt is associated with an intensified export of meltwater. This nonetheless only applies to the glacial meltwater coming from ice shelf melting (the iceberg part being non negligible, e.g., in Rignot et al., Science, 2013 or Greene et al., Nature, 2022).

Our revised manuscript includes discussion on this point. Any ice sheet mass loss via exported icebergs will, of course, not be included in our meteoric water inventories. We have added further discussion on the meltwater potential contained within icebergs, as well as the circulation caveats around using mean inventories to interpret melt trends.

**Referee #1**

I am pleased to see that the authors have addressed some of my detailed comments and suggestions. I still hope that this paper, with its valuable technical insights, will be published for the benefit of the community. However, it is somewhat frustrating to note that most of my suggestions, which would have required substantial revisions to the manuscript, were politely rejected by the authors and/or does not reflect the work needed in the manuscript. The paper may benefit from further work, additional analyses, or retaining the annexes to create a new comprehensive technical paper. In light of my significant disagreements with the authors, both previously in Nature Comm&Env and now in The Cryosphere, I have chosen to terminate my review at this juncture. This decision stems from the realization that I may no longer be the most qualified individual to provide a fair assessment. Additionally, I would like to reiterate my primary concern:

With respect to what seems to be their primary concern (that GMW and precipitation in this region cannot be distinguished based on $\delta^{18}O$), as this is an undeniably incorrect view on their part. There are a few comments made by this reviewer that have allowed us to improve the manuscript.

- One particular concern is the authors' argument, unproven, that the use of salinity and $\delta18O$ in tandem allows for the estimation of glacial meltwater content, for instance: this statement is made in Line 49. Additionally, in the Methods section, they assert that meteoric water deeper than 200 meters is dominated by ice shelf basal melt, but they also mention surface waters (Lines 111 and 131). It isn't until Line 220 that the authors discuss that meteoric water inventories are likely composed of 90% glacial meltwater. It would be more logical to introduce this hypothesis, supported by literature and analyses, in the Methods section to establish that meteoric water in this study is synonymous with glacial meltwater. This consistency would eliminate the need to use terms like 'glacial' or 'GMW' in parentheses in the manuscript. I concur that deep freshwater primarily originates from glacial melt, which is not a novel observation. However, considering that $\delta18O$ cannot effectively distinguish meteoric sources in this region, it would have been beneficial for this hypothesis to have been proposed and substantiated earlier in the study.

Many other studies have distinguished between precipitation and glacial melt using $\delta^{18}O$. (Jacobs et al., 1985; Meredith et al., 2008, 2010, 2013; Randall-Goodwin et al., 2015). Several studies have used mixing lines on $\delta^{18}O$-salinity plots to distinguish these freshwater sources (Fairbanks, 1982; Jacobs et al., 1985; Meredith et al., 2008, 2010, 2013; Randall-Goodwin et al., 2015). A study by Potter and Paren (1985) observed that the ice flux into the ice shelf had a $\delta^{18}O$ value of around -20‰, whereas direct accumulation onto the northern part of the ice shelf in the form of precipitation had a much higher $\delta^{18}O$ value, around -13‰. This study and the ability to distinguish between precipitation and glacial melt is also called out explicitly in (Meredith et al., 2008).

Our greatly expanded discussion cites all available and accessible coastal precipitation data from the northern WAP to well west of our study area. These data show a clear latitude-correlated trend. Snow we collected and analyzed in the field during the 2019 cruise had a $\delta^{18}O$ of -15‰ (while the local glacial ice has an endmember -30‰), consistent with results from Masson-

Delmotte et al. (2008), and all of the available $\delta^{18}$O precipitation data across West Antarctica. (IAEA/WMO, 2019)

In several prior papers using stable isotopes in a similar fashion, 150 m is indicated as the depth below which all, or nearly all meteoric water is presumed to be glacial melt (Randall-Goodwin et al., 2015; Biddle et al., 2019). With atmospheric influences minimized (or even eliminated), meteoric water below 150m consists almost entirely of glacial meltwater (Jenkins, 1999; Randall-Goodwin et al., 2015; Biddle et al., 2019).

In our 7 years of data, we see surface influence diverging from the mixing line as deep as 160m (2000). Including shallower depths, where data move off the mixing line produces a less negative intercept, as the mixture now incorporates sea ice melt and may also include less-depleted precipitation. We took a more conservative approach than previous studies, selecting 200m as the depth below which surface influences are minimized, to determine the mean meteoric water endmember from the seafloor up to the point where surface influences appear.

The relative enrichment of $\delta^{18}$O in surface waters relative to salinity is due to the influence of sea ice melt (which has a slightly positive $\delta^{18}$O) and some amount of precipitation (which has a less negative $\delta^{18}$O value than glacial meltwater). While surface waters will include precipitation, most of the surface meteoric water will still consist of glacial meltwater. (Meredith et al., 2008, 2010, 2013; Bett et al., 2020).

- Regarding the methodology, the authors suggest using a simple linear regression on all data below 200 meters depth to constrain the meteoric water endmember each sampling year. Upon initial examination of the $\delta18O$ /S plot, the data appear to form a single mixing line for salinities greater than 34. This is surprising considering the clear distinction in the T/S plot between waters influenced by melt and those composed of simple CDW/WW mixtures with WW characterized by salinity higher than 34 and temperatures near the surface freezing point.

$\delta^{18}$O is a distinct tracer from temperature, and plots against salinity reveal different things. Winter water does not have as distinct a fingerprint in $\delta^{18}$O-S space as it does in T-S space in this region (see Figure A1 in manuscript, where there is no $\delta^{18}$O distinction between points falling on the GMW mixing line vs the WW mixing line). Biddle et al. (2019), who discusses winter water in their noble gas and traditional hydrographic tracer study, do not attempt to determine a $\delta^{18}$O signature for winter water. The production of winter water is indirectly accounted for in the 3-endmember mixing model by the negative sea ice melt fractions (indicating sea ice formation).

The implication is that runoff and precipitation are less negative in δ18O than ice shelf basal melt. However, when sea ice formation increases surface water salinity to around 34, the δ18O/S of the WW ends up to the right of the meltwater mixing line influencing the CDW/GMW line estimated by the authors. The extrapolation of the meltwater mixing line to zero salinity gives a more negative intercept due to the influence of deep WW. Furthermore, the authors must acknowledge that the estimated freshwater content reflects contributions from large spatial and temporal scales, including meltwater from ice shelves upstream, icebergs within the Bellingshausen Sea and WAP, and precipitation inputs carried into the region by surface circulation.

There is no distinct salinity- $\delta^{18}O$ fingerprint for winter water in this region (see Figure A1 in manuscript). Work by Jenkins (1999, p.199) showed that winter water has a less negative $\delta^{18}O$ than waters influenced by GMW, in temperature-$\delta^{18}O$ space. The intercept method has been used to determine a glacial freshwater endmember in numerous other studies (Fairbanks, 1982; Jacobs et al., 1985; Paren and Potter, 1984; Potter et al., 1984; Hellmer et al., 1998; Jenkins, 1999).

Further, the intercepts produced using our method are consistent with local ice cores and glacial melt. Studies from further northeast show less negative intercepts, consistent with ice cores near those study areas (Meredith et al., 2008; Thomas et al., 2009; Meredith et al., 2010, 2013; Goodwin et al., 2016).

We conducted 2 exercises to assess the influence of winter water on our intercepts:

1) We applied an aggressive "winter water filter" to our data (removing all points with S>34 and $\Theta$ <-1.5°C) and recalculated our intercepts. In 3 years, the intercept was unaffected (2007, 2014, 2019), in 2 years, removing the "winter water" resulted in a *more negative* intercept (1994, 2000, 2020). Only in 2009 did the intercept value increase (by 0.3‰) after removing the "winter water" data from the regression. The largest change in intercept value was a decrease of 0.5‰ in 1994.
2) We selected only those data falling on the mCDW-GMW mixing line in T-S space, and then ran the salinity- $\delta^{18}O$ regression through only those data. Once again, our intercepts were unaffected.

Our uncertainty analysis applies uncertainty with a SD of 1.9 ‰ to our meteoric water endmember, in addition to changes in the meteoric endmember resulting from geographic sampling selection and instrumental precision (also accounted for) so any confounding influence of winter water is accounted for within our error envelope as defined by the SD. We have added discussion of winter water to our discussion, results, and appendices.

We have elaborated on meltwater input from upstream, however we note that the freshwater endmember identified using our intercept approach is most consistent with local sources.

The use of data below 200 meters excludes a portion of meteoric water in the surface and subsurface layers. Therefore, it is essential to consider the broader context of freshwater sources and their uncertainties. Distinguishing subsurface glacial meltwater, a meteoric input, from surface- derived meteoric inputs using $\delta18O$ is challenging, as isotopic values in local precipitation and ice shelf melt are similar.

Values in local precipitation and glacial ice melt are not similar. Masson-Delmotte et al.'s (2008) study of Antarctic precipitation presents an Antarctic-wide study of precipitation $\delta^{18}O$ where they show that 88% of the variance in precipitation $\delta^{18}O$ can be explained by latitude, distance from the coast, and elevation – with elevation being the most important factor. The value of our collected snow samples (-15‰) is consistent with the model presented in this study. As mentioned previously, many other studies demonstrate the ability to distinguish between local precipitation and glacial melt on the basis of $\delta^{18}O$ (Jacobs et al., 1985; Potter and Paren, 1985;

Meredith et al., 2008, 2010, 2013). This seems to be a recurrent sticking point for this reviewer – but they are demonstrably wrong on this point. We ask the editor to check with some experts and/or consult the literature. They will find that this statement by reviewer 1 is wrong.

The age and source of ice melting at the base of the ice shelves remain uncertain, further complicating the $\delta 18O$ analysis. Given these uncertainties, considering a larger range of plausible $\delta 18O$ meteoric endmembers based on regional literature would provide more realistic estimates with appropriate uncertainties.

No other papers employing similar methodologies (Meredith et al., 2008, 2010, 2013; Randall-Goodwin et al., 2015; Silvano et al., 2018; Biddle et al., 2019) use a range of $\delta^{18}O$ endmembers (though Biddle et al. use a similar uncertainty analysis). We vary endmember $\delta^{18}O$ determined by the regression by a SD 1.9 ‰ (on top of changes to the endmember from geographic variability, and perturbations to account for instrumental precision) in our uncertainty analysis. The age of the glacial ice is not a great cause for concern in this case – especially since we can estimate the $\delta^{18}O$ of the glacial melt using our intercept approach. We have consulted with several glaciology colleagues. The Antarctic cores that we cite go to bedrock (i.e., Epstein et al., 1970) and show reasonably consistent mean $\delta^{18}O$ values to about 1100m, whereafter they have more negative $\delta^{18}O$ on average. As described previously, salinity- $\delta^{18}O$ intercepts have been used by numerous studies to identify mean glacial meltwater endmembers.

For our purposes, the age of the ice itself is immaterial. We estimate the mean glacial meteoric water endmember from observational data; this will estimate the $\delta^{18}O$ of the mean meteoric water endmember being introduced at depth – the age of the ice does not matter. The values derived from our intercept calculations are consistent with the values from local ice cores. If we were to use a "plausible meteoric endmember from regional literature", we would use the values from those ice cores.

However, to account for the presence of precipitation more explicitly, our updated analysis includes the use of a different mean meteoric water endmember in the upper 200m, determined by a mixture of the glacial endmember and precipitation. Both the relative proportions of the GMW-precipitation meteoric water, and the $\delta^{18}O$ of precipitation are varied in the uncertainty analysis – on top of the uncertainty already preformed.

Finally, I also disagree with the authors' claim that Antarctic precipitation at sea level substantially differs from its values in continental precipitation and glacial ice in the region of their study.

We have addressed this concern previously. The data is clear on this point. The reviewer can disagree all they want but they are demonstrably wrong and is not appropriate for us to degrade the credibility of our paper by trying to address their miscomprehension of facts.

- In their mass balance calculation only three endmembers are used, and even I am a passionate user of this approach, it does make numerous assumptions, necessitating the consideration and propagation of errors. Among other water masses in the region, the authors also fail to address the presence of WW in their analysis, which can extend deeper than 200m.

Winter water is not explicitly addressed in the model, as it does not have a distinct salinity- $\delta^{18}O$ fingerprint in this region. However, the negative sea ice melt fractions are implicitly indicative of winter water – indeed sea ice melt and winter water are two sides of the same coin. Our study is intended to focus on glacial meltwater – it is not a hydrography paper - however we have pulled further sea ice discussion forward into the main body of the manuscript.

We conducted extensive uncertainty analysis, including environmental variability, instrument precision, and endmember uncertainty – testing the influence and propagation of errors from the raw sample data through to the output of the mixing model at every step. This was included in the first version of this manuscript that was submitted, and further uncertainties were introduced and propagated in the last submission. Very detailed accounts of the specifics of the uncertainty analysis are found in the Appendix, and the high-level results are clearly and prominently displayed on the yearly mean meteoric water inventory figure.

Our updated revision includes *further* uncertainty analysis, with the inclusion of precipitation with a varying fraction and $\delta^{18}O$ in the upper water column.

The use of 200 meters as the limit between the surface layer and the mCDW layer is questionable and requires clarification. Again, the authors rely on a single value of approximately -30‰ to represent the $\delta18O$ of GMW, which is weak given the accumulation of meteoric water in the WW layers from various sources, both local and remote.

200 m is not used as a "limit" between the surface layer and mCDW and was never described as such.

It is the depth below which we consider that meteoric water inputs will be nearly entirely glacial. Other studies in this region use 100m-150m as the depth below which waters are free of atmospheric influence, and therefore consisting of nearly pure glacial meltwater (Jenkins, 1999; Randall-Goodwin et al., 2015; Biddle et al., 2019). We chose to use 200 m as a more conservative estimate to identify the glacial meteoric water endmember. The same approach has been used by numerous other studies. (Fairbanks, 1982; Paren and Potter, 1984; Potter et al., 1984; Hellmer et al., 1998). This is very clearly described in the text.

We do not "rely on a single value" for our meteoric water (or any other) endmember. In our uncertainty analysis, the mCDW and meteoric endmembers change with each iteration based on randomized spatial selection of data, introduced perturbations to account for instrumental precision. On top of these perturbations, we varied the calculated endmember in each simulation randomly by a SD of 1.9‰ – the standard deviation of local ice core $\delta^{18}O$ (which, as noted, have very similar values to our calculated intercepts). In the most extreme cases, this means shifting the meteoric water endmember produced by the intercept regression by up to 7‰ - and again, the endmember produced by the regression changes in each iteration dependent on the random geographic selection of stations, and the perturbations to observations to account for instrumental precision. At the far end of the distribution, this results in possible perturbations of the meteoric endmember of up to 21‰! This is a much greater degree of uncertainty analysis, with a much wider array of endmembers tested than any of the published studies.

All of this has been described clearly in every version of the manuscript. The results and impacts of the uncertainty are explicitly described in the text, and the error bars in the mean meteoric water column inventory by year figure represent those analyses. Our updated analysis includes all the previously accounted for uncertainty, with the addition of precipitation – varied widely in both quantity and $\delta^{18}O$ – in the upper 200m.

The authors claim that the mean $\delta18O$ of the glacial melt may be represented by their methodology, but it does not offer a more accurate estimate of freshwater source fractions compared to previous studies by for instance Meredith et al. 2010 or Biddle et al., 2019 (but there are many other works), because without a proper GMW isotopic composition range, they might over or under estimate the freshwater source fractions.

We disagree. The results of the 3-endmember model used in this study, and by previous studies in the region (Meredith et al., 2008, 2010, 2013; Randall-Goodwin et al., 2015; Biddle et al., 2019) are *strongly* dependent upon the endmembers used – therefore more accurate endmembers will produce more accurate water fractions. We used the data collected in the study area to calculate the mean freshwater mixing endmember, while the other studies merely selected a plausible mean endmember. Using collected observational data to estimate the mean meteoric endmember directly allows meteoric water fractions derived from data collected from different labs without concern for potential calibration offsets – while if the same endmember is used with offset data, the meteoric water fractions *will* be significantly impacted. This is described in the main text, with further detail in the appendix. This is one of the key advancements of this study.

Our subsequent analysis has determined that >87% of the meteoric water input in this area consists of glacial meltwater. Extrapolations of $\delta^{18}O$-salinity data have been used to determine glacial endmembers by numerous other studies. (Fairbanks, 1982; Paren and Potter, 1984; Potter et al., 1984; Jacobs et al., 1985; Hellmer et al., 1998; Jenkins, 1999; Meredith et al., 2008; Randall-Goodwin et al., 2015).

Using a midpoint between precipitation and glacial meltwater as a mean meteoric water endmember (which intrinsically assumes a 50/50 composition of precipitation and glacial melt), as in Meredith et al. (2008, 2010, 2013); Randall-Goodwin et al. (2015) is less appropriate, and will overestimate meteoric water inputs – especially in deep waters, where meteoric water *will* be pure or near-pure glacial meltwater. Ensuring that endmembers are accurate is critical for determining source water fractions using this approach. Our approach better constrains the meteoric endmember using in-situ data, producing more accurate source fractions. This is one of the advances we highlight in this paper, but this reviewer does not seem to appreciate the nature of the advance. We have tried to make this point more clearly and prominently in our revision.

In our revision, we make an additional advance the method through use of a 2-component GMW-precipitation meteoric water endmember in the upper 200m, further improving the accuracy of the technique.

An invariant $\delta18O$ endmember is not a surprise as we would not expect GMW isotopic change in annual time scales.

We agree. This was included to show the reliability of the technique year-on-year, with data from different laboratories.

The primary source of uncertainty lies in the meteoric endmember, which dominates the uncertainty in the freshwater fractions. Furthermore, subsurface inputs of glacial meltwater can be differentiated from other freshwater sources based on potential temperature and dissolved oxygen concentration, making $\delta18O$ analysis less advantageous.

We disagree. We have significantly reduced the uncertainty associated with the meteoric water endmember here. if anything, $\delta^{18}O$ measurements when used for GMW estimation are more accurate and are at least additive to other tracers, including dissolved oxygen and noble gas concentrations. Further, traditional hydrographic tracers (and noble gases) cannot be used to estimate GMW in the upper water column due to influence from atmospheric processes and sea ice melt (Jenkins, 1999; Biddle et al., 2017, 2019). $\delta^{18}O$ has no such limitation, which is why $\delta^{18}O$-derived meteoric water estimates are used to calibrate new techniques to detect GMW in surface waters from satellites (Pan et al., 2023). We have improved this technique further in the revision, by including a realistic fraction of precipitation in the mean meteoric water endmember used there.

Since the authors concentrated solely on meltwater fractions in the main manuscript without discussing any sea ice influence, they could be calculated using the Gade line analysis. Therefore, I fail to see the advantage of employing $\delta18O$ in this context, again.

We have added a section on sea ice melt to the main body of the manuscript, pulling much of the sea ice melt discussion from the Appendix forward. The focus of our paper is on using a large $\delta^{18}O$ dataset (the largest ever, for a study of this kind – the unpublished data included in this study increases the *total* amount of Antarctic $\delta^{18}O$ data available by nearly 60%) to discuss glacial meltwater. Sea ice is discussed, but not a focus, and in any event is unlikely to impact our results. Recent advancements in remote sensing of GMW calibrate using $\delta^{18}O$ estimates because $\delta^{18}O$ can be used to estimate meteoric water in near-surface waters, which traditional hydrographic tracers cannot.(Pan et al., 2023)

- Another issue is the potential outlier or interannual variability in the year 1994. The claim of an increase in meteoric water inventories based on data spanning only seven years between 1994 and 2020, with limited observations in the initial year, lacks statistical significance.

Our geographic sensitivity and randomization analyses indicate that even with the small number of sample locations, that a meaningful average meteoric water inventory can be obtained from the 1994 dataset. However, the manuscript explicitly discusses the impact of the low inventory in 1994 as responsible for the trend (or lack thereof) and explicitly states that the data show greater interannual variability than trend in calculated meteoric water inventories. It is also mentioned explicitly that the uncertainty in 1994 may be artificially low due to excellent fit of the data, but relatively low number of samples.

The authors assert the stability of meteoric water inventories after 1994, but it's important to note that this conclusion is based solely on the data presented in this study, which spans only seven sampling years and may not fully represent the broader dynamics of meteoric water in the region.

We have been even more explicit in this latest revision that we are only discussing the data presented within it.

These findings also do not account for the significant 2012 event in the region, and overall, they suggest a relatively stable pattern. The absence of sensitivity tests on the linear regression is a concern, and the authors' reliance on visual assessment to justify the trend as an answer to my previous review is not scientifically robust.

We did perform extensive sensitivity tests on the linear regression (and the meteoric water fractions from which it is derived). Two different expressions of uncertainty are clearly displayed on the figure and discussed within the text. Details of the sensitivity and uncertainty analysis are in the main text, with further details in the supplement. The reviewer appears to have missed the presence of this text. We also note explicitly that the apparent trend is insignificant and based dependent on the low inventory in 1994.

- Additionally, it is unclear whether the authors are using problematic data in their analysis, likely stemming from sample storage issues. If these data points are problematic and from storage issues, they should not be discussed in the main results as they do not contribute meaningful insights to the study.

Great effort went into identifying and flagging problematic data, more than for any other seawater stable isotopic paper that we are aware of. This is *not* a problem with this data set. We suspect that this reviewer may not be well-informed on seawater stable isotopic analyses and QC'ing. We have, however, decreased the amount of discussion of this quality control process in the main text, instead directing readers who may be interested to the appendix.

L. 90: What does "scrutinized visually" entail or involve?

We elaborate on our sample and data QC for a portion of the 2009 data in the supplement to make the methods more explicit. When the water level in an unopened bottle is lower than its peers – it has likely been subject to evaporation. The text has been modified to make this point more clearly.

L. 96: If the comparison between CRDS and IRMS analyses revealed no discernible salt effect, it could be argued that this paragraph may not provide significant value to the current discussion and might be more appropriately placed in an appendix.

We added this based on earlier feedback from another reviewer who suggested that a salt effect may be impacting our results. In addition, the fact that the two methods yield similar results with seawater samples is not yet well-known or well published.

L. 140: I'm having difficulty comprehending the trends that the authors are discussing

We have multiple years in which we're calculating meteoric water fractions. Any change taking place across these years may be indicative of a trend. The fractions produced will be dependent on endmembers used in the model. We have tried to make this clearer in the revision.

L. 193: The characteristics of mCDW are derived from observations, while those of meteoric water are estimated rather than directly observed in this context

Meteoric water endmembers used in our study are estimated/calculated directly from our data, rather than using a largely hypothetical "plausible mean" value, as in other studies. We have addressed the issue with assuming a "plausible mean" meteoric water endmember both in the most recent submission, and in earlier responses. We have advanced our approach further by using a 2-component meteoric water endmember that includes precipitation in the upper water column.

Figure 3: If the authors assert that negative meteoric fractions, which are theoretically impossible, result from erroneously flagged data, they may need to consider either removing such data points and explain why from the analysis or adjusting the mass balance calculation to restrict negativity to only the sea ice term.

These do not result from erroneously flagged data. We used an average mCDW endmember. This means that some of the mCDW samples had less depleted $\delta^{18}O$ than our mean mCDW endmember (because that's how averages work). When you run the model on these sample points, the result is negative GMW. The same happens if the sample is saltier than the mCDW endmember. If we just use the max $\delta^{18}O$ and max S as our mCDW endmember, then no negative meteoric water fractions will be produced, but we do not feel that this would be appropriate.

All other published studies (Meredith et al., 2008, 2010, 2013; Randall-Goodwin et al., 2015; Biddle et al., 2019) using this technique also have negative meteoric water fractions at depth clearly visible in their figures, and in their data if you recalculate their fractions. They, however, do not acknowledge the negative fractions in the text. We carefully address this result, for any future readers who may notice what appears to be a theoretical impossibility.

L. 210: What is the rationale for discussing sea ice melt and mCDW fractions in the appendix? L. 219: What does the environmental variability of the model inputs entail?

Environmental variability is based on the range of possible values for each endmember, as all will vary naturally within the environment (e.g., mCDW is not a single S and $\delta^{18}O$ value, even within a single year, but rather varies slightly through space and time). The specific values by which each endmember was perturbed, and how they were derived, are described explicitly in Appendix A3, and referenced in the main text. "Environmental Variability" is the same terminology used in Biddle et al. (2019) for perturbation of the endmembers in their uncertainty analysis.

L. 233: I continue to be impressed by the superior analytical precision offered by CRDS compared to IRMS.

Thank you. We worked very hard over several years to develop methods and monitoring to achieve the level of precision and calibration we do using the CRDS. We're hopeful that the broader community can take advantage of our work, here.

L. 236: How do the authors go about estimating the uncertainty related to environmental variability? Additionally, while the term "environmental variability" is recurrent in the text, I'm still curious about the specific processes that the authors consider within this framework.

Described above.

L. 281: Is there any substantial evidence supporting the notion that the increased inventories in group B are indicative of basal melt accumulation, or is this interpretation more in the realm of speculation?

Modelling (Nakayama et al., 2019) and observational (Wåhlin et al., 2021) studies show the pathway of meltwater from PIG underneath Thwaites, before exiting beneath Thwaites on the NW.

L. 283: With the exception of those locations adjacent to the TIS, the sensitivity of sampling location remains a significant factor to consider.

While sampling location consistency will always be of some concern, the results of our spatial sensitivity and randomization analysis indicate that precise reoccupation may not be necessary to understand mean meteoric water content in the region. Strategic sampling could provide a meaningful mean inventory with relatively few sampling locations – furthering the utility of this technique for environmental monitoring.

L. 393: This is a speculative assertion.

Yes, it is. We have changed from "demonstrating the utility…" to "demonstrating the *potential* utility"

L. 397: The GMW inputs exhibit consistent standard deviations from 1994 to 2020. However, the authors do highlight a lower meteoric water content in 1994 and a higher content between 2000 and 2020.

It's not clear to us what the reviewer is asking.

**Referee #2**

I would like to thank the authors for considering the comments of both reviewers in the previous round. While I think that some aspects of the paper have improved and some concerns could be alleviated, one main concern raised by both reviewers, i.e. the novelty of the results and their implications, could have been given a bit more thought in the revision and the responses. I think that the paper remains rather technical without adding much discussion and understanding regarding the temporal variability of meltwater in the region, but certainly highlighting the usefulness of seawater isotopes and potential solutions to overcome certain issues. Therefore, I still think that these results should be published in principal, but my recommendation to think more about the implications still holds and might be further addressed in a revision (specific comment 1 below). While the revisions helped to clarify and better understand certain aspects of the analysis, they also brought up some new issues, that I think require further consideration in another major revision (specific comments 2-5 below). These issues concern in particular the interpretation of the sea ice melt fraction and the implication for the meteoric fraction in that balance, as well as the spatial analysis.

Specific comments:
1. I still think that putting the results a bit more in context would help. A particular example is the last sentence of the abstract. What does this imply? Could you add a last sentence with the implication of this result?

Thank you. We have added the following concluding sentence to the abstract, and manuscript to help strengthen and/or clarify our conclusions.

"The relatively long residence time in the SE Amundsen Sea allows changes in mean meteoric water inventories to diagnose large changes in local melt rates, and improved understanding of regional circulation could produce well-constrained glacial meltwater fluxes. Though local circulation, and melt-induced changes to circulation remain a source of uncertainty, the mean meteoric water inventories have remained relatively stable, and do not show a clear signal of accelerating melt rates. The advancement in our 2-component meteoric water technique improves the accuracy of the sea ice melt and meteoric fractions estimated from seawater $\delta^{18}O$ measurements throughout the entire water column and increases the utility for the broader application of these estimates."

2. Sea ice melt: I found the added discussion on sea ice melt/formation quite useful, but I also have some issues with the current interpretation:

a. The coastal/shelf regions around the Antarctic are well known as sea ice production sites, where ice is preferentially formed and then exported to the open ocean. Therefore, I would have expected a net sea ice formation signal from the region, as it is reported in the discussed paper by Biddle et al. (2019). I am surprised to see such strong positive signals in the column integrated sea ice melt. I understand that the samples come from the summer season. So a net sea ice melt at the surface can be expected. However, integrated over the water column, the signal should be close to zero or negative, unless sea ice is imported to the region or waters that are formed during sea ice formation in winter are not captured by these measurements. I understand that sea ice

melt is not the focus of this study, but the results show that either the meteoric water endmember is too low (leading to an underestimation of sea ice formation), or water masses that are formed in winter during sea ice formation are not captured by these measurements. At least, I would be much more cautious on the interpretation of this result, as it seems counterintuitive.

The strong positive SIM signals in 2007, 2009 and 2014 at depths well below the surface (>200m) both result from samples diverging significantly from the $\delta^{18}$O-S mixing line in the positive $\delta^{18}$O direction (consistent with evaporation), suggesting that they may have been subject to evaporation. In the case of the 2009 data, we excluded any samples run in 2019/20 that appeared anomalous. The 2014 data has already been published in (Biddle et al., 2019), and so we have left it untouched.

We would argue that our mean integrated sea ice fractions *are* close to zero, given the size of the uncertainty (Table A6). The SIM fractions at below the surface are slightly negative in most years, resulting in a largely negative net SIM fraction, spatially (Fig A7). In all years, except 2007 and 2020, most locations show negative net SIM, e.g., sea ice growth. We note that the mean SIM inventories were positive for all years except 2009 – this result was produced by integrating the Gaussian fit of the data with depth and may have been unduly influenced by some samples with particularly high SIM fractions near the surface. All of the samples in this study were collected in the Austral summer. Ackley et al. (2020) observed "surface flooding" of sea ice melt in the Amundsen Sea persisting through mid-February – much of which was advected in from the Bellingshausen Sea. This may explain our observations, particularly for 2020, where almost no locations show net sea ice formation.

In our revised analysis, we include a fraction of precipitation in the upper water column and use a less negative meteoric endmember for those data. This re-analysis decreases sea ice fractions, bringing them closer to zero – with half of the years showing small net sea ice growth, and the other showing small net sea ice melt. The revised analysis also shows a stronger sea ice formation signal in the upper 100 m, more similar to that produced by Biddle et al. (2019).

b. The issue might arise from the way that the meteoric endmember is defined in this study using the water samples below 200m. As these samples have the advantage of not being influenced by sea ice and snow melt or precipitation at the surface (as motivated in the paper), they could well be influenced by sea ice formation as sea ice is formed in winter and the brine mixes deep in the water column. Consequently, this signal could pull the regression line towards saltier values, which could result in an artificial lowering of the meteoric endmember derived from the regression line. I think this issue might require some attention.

While winter water does not have a distinct fingerprint in salinity-$\delta^{18}$O space (see Figure A1 in manuscript, where there is no $\delta^{18}$O distinction between the WW and GMW mixing lines on the T-S plot), work by Jenkins (1999) showed that winter has a *less* negative $\delta^{18}$O than waters influenced by GMW, in temperature-$\delta^{18}$O space. Many other studies have used regressions of $\delta^{18}$O-S data to accurately determine glacial meltwater endmembers. (Fairbanks, 1982; Paren and Potter, 1984; Potter et al., 1984; Jacobs et al., 1985; Hellmer et al., 1998; Jenkins, 1999; Meredith et al., 2008).

We performed 2 exercises to test the influence of winter water on our 0-salinity intercepts:

1) We applied an aggressive "winter water filter" to our data (removing all points with S>34 and $\Theta$ <-1.5°C) and recalculated our intercepts. In 3 years, the intercept was unaffected (2007, 2014, 2019), in 2 years, removing the "winter water" resulted in a *more negative* intercept (1994, 2000, 2020). Only in 2009 did the intercept value increase (by 0.3‰) after removing the "winter water" data from the regression. The largest change in intercept value was a decrease of 0.5‰ in 1994.
2) We selected only those data falling on the mCDW-GMW mixing line in T-S space, and then ran the salinity- $\delta^{18}O$ regression through only those data. Once again, our intercepts were unaffected.

Our uncertainty analysis uses a SD of 1.9‰ for the meteoric water endmember perturbation, on top of the changes in meteoric endmember resulting from instrumental precision and geographic sampling variability—much larger than any possible influence of winter water variability. Our meteoric water endmember as calculated using the 0-salinity intercepts is also remarkably consistent with the local ice cores, as noted in the manuscript. In our updated analysis, the upper water column meteoric water endmember is subject to further perturbation, widely varying both the fraction and $\delta^{18}O$ of precipitation.

c. As a consequence, I do not fully agree with the interpretation in lines 301 – 312. If anything, I would argue that in light of the net water column sea ice melt signal detected in this study, the higher estimate by Biddle et al. (2019), or some intermediate solution, seem more realistic.

While we maintain that the signal below 200 m is one of nearly pure GMW, we concede that waters in the upper water column may be influenced by less depleted freshwater sources. In our revised manuscript, we have re-run our analysis using a 2 component meteoric water endmember for depths shallower than 200m. Using the same approach used in the last manuscript to determine the fraction of total water column meteoric water consisting of GMW, we determine an approximate GMW fraction for only the upper 200m (~75%).

We use a precipitation endmember of -15‰, and to create a new compound meteoric water endmember between precipitation and the 0-salinity intercept, that we use at all depths shallower than 200m. In our updated uncertainty analysis, we vary the precipitation fraction by 5%, and the endmember by the standard deviation associated with snow collection at Halley Bay and Rothera Point. The 2-component meteoric water endmember approach results in a meteoric endmember ranging from -27.3‰ to -25.5‰, similar to the -25‰ endmember used by Biddle et al. (2019) and Randall-Goodwin et al. (2015).

This re-analysis results in higher meteoric water fractions in the upper 200m (where most of the meteoric water resides) and lower (often negative) sea ice melt fractions. While we feel that using a GMW endmember is most appropriate for depths well below the surface, using a precipitation-influenced endmember in the upper water column could produce a more accurate result for total meteoric water (and by extension, sea ice melt/formation).

d. I don't quite understand the reasoning (lines 648 – 655) that sea ice melt/formation signals might be an artifact of the potential evaporation from the bottles. Ultimately, one of the strength of using seawater d18O and salinity is to separate meteoric water and sea ice melt/formation signals. So, if there is an issue of evaporation in the bottles is should show up as a meteoric water signal and not a sea ice signal.

Water containing lighter isotopes (i.e. $^{16}O$) evaporates preferentially, leaving the remaining sample with less $^{16}O$ (higher relative $^{18}O$). Higher $\delta^{18}O$ values for a given salinity will produce increased (positive) sea ice melt fractions, and lower meteoric water fractions, since the result of the model is an overall mixture of meteoric water/sea ice melt/mCDW – if sea ice melt increases, meteoric water decreases, and vice versa – so ultimately, it will affect both. This can be clearly seen in both the meteoric water vs depth and sea ice melt vs depth profiles for 2014 – the data that fall off of the $\delta^{18}O$ -salinity mixing line in the positive direction (suggesting potential evaporation) directly correspond to those points showing anomalously high sea ice melt and anomalously low meteoric water fractions at those depths.

3. I have to say that I do not understand the unit of the meteoric water fraction e.g. in Tables 3 and 4. Why is it in g/kg? According to the definition in Eq. 1-3 it should be in %.

We used g/kg for consistency with other studies in the region using similar techniques (Jenkins et al., 2010; Jacobs et al., 2011; Randall-Goodwin et al., 2015; Jenkins et al., 2018; Biddle et al., 2019). Our 2014 data is the same data used in Biddle et al. (2019).

4. I do not understand what is shown in Table 4. Why do the results differ from Table 2 and 3 and what numbers should the reader trust? Wouldn't it make sense to report a single number that includes all the uncertainties (analytical, environmental, spatial)?

Table 2 shows the endmembers and associated uncertainty used in the "base" analysis.

Table 3 shows the results of uncertainty analysis for the model outputs (i.e. meteoric water fraction and integrated meteoric water inventory). Those results were produced by perturbing the endmember inputs by both analytical precision and environmental variability (for mCDW and meteoric water this is indicated in Table 2, and for sea ice melt we use values from the literature).

For Table 4, all of the data was left unperturbed, but we:
1. Randomly selected 3 stations
2. Calculated mCDW and meteoric water endmember using only those data
3. Calculated the mCDW, meteoric water and sea ice melt fractions for those stations using endmembers from [2]
4. Integrated the freshwater fractions with depth using the Gaussian fit (as we did with all of the data in Table 2)
5. Repeat steps 1-4 10,000 times
6. Record each of the results of [1-4]

The numbers in Table 4 are the averages and SD of each of the 10,000 simulations.

In our updated revision, we have streamlined our uncertainty discussion in the main body of the revised manuscript to show only a single inclusive measure of uncertainty, with detailed breakdowns of the attribution to each source of uncertainty in the appendix.

5. A) I found the spatial analysis that is being done quite informative. It appears that the spatial variability is larger than most of the temporal variability. This also suggests, that the small positive trend shown in Figure 4 may be an artifact or at least not coming from the Amundsen Sea glaciers.

Our sensitivity analysis indicates that endmember uncertainty has the greatest impact on mean meteoric water inventories (with spatial variability being a close second). While the results of the "grouping" analysis may suggest very large variability, this is almost entirely due to a few select locations with uncharacteristically high or low inventories – with the high inventories occurring alongside Thwaites Ice Tongue, and low inventories generally being the result of a sample depth resolution in stations furthest from the ice shelves. Our uncertainty analysis suggests that a meaningful mean may be obtained with a relatively small number of strategically located samples. We have some text to the discussion and conclusions to this effect.

None of the meltwater source regions b-d in Table 5 show a positive trend, i.e. an indication of increased melt from PIIS or TIS, or a large interannual variability. The overall variability in Figure 4, seems to be dominated by the variability in region a. So, what is driving the variability in region a?

With the exception of 1994 and 2009 (which use very little data, as indicated in the figure) There isn't significant variability in Region **a**. 2007 does appear higher, but given the size of the uncertainty around the inventories in each year, we don't think we can meaningfully draw any conclusions about spatial variability in this subregion.

I found it very difficult to interpret this result and as a consequence the result in Figure 4 and some of the overall findings of the paper. Could it be, that the variability observed in Figure 4 is dominated by variability induced elsewhere? Or could it be due to the variability in the source water mass, i.e. mCDW?

The error bars produced here include environmental and analytical uncertainty. Variability in melt observed here could be due to differences in melting upstream, or due to differences in mCDW transport toward the ice shelves. We were surprised that there seemed to be remarkably *little* variability, given the constraints of the study, and the extent to which we introduced and propagated errors in our uncertainty analysis.

B) I also do not understand how to reconcile the total region average meteoric water column inventory (Table 3 and 4) and the average values for the subregions for a given year (Table 5). In 2020, the total values in table 3 and 4 are 9.2 and 8.8 m, respectively. However, the regional values are all 8.7 m or lower. How is it possible that the overall average is higher than the regions separately?

In the regional analysis, we defined the mCDW and meteoric water endmembers (as described in the methods) in each sub-region using only data from that sub-region. As a result, each region uses slightly different endmembers than the analysis including all of the data. The endmembers used in each sub-region are detailed in Table A2.

In the case of 2020, the data from groups c & d had a salinity lower than the salinity max for the whole region. Both groups also produced a meteoric endmember slightly more negative meteoric endmember than the data for the region as a whole, and the intersection line between the S-max and the linear regression through the >200m S-$\delta^{18}$O data produced a slightly more negative mCDW $\delta^{18}$O endmember than using the whole region's data.

These three factors all produce slightly lower meteoric water fractions than what are produced using the entire region's data in aggregate to calculate endmembers from. For group a in 2020, mCDW does fall at the S-max for the region, however the linear regression through data also gives the mCDW a more negative $\delta18$O data than when the data for the whole region is used – once again resulting in slightly lower meteoric water fractions.

The case is much the same with the spatial sensitivity analysis, wherein using data from only 3 randomly selected stations most often produced mCDW endmembers that were less salty or more depleted than the region as a whole, and/or produced more negative meteoric water endmembers using the linear regression through the $\delta^{18}$O-S plots.

Had we used the same endmembers for each of these analyses, you would not see the apparent discrepancy, however the purpose of the exercise was to illustrate the differences produced using only data from those groups of stations (including for the endmember calculation). The size of the error bars produced, accounting for analytical precision, endmember uncertainty, and spatial variability all have complete overlap in the mean values.

Variation between these results is to be expected, but our analysis shows that they are remarkably similar. We have added additional discussion to the results to explicitly address this apparent anomaly.

Technical corrections:
- Lines 212-221: Despite the previous review comment, Figure 4 is still not referenced in this paragraph.

Apologies for this oversight – it has been corrected.

[revised manuscript text omitted]

---

## Author Response (AR3)

**Public justification (visible to the public if the article is accepted and published)**:
Dear Andrew Hennig and co-authors,

Thank you for having replied to the comments from the two reviewers. I think that the manuscript is now easier to read and clearer on the methods and results. There are nonetheless still a few minor points to address before publication as you can see below.

Best regards,

Nicolas Jourdain

Dear Dr. Jourdain,

Thank you for your thoughtful and detailed comments. We have revised the manuscript as described below, and hope that these changes will satisfy the requirements for publication.
* * *
Comments:

- L. 60: expand GMW.
Corrected

- L. 237 and caption of Fig. 4: I don't get why negative sea ice melt fractions would indicate regions of net sea ice formation. I would argue that it results from sea ice formation, possibly during previous months, but not necessarily that this location experiences net annual sea ice formation. Sea-ice formation and associated convection can indeed produce a signal in the subsurface and be followed by the opposite signal closer to the surface during the sea-ice melting season, whatever the sign of the annual net balance between sea ice melt and sea ice formation.
We have clarified the text here, more clearly describing the signals of sea ice melt and vs sea ice formation.

- Figs 3-4: given that the meteoric water fraction and the sea ice melt fraction are defined in [0-1] with no unit in equations 1-3, you need to clarify the g/kg by adding something like "g/kg, i.e., g of meteoric water per kg of seawater" in the captions.
Captions expanded to include g of meteoric water/sea ice melt per kg of seawater.

- The caption and labels of Fig. 4 would be clearer with "sea ice melt fraction" than with "sea ice melt".
Captions expanded to include g of meteoric water/sea ice melt per kg of seawater

- L. 268-270 "These results are consistent with recent studies showing an increase in basal melt through the 1990s, followed by relative stability and interannual variability from 2000 through 2020, with interannual variability that is larger than the increasing trend in meteoric water content". Please provide references for this. If you look at the hydrographic estimates of ice shelf melt rates summarised in Fig. 4 of Joughin et al. (2021), basal melt seems significantly higher in ~2009 than in the early 2000s. And 1994 appears as an average year, so what would explain such

low meteoric water column inventory for 1994?

Joughin, I., Shapero, D., Dutrieux, P. and Smith, B. (2021). Ocean-induced melt volume directly paces ice loss from Pine Island Glacier. Science advances, 7(43), eabi5738.

We have added a reference to Flexas et al., (2022) here, and expanded on the discussion in addressing your later comments re: L. 438-442.

- L. 275-276: replace "where icebergs are exported out of the study area" with "where icebergs melt out of the study area".

Corrected

- L. 358: "show mixing clear mixing lines".

Corrected

- L. 422: "we have estimate" -> we have estimated?

Corrected

- L. 438-442: Things are a bit less clear than what is suggested in this sentence. Some of the first references do not clearly support stable ice-shelf melt rates. It is not clear to me which part of Paolo et al. (2018) supports this statement. No melt rates were estimated by Dotto et al. (2019). You should probably cite Paolo et al. (2023) that shows some trend. Then, you pick up the modelling study of Flexas et al. (2022), but other models show different variability. Check for example Fig. S4 of Naughten et al. (2022).

Naughten, K. A., Holland, P. R., Dutrieux, P., Kimura, S., Bett, D. T. and Jenkins, A. (2022). Simulated twentieth century ocean warming in the Amundsen Sea, West Antarctica. Geophysical Research Letters, 49, e2021GL094566.

Paolo, F. S., Gardner, A. S., Greene, C. A., Nilsson, J., Schodlok, M. P., Schlegel, N.-J., and Fricker, H. A. (2023). Widespread slowdown in thinning rates of West Antarctic ice shelves, The Cryosphere, 17, 3409–3433.

We have removed the references to Dotto et al. (2019) and Paolo et al. (2018) here and expanded this section, discussing our results more explicitly in contrast to other studies, including Naughten et al. (2022) and Joughin et al. (2021). [L446-460 in revised manuscript].

- L. 490-491, in the conclusion: it may be worth reminding that a large part of the local precipitation is likely exported by sea ice without entering the ocean surface layer.

A sentence has been added here describing the export of precipitation on sea ice

- L. 706: "at had"

Corrected

- L. 749-750 "likely the resulted".

Corrected